# Dual CXCR4 and E-Selectin Inhibitor, GMI-1359, Shows Anti-Bone Metastatic Effects and Synergizes with Docetaxel in Prostate Cancer Cell Intraosseous Growth

**DOI:** 10.3390/cells9010032

**Published:** 2019-12-20

**Authors:** Claudio Festuccia, Andrea Mancini, Giovanni Luca Gravina, Alessandro Colapietro, Antonella Vetuschi, Simona Pompili, Luca Ventura, Simona Delle Monache, Roberto Iorio, Andrea Del Fattore, William Fogler, John Magnani

**Affiliations:** 1Department of Biotechnological and Applied Clinical Sciences, Laboratory of Radiobiology, University of L’Aquila, 67100 L’Aquila, Italy; mancio_1982@hotmail.com (A.M.); giovanniluca.gravina@univaq.it (G.L.G.); alecolapietro@gmail.com (A.C.); 2Multi-Factorial Disease and Complex Phenotype Research Area, Bambino Gesù Children’s Hospital, IRCCS, Viale di San Paolo 15, 00146 Rome, Italy; andrea.delf@gmail.com; 3Department of Biotechnological and Applied Clinical Sciences, Division of Radiation Oncology, University of L’Aquila, 67100 L’Aquila, Italy; 4Department of Biotechnological and Applied Clinical Sciences, Laboratory of Human Anatomy, University of L’Aquila, 67100 L’Aquila, Italy; antonella.vetuschi@univaq.it (A.V.); pompili.simona@virgilio.it (S.P.); 5Pathology Department, San Salvatore Hospital, 67100 L’Aquila, Italy; 6Department of Biotechnological and Applied Clinical Sciences, Laboratory of Applied Biology, University of L’Aquila, 67100 L’Aquila, Italy; simona.dellemonache@univaq.it; 7Department of Life, Health and Environmental Sciences, University of L’Aquila, 67100 L’Aquila, Italy; roberto.iorio@univaq.it; 8GlycoMimetics Inc., Gaithersburg, MD 20850, USA; wfogler@glycomimetics.com (W.F.); jmagnani@glycomimetics.com (J.M.)

**Keywords:** prostate cancer, CXCR4, E-selectin, GMI-1359, bone metastases

## Abstract

Metastatic castration resistant prostate cancer (mCRPC) relapses due to acquired resistance to docetaxel-based chemotherapy and remains a major threat to patient survival. In this report, we tested the effectiveness of a dual CXCR4/E-selectin antagonist, GM-I1359, in vitro and in vivo, as a single agent or in combination with docetaxel (DTX). This agent was compared to the single CXCR4 antagonist, CTCE-9908, and E-selectin antagonist, GMI-1271. Here we demonstrate that CXCR4 antagonism reduced growth and enhanced DTX treatment in PCa cell lines as well as restored DTX effectiveness in DTX-resistant cell models. The efficacy of dual antagonist was higher respect to those observed for single CXCR4 antagonism. GM1359 impacted bone marrow colonization and growth in intraventricular and intratibial cell injection models. The anti-proliferative effects of GMI-1359 and DTX correlated with decreased size, osteolysis and serum levels of both mTRAP and type I collagen fragment (CTX) in intra-osseous tumours suggesting that the dual CXCR4/E-selectin antagonist was a docetaxel-sensitizing agent for bone metastatic growth. Single agent CXCR4 (CTCE-9908) and E-selectin (GMI-1271) antagonists resulted in lower sensitizing effects compared to GMI-1359. These data provide a biologic rationale for the use of a dual E-selectin/CXCR4 inhibitor as an adjuvant to taxane-based chemotherapy in men with mCRPC to prevent and reduce bone metastases.

## 1. Introduction

Prostate cancer (PCa) remains a major health problem for men in industrialized countries [1]. Treatment with anti-hormone manipulation leads to androgen-independent (or castration resistant, CRPC) disease development, resulting in more aggressive and metastatic cancer mCRPC, [2]. Taxane administration represents the first-line chemotherapy for mCRPC [3], however, relapse eventually occurs, with development of chemo-resistant disease requiring the identification of new therapeutic approaches [4]. Local PCa that progresses and invades outside the gland preferentially metastasizes to bone [5]. The formation of micrometastases and the subsequent growth of macroscopic tumours results in bone pain and potentially pathologic fractures. Bone metastases are typically mixed osteoblastic/osteolytic lesions [6]. Commonly, bone lysis to osteosclerosis takes place in the most advanced metastatic stages on the basis of changes in the equilibrium between osteogenic and lytic factors. The specific mechanisms that promotes metastasis to bone are not completely understood, however, it has been shown that alterations in tumour cell adhesion to the bone marrow affect the metastatic potential of PCa cells. This suggests that the bone microenvironment is an important player in the metastatic process. The bone endosteum is comprised of a layer of cells lining the internal trabecular bone characterized by the presence of osteo-progenitor stem cells, resting and active osteoblasts, and osteoclasts. This represents the osteoblastic niche, which has been shown to be important for hematopoietic stem cell self-renewal [7] and hypothetically for bone metastases. E-selectin is a transmembrane endothelial cell adhesion protein that is recognized by E-selectin binding ligands present on tumour cells. These molecular interactions mediate tumour cell rolling on endothelium [8], and overexpression of E-selectin has been associated with tumour angiogenesis and metastasis in several cancers [9,10] making it an attractive candidate to target for the inhibition of tumour metastases and angiogenesis [11]. PCa cells preferentially adhere to bone marrow endothelial cells expressing elevated levels E-selectin when compared to endothelial linings from other tissue microvessels [10,11]. Another important mediator of cancer cell homing and bone metastases is the G-protein-coupled chemokine receptor, CXCR4. [12,13]. Levels of ligands for CXCR4 (SDF-1 and MCP1) are high in bone marrow. The clinical relevance of the expression and activity of CXCR4 in PCa remains controversial [14,15,16,17]. Silencing or inhibition of CXCR4 may be able to block metastatic spread [15,18,19] in animal models. Similarly, small molecules blocking CXCR4 activation modulate the tumour/stem cell niche as shown by the CXCR4 antagonist plerixafor (AMD3100) [20]. Increased expression of CXCR4 may be observed in specialized tumour niches [21]. The SDF-1α/CXCR4 axes triggers the adhesion of PCa to bone marrow endothelial cells activating β_3_ integrins [22,23] and selectin [24]. CXCR4 expression has been also associated with increased angiogenesis [25], secretion of vascular endothelial cell growth factor (VEGF) [26], interleukin-8 (IL-8) [27], and MMP-9 [28], facilitating tumour invasiveness and metastatic progression in the bone. In the present report, we evaluated the anti-osteolytic effects as well as the chemo-sensitizing activity versus docetaxel (DTX) of a novel compound targeting both E-Selectin and CXCR4, GMI-1359 by using in vitro and animal models of CRPC. This compound has been compared with the single CXCR4 (CTCE-9908) and E-selectin (GMI-1271) antagonists. One major obstacle to the conducting the most clinically relevant prostate cancer (PCa) research has been the lack of cell lines that closely mimic human disease progression [29]. Two hallmarks of metastatic human prostate cancer include, as indicated above, the shift of aggressive PCa from androgen-sensitivity to an androgen insensitive (AI) state, and the propensity of PCa to metastasize to bone. For the in vitro studies we use CRPC models with high (LnCaP [30], C4–2B [31], VCaP [32] and PC3 [33]) and low (22rv1 [34] and DU145 [35]) bone tropism. Literature data indicate that 22RV1, C4–2B and VCaP cell lines [29,36,37,38] produce mixed osteosclerotic/osteolytic lesions when injected intratibially (IT). These cells, however, have a very low intratibial engraftment rate and are unable to generate bone metastases in mice when injected by an intra-cardiac (IC) route. DU145 cells which are derived from a brain metastases induce osteolysis [29]. Similarly the PC3 cell line, instead, and its more bone metastatic PCb2 cell derivative [29,39], exclusively induce massive osteolysis. Thus, we selected the 22rv1 cells only for the subcutaneous model, the PC3 cells for the IT model and the PCb2 for the IC model. In the literature, the 22rv1 cell line is, indeed, a widely used to mimic an aggressive but non-metastatic CRPC disease. 22rv1 xenografts show elevated angiogenesis and inflammation. CXCR4 and E-selectin participate in the recruitment of endothelial precursors [40], inducing chemotaxis of human endothelial cells [41] in vitro and modulating angiogenesis in vivo [42].

## 2. Materials and Methods

### 2.1. Reagents and Drug Preparation

All materials for tissue culture were purchased from the Euroclone group (Milan, Italy). Antibodies including anti-human E-selectin H-300 [sc-14011] were purchased from Santa Cruz (Santa Cruz, CA, USA). Anti-human E-selectin lidands (sLe^a^ and sLe^a/^sialyl Lewis X) moAb HECA-452 was purchased from BD Biosciences (BD Italia, Milan, Itlay). The dual E-selectin/CXCR4 antagonist GMI-1359 and the E-selectin specific antagonist were designed and synthesized by GlycoMimetics, Inc. (Rockville, MD, USA). CTCE-9908 was kindly provided by Chemokine Therapeutics, Inc. (Vancouver, BC, Canada). Docetaxel was purchased from Selleck Chemicals (Aurogene, Roma, Italy). ELH-Eselectin1 (Soluble) Human ELISA Kit was purchased from Ray Biotech through the Italian distributor (Prodotti Gianni S.p.A, Milan, Italy). Phospho-CXCR4 (Ser339) Colorimetric Cell-Based ELISA Kit (OKAG01771) was purchased from Aviva Systems Biology, Corp (San Diego, CA, USA).

### 2.2. Cell Lines

PC3, LnCaP, C4–2B and DU145 cells were obtained from ATCC (LCG standard Italian distributor, Milan, Italy, USA). 22rv1 cells were obtained from the Leibniz Institute DSMZ (German Collection of Microorganisms and Cell Cultures, Braunschweig, Germany). PC3 bone derivative, PCb2 was generated in our lab through serial selection of excised tumors with bone tropism in nude mice and partially characterized in a previous our study [43]. Murine bone marrow derived stromal cells were obtained. Cells were authenticated by STR profiling using GenePrint^®^ 10 System (Promega Corporation, Madison, WI, USA). Murine CAFs were obtained from cultures obtained from 22rv1 xenografts. Tissue samples were carefully dissected, minced with crossed scalpels and cultured in DMEM supplemented with 5% fetal bovine serum, 0.5 μg/mL R1881 (Sigma-Aldrich, Italian distributor, Milan Italy), 5 μg/mL insulin (Sigma-Aldrich), and 1% penicillin/streptomycin. After 1 week, cells that migrated from the tissue clumps were trypsinized, transferred to new culture dishes, and allowed for surface attachment for 1 to 10 min. Cells that were not attached were then removed, and the dishes with firmly attached cells were washed with PBS to further eliminate loosely attached cells. Based on the observations of the cell morphology, the cultures that contained most homogenous fibroblastic-like cells were expanded and used for experiments [36]. Bone marrow stromal cells were seeded onto 12- or 24-well culture dishes and then allowed to grow to confluence when (d 0), cells were transferred to α-MEM containing 10% FCS and cultured as previously described [7,44,45]. The medium was changed initially at day 4 and then every other day thereafter until the cultures reached confluence. Osteoblast-like MC3T3-E1 cells were kindly provided by Dr. G Tulipano (University of Brescia, Brescia, Italy). Osteo-derived cells were seeded onto 24-well culture dishes and then allowed to grow to confluence. Conditioned media from untreated and treated PC3 and C4–2B cells were used at 30% in the presence of α-modified eagle medium (αMEM) containing 10% fetal calf serum (FCS) or mineralized medium (αMEM containing 10% FCS, 100 µg/mL ascorbic acid, and 5 mM glycerol-2-phosphate) for an additional 7–21 days. An indirect quantification of osteoblasts was also used by Alizarin Red S Staining Quantification Assay (ARed-Q; ScienCell, Carlsbad, CA, USA). Osteoblast phenotype was evaluated by alkaline phosphatase (ALP) activity kit (Sigma-Aldrich) and calcification reactions were stained von Kossa’s silver nitrate method. The methods of the staining were briefly as follows. After fixation of the cells on the bottom of wells, calcified product in the wells was soaked in 5% silver nitrate solution for 60 min. Then, under sunlight, the product was reacted with 5% sodium thiosulfate (Wako Pure Chemical Industries, Ltd., italian distributor, Milan, Italy). Moreover, the cells were counterstained with Kernechtrot solution for 5 min. RAW 264.7 was kindly provided by Prof E. Tolosano, (University of Torino, Torino, Italy). Cells were grown as described in the original reports. Conditioned media were harvested from sub-confluent cultures treated with serum free fresh medium for 24

### 2.3. Generation of Docetaxel Resistant Cells

DTX-resistant CRPC cell lines (designated as DTXR) were generated by serial desensitization of cells as previously described [46,47]. Cells were initially cultured in 1 ng/mL DTX and maintained until the DTX-sensitive clones died. The surviving PCa cells repopulated the flask and continued to divide through four passages. This process was repeated using a DTX concentration of 5 ng/mL and subsequently 10 ng/mL. Once PCa cells were freely dividing in 10 ng/mL DTX medium, they were considered resistant. The IC50 values for DTX calculated in these cells revealed the follow changes: DU145 cells went from 32,2 nM to >100 nM; PC3 cells went from 32.3 nM to 63.4 nM; 22rv1 went from 10.7 nM to 42.0 nM and C4–2B went from 27.4 nM to 78.5 nM.

### 2.4. Facs Analysis

Expression of surface antigens in PCa cell lines was quantified by flow cytometry. Cells were fixed with 4% paraformaldehyde for 10 min at 4 °C and, after washing, cells were incubated for 1 h at RT with anti-CXCR4 and HECA-452 followed by additional 30 min with CY5-conjugated anti-mouse IgG purchased from Abcam (Cambridge, UK). All samples were analyzed using a BD AccuriTM C6 Plus Flow cytometer (Becton Dickinson Italia SpA, Milan, Italy) equipped with a blue laser (488 nm) and a red laser (640 nm). At least 10,000 events were acquired. Negative controls were obtained analyzing samples treated without the primary antibody. To compare the expression of HECA-452 and CXCR4 in different cell lines, we considered the parameter “Mean Fluorescence Intensity (MFI)” which is the average intensity of fluorescence of the sample in exam considering all cells of interest divided by the fluorescence intensity of the relative control (a non relevant immunoglobulin labeled with similar fluorochrome as the staining antibody). The MFI [48] was calculated for each samples. Practically MFI measure the shift of the fluorescence peak with respect to the control: i.e., a value of MFI ≤1 indicates that the cell population is scarcely o is negative for the examined marker since the two fluorescence plots are similar. An MFI = 2 indicates that the fluorescence intensity of the given marker is two times greater than the threshold/negative value. Values are the mean values ± standard deviations of single experimental conditions obtained analyzing the average values of MFIs calculated in three separate analyzes at FACS.

### 2.5. Growth Assays

Cells were seeded at a density of 2 × 10^4^ cells/mL in 24 well plates. Cells were left to attach and grow in 5% FCS DMEM for 24 h. After this time, cells were maintained in the control or experimental culture conditions for the considered time. Morphological controls were assessed daily with an inverted phase-contrast photomicroscope (Nikon Diaphot, Tokyo, Japan), before cell trypsinization and counting. Cells trypsinized and resuspended in 1.0 mL of saline were counted using the NucleoCounter^TM^ NC-100 (automated cell counter systems, Chemotec, Cydevang, Denmark). The effect on cell proliferation was measured by taking the mean cell number with respect to controls over time for the different treatment groups as described [49,50]. Results were represented as data from three independent experiments performed in triplicate. IC50 values were calculated by using the GraFit (Erithacus Software Ltd., Staines, UK) plotting the percentage of inhibition versus drug concentration in a semi-logarithmic way.

### 2.6. Western Blot

Cell extracts from treated and untreated cells were electrophoresed under reducing conditions and transferred to nitrocellulose filters (Schleicher and Schuell GmbH, Dassel, Germany). Blots were incubated according to standard protocol and visualised by chemo-luminescent detection kit (Supersignal, Perbio Science, Tattenhall, UK) in the Bio-Rad gel Doc^TM^ (Bio-Rad Laboratories S.r.l., Milan, Italy).

### 2.7. Dot Blot

Tissue extracts from untreated or treated 22rv1 xenografts were loaded (200 µg/dot) in a Slot Blot Manifold apparatus (Sigma-Aldrich) and transferred in a nitro-cellulose paper and treated as below described for western blot. Dots were converted at 16 bit in grayscale and analyzed by Image J (University of Wisconsin-Medison. WI, USA) with the attribution of arbitrary units. 

### 2.8. Subcutaneous Xenograft Model

Male CD1 nude mice (Charles River, Milan, Italy) were maintained under the guidelines established by our Institution (University of L’Aquila, Medical School and Science and Technology School Board Regulations, complying with the Italian government regulation n.116 27 January 1992 for the use of laboratory animals). All mice were anesthetized with a mixture of ketamine (100 mg/kg)/xylazine (5 mg/kg) in saline and subsequently received S.C. flank injections of 1 × 10^6^ PC3 and 22v1 cells and DTX resistant strains. Tumour growth was assessed by bi-weekly measurement of tumour diameters with a Vernier calliper (length × width). Randomization was performed when subcutaneous tumors reached volumes ranged between 80 and 100 mm^3^. This was commonly obtained 7–10 days after cell injection. Tumour weight was calculated according to the formula: TW (mg) = tumour volume (mm^3^) = d2 × D/2, where d and D are the shortest and longest diameters, respectively. The effects of the treatments were examined as previously described [50]. 

### 2.9. Evaluation of Treatment Response In Vivo

In order to get closer to the parameters used to analyze thee pharmacological efficacy assessments in the man, we quantified the antitumor effects of different treatments as previously described [15,44,45,50]. Briefly: (1) tumor volume, measured throughout the experiment, (2) tumor weight, measured at the end of experiment; (3) complete response (CR) defined as the disappearance of the tumor; (4) partial response (PR) defined as a reduction of greater than 50% of tumor volume with respect to baseline; (5) stable disease (SD) defined as a reduction of less than 50% or an increase of less than 100% of tumor volume with respect to baseline; (6) tumor progression (TP) defined as an increase of greater than 50% of tumor volume with respect to baseline; (7) time to progression (TTP) defined as the time necessary to have progression. These modalities of analysis reduced both the differences of single tumor volume measurements in the time linked to differences of engraftment efficacy of the tumor cells as well as the individual variability of the response (even though the mice For the in vivo analysis, the synergy/additivity index (combination index, CI) may be calculated in different manner i.e., by using relative risk (RR), odds ratios (OR), or hazard ratios (HR). We considered the HR values through the formula: CI = [HR(a) + HR(b)]: HR (a + b) as described [51] where a and b represent the drug 1 and 2 when two compounds are combined. CI > 1.3 indicates antagonism, CI = 1.1 to 1.3 moderate antagonism, CI = 0.9 to 1.1 additive effects, CI ≤ 0.9 synergism.

### 2.10. Intracardiac (IC) Tumour Model

Briefly, anesthetized animals received cardiac injection of 1 × 10^5^ PCb2 cells in 0.1 mL of PBS was performed as previously described [39,50]. In the intracardiac tumour model, treatments were started 2 days before tumour cell injection and stopped after 50 days. Animals were sacrificed by carbon dioxide inhalation 70 days after heart injections, or earlier if there were early signs of serious distress. All animals were subjected to an accurate necroscopy and portions of various organs were processed for routine histological examination. 

### 2.11. Intratibial (IT) Tumour Model

Intratibial tumour injection was performed as previously described [39,50]. Briefly, anesthetized animals received 1 × 10^5^ luciferase transfected PC3 cells/10 µL PBS which were inoculated in the tibiae. Treatments were started 2 days after tumour injection and stopped after 28 days the end of drug administration.

### 2.12. Assessment of Treatment Response in Bone Tumour Models

Tumour and bone treatment response was determined using in vivo and ex vivo evaluations. The development of metastases was monitored by radiography using a Faxitron cabinet x-ray system (Faxitron x-Ray Corp., Wheeling, IL, USA) and lytic units were quantified by Image J analysis performed on lytic lesions as arbitrary densitometric units. Mice were administered 150 mg/kg body weight luciferin (Promega, Italian distributor, Milan, Italy) subcutaneously and imaged 15 min later in a bioluminescence imager (Hamamatsu Photonics Italy S.R.L, Arese, Italy) to identify intra-tibial implants similar to the method described by Kemper et al. [52]. The mice were photographed while placed on their front. Imaging data were normalised to the acquisition conditions and expressed as radiance (photons/second/cm^2^/steradian (p/s/cm^2^/sr)), and the colour scale was adjusted according to the strength of signal detected by using wasabi software, a Hamamatsu dedicated software. This semi-quantitative analyses [53] allows to us to determine the bioluminescence intensity (BLI) which was measured in the region of interest. The BLI values were used to calculate the BLI increment for each individual animal. Mice were euthanized when they displayed distress signs (eg, altered gait, tremors/seizures, lethargy) or weight loss of 20% or greater of pre-surgical weight. All animals were subjected to an accurate necroscopy to document secondary sites of metastases.

### 2.13. Treatments for In Vivo Experiments

Before the start of treatment, xenografts were randomised into eight groups as follows: Group 1: mice (10 animals) receiving intraperitoneal (i.p.) injections of 100 µL PBS; Group 2: mice (10 animals) receiving GMI-1359 twice a day i.p. at 40 mg/kg for consecutive 14 days [54]; Group 3: mice (10 animals) receiving CTCE-9908 at 25 mg/kg/day IP [55]. Group 4: mice (10 animals) received GMI-1271 25 mg/Kg/day for 28 days [54,56,57]; Group 5: mice (10 animals) receiving docetaxel alone (DTX i.p. injection of 7.5 mg/kg per week [47,57,58]); Group 6: mice (10 animals) received CTCE-9908 plus DXT at doses above mentioned for single treatments; Group 7: mice (10 animals) receiving GMI-1359 plus DTX at dose mentioned above for single treatments and Group 8: mice (10 animals) receiving GMI-1271 plus DTX at doses mentioned above for single treatments. The choice of these doses derived from the literature data. Groups 1–4 and groups 1–8 were considered for the IC and IT experiment, respectively. Measurements of serum mouse cross linked C-telopeptide of type I collagen (CTX-I, USCN Life Science Inc., Houston, TX, USA) and mouse TRAP-5b (mTRAP, MyBioSource, Inc., San Diego, CA, USA) were performed according to the manufacturer’s protocol on blood plasma. Cachexia was also considered and analyzed. This is a complex syndrome associated with an underlying illness causing ongoing muscle loss that is not entirely reversed with nutritional supplementation. A body weight loss of 15–20% is indicative for cachecsia. This is associated to anorexia, curved spine and absence or scarcity of responses to external solicitations. All these aspects were evaluated during the experiments and accounted at the end of analyses. A chi square test analyses for trend followed by Bonferroni correction.

#### Immunohistochemical Analyses

Indirect immunoperoxidase staining was performed on 4 μm paraffin-embedded tissue sections from human and experimental xenograft and intra-bone experimental tumors. The immunohistochemical analysis was also performed on tissue arrays from primary tumours (46 cases) and bone metastases (8 cases) from US Biomax (Rockville, MD, USA). Immunohistochemical staining was carried out using Rabbit anti-CXCR4 polyclonal antibody (GenScript); anti-human E-selectin H-300 [Santa Cruz] and Anti-human E-selectin ligands (sLea and sLea/sialyl Lewis X) moAb HECA-452 was purchased from BD Biosciences as described above. In brief, formalin-fixed, paraffin-embedded specimens were deparaffinised in xylene and dehydrated with ethanol. Endogenous peroxidase was blocked with 0.1% hydrogen peroxide–methanol for 30 min at room temperature. After washing with distilled water, the specimens were incubated in a microwave oven in target retrieval solution (Dako, Glostrup, Denmark) for 10 min and then washed with distilled water. Non-specific binding was blocked by treatment with a special blocking reagent for 15 min. Anti-CXCR4 antibody was applied at a dilution of 1:100, and the sections were then incubated in a moist chamber overnight at 4 °C. Localisation of CXCR4 protein was performed using a HRP Conjugated anti-rabbit IgG Super Vision Kit (Italian distributor of Boster Immunoleader USA, Tema Ricerca, Bologna, Italy) and visualised by the Pierce DAB Substrate Kit (Thermo Scientific, Italian distributor Tema Ricerca) for 5 min at room temperature. Cells were counterstained using Meyer’s haematoxylin. The evaluations were recorded as the percentage of positively stained tumour cells in each of three intensity categories. A consensus judgment was adopted as to the proper immunohistochemical score of the tumours based on the strength of CXCR4 expression: negative, weak staining, moderate staining, or strong staining. In the present study, as in a previous study, the distribution of positive cells was also recorded to portray the diffuse or focal nature of the positive cells: sporadic (positive cells <5%); focal (positive cells >10% but less than 50%); or diffuse (positive cells >50%).

### 2.14. Statistical Analysis

Continuous variables were summarised as mean and S.D. or 95% CI for the mean. Statistical comparisons between controls and treated groups were established by carrying out the ANOVA test or by Student’s t test for unpaired data (for two comparisons). Dichotomous variables were summarised by absolute and/or relative frequencies. For dichotomous variables, statistical comparisons between control and treated groups were established by carrying out the exact Fisher’s test. For multiple comparisons, the level of significance was corrected by multiplying the P value by the number of comparisons performed (*n*) according to Bonferroni correction. All tests were two-sided and were determined by Monte Carlo significance. P values at least <0.05 were considered statistically significant. In the figures in which statistical analysis was performed, significance was indicated by an asterisk. SPSS (statistical analysis software package, IBM Corp., Armonk, NY, USA) version 10.0 and MedCalc statistical software for biomedical research (Ostend, Belgium) were used for statistical analysis and graphic presentation. Overall survival was analysed by Kaplan–Meier curves and Gehan’s generalised Wilcoxon test. When more than two survival curves were compared, the logRank test for trend was used. This tests the probability that there is a trend in survival scores across the groups.

## 3. Results

### 3.1. CXCR4 and HECA-452 Immune-Reactivity in Prostate Cancer Cells

Before to study the effects of GMI-1359 in comparison with CTCE-9908 and GMI-1271, we evaluated the expression of the respective targets (CXCR4 and E-selctin ligands) by western blot and FACS analyses. In Figure 1A we show FACS results for CXCR4 and HECA-452 immune-reactivity (expressed as Mean Fluorescence index) in different prostate cancer cell lines. Figure 1B demonstrates that the expression of CXCR4 as well as the IR versus HECA-452 were higher in cancer cells derived from bone metastases (PC3, PC3b, C4–2B and VCaP) when compared to prostate tumour cell lines derived from primary (22rv1), lymph-node metastasis (LnCaP) and brain metastasis (DU145). The immune-reactivity levels were expressed as mean fluorescence index (MFI) ± SD. The MFI values for CXCR4 were 6.22 ± 0.90 for bone metastatic and 2.63 ± 0.95 (*p* = 0.0434) for non-bone metastatic PCa cells. This was in agreement with a previous report [15]. Conversely, the IR versus HECA-452 resulted not statistically different (*p* = 0.4680 NS) in bone metastatic (2.42 ± 0.57) or non-bone metastatic PCa cell models (1.73 ± 0.67). Next we verified if CXCR4 or HECA-452 levels were amplified by conditioned media collected from carcinoma associated fibroblast (mCAF) as well as by exogenous SDF1α 10 ng/mL in non-metastatic (22rv1) and bone metastatic cells (PC3) cells, chosen as models (see above). We found that MFI values for CXCR4 increased significantly in 22rv1 treated with CAF (2.5-fold) and SDF1α (2.0-fold) with marginal effects on PC3 cells (Figure 1C). It is necessary to remember that the basal levels of CXCR4 were higher in PC3 cells. Similarly, in Figure 1D we show that HECA-452 levels were significantly increased in the 22rv1 cells after administration of both conditioned media derived from mCAF (1.77-fold) and SDF1α (2.22-fold). HECA-452 induction in PC3 cells was minimal for mCAF and significantly higher for SDF1α (1.56-fold). 

In order to verify if the immune-reactivity for CXCR4 and HECA-452 was modified in the presence of conditioned media from bone derived cells, we analyzed the effects of three bone derived cell populations such as: (i) murine bone stromal cells (BMS); (ii) murine osteoblast-like MC3T3-E1 cells (OB) or (iii) RAW-264.7 (osteoclast precursor model). In Figure 1E we show that the administration of bone derived conditioned media induced CXCR4 expression mainly in PC3 in which OB-CM, BMS-CM and RAW-CM increased the levels of CXCR4 of about 1.58-, 1.84- and 1.32-fold. CXCR4 induction in 22rv1 cells were not statistically significant for the administration of CMs derived from BMS, OB whereas the increment of CXCR4 was 2.0-fold in presence of conditioned media from RAW cells. Next we analyzed the modification of HECA-452 immune-reactivity in the same cells. When PC3 and 22rv1 cells were triggered with bone derived conditioned media we observed that the immune-reactivity of HECA-452 was induced in PC3 of about 1.86 (OC-CM), 2.14 (BMS-CM) and 3.21 (RAW-CM). Increments of HECA-452 positivity were lower and not statistically significant in 22rv1 except for BMS-CM with 1.56-fold increase (Figure 1F).

### 3.2. Docetaxel (DTX) Increases CXCR4 Expression in Docetaxel Sensitive and Resistant Cells In Vitro

This compound is the first chemotherapy agent approved for treatment of mCRPC but the limited survival benefit associated with DTX administration and the development of resistance typify the need for combination treatments with diminished systemic toxicity and increased efficacy. It has been hypothesized that DTX induced expression and/or activation of CXCR4 in solid tumors, which in turn is able to increase pharmacological resistance [59], so we tested if non-cytotoxic concentrations of DTX (20 nM) were able to increase the expression of CXCR4 in PC3, DU145 and 22rv1. In Figure 2A we show that CXCR4 protein expression by western blot analysis was increased after administration of DTX in a time dependent manner. The FACS analyses performed at 96 hr (Figure 2B) confirm western blot analyses and show that MFI values for CXCR4 were significantly higher only in DTX-treated cells of about 1.75 (22rv1), 1.25 (DU145) and 1.83 (PC3). Interestingly, western blot and FACS analyses show higher CXCR4 expression levels in DTX-resistant cells (Figure 2C, D) with 1.88- (22rv1), 1.43- (DU145) and 2.50- (PC3) fold increases.

Differently, the HECA-452 immune-reactivity was not different in parental cells treated with DTX except for the PC3 cells in which DTX induced an increment of about 1.48-fold (Figure 2E) as well as in DTX resistant clones (Figure 2F). Next we evaluated if the sensitivity against DTX was related to bone tropism of PCa cell lines. In Figure 3A we show a trend of lower DTX-sensitivity (higher IC50) in cells derived from bone metastases (PC3, PC3b, C4–2B and VCaP) when compared to cell lines derived from primary (22rv1), lymph-node (LnCaP) and brain (DU145) with an IC_50_ (mean ± SD) of 34.4 nM ± 4.93 vs. 17.37 nM ± 5.89 (*p* = 0.0755 NS). Correlation analysis between CXCR4 expression and IC_50_ values indicated a significant value (r = 0.7351; *p* = 0.0099). This suggests that CXCR4 expression modulates DTX sensitivity/resistance also in PCa models. Although we found only marginal effects on HECA-452 IR in DTXR clones, we wanted to verify if a correlation between HECA-452 and IC_50_ for DTX was present. We observed that no statistically significant correlation was evident (r = 0.4011, *p* = 0.3725 NS). A multiple regression was also performed considering IC_50_ as dependent variable of both CXCR4 and HECA-452 expression. No significant contribution of both antigens in DTX sensitivity was found with a coefficient of determination R^2^ = 0.1629 and *p* = 0.7007 (NS). In addition we verified if SDF1α administration (10–100 ng/mL) modulated the DTX sensitivity in 2rv1, DU145, PC3 and C4-2B. We find that the sensitivity to DTX was reduced with exogenous SDF1α (Figure 3B). The effects were more marked in 22rv1, C4–2B and DU145 cells when compared to PC3 cells. In Figure 3C we show, also, that DTX-resistant cell derivatives showed IC_50_ values significantly higher than of their DTX-sensitivity parental cells.

### 3.3. GMI-1359 Potentiates the Cytotoxicity of DTX in Drug Sensitive Cells and Sensitizes to DTX in Drug-Resistant Cells In Vitro: Comparison with CTCE-9908 and GMI1272

Next, we evaluated if the increased expression of CXCR4 and reduced DTX sensitivity may support the use of a combinatory treatment between a dual CXCR4/E-selectin antagonist and DTX. So, GMI-1359, CTCE-9908 and GMI-1271 were co-administered at respective IC20 values with different doses of DTX in manner to calculate the IC50 values for this chemotherapeutic agent both in DTX sensitive and resistant cells. Here, we demonstrated that these combinatory strategies may reduce IC50 values for DTX both sensitive or resistant cells. As shown in panels C and D of Figure 3, GMI-1359 sensitizes to DTX administration (Figure 3C) and, at the same time, reverts (Figure 3D) the DTX resistance of PCa cell lines. Similarly, CTCE-9908 and DTX combination results also effective in these cell models. Nevertheless, the DTX sensitization was higher in presence of a co-treatment with GMI-1359 (4.8-fold on the CTRL) when compared to the single E-selectin (GMI-1271, 1.6 times on CTRL) or CXCR4 (CTCE-9908, 2.7 times vs. CTRL) in PC3 DTX sensitive cells with an increment of 3 and 1.78 times on GMI-1271 and CTCE-9908, respectively. We observed also that these differences were less pronounced in other PCa cell lines with values of 1.43 (C4-2B), 1.58 (DU145) and 1.65 (22rv1) times. GMI-1271 showed always lower sensitizing effects. When we have compared the effectiveness of different combinations in DTX resistant cell lines, we showed that GMI-1359 was still more active versus untreated animals in all cell models (with values of 2.47 [DU145DTXR], 2.78 [PC3DTXR], 3.00 [C4-2BDTXR] and 3.21 [22rv1DTXR] times) when compared to CTCE-9908 (with values of 1.76 [DU145DTXR], 2.31 [22rv1DTXR], 2.57 [PC3DTXR] and to 2.64 [C4-2BDTXR] times) and GMI-1271 (with values starting 1.2 up to 1.7 times vs. CTRL). The increment of sensitivity was of 1.08 [PC3DTXR], 1.14 [C4-2B], 1.39 [C4-2BDTXR and 22rv1DTXR] times versus CTCE-9908. These data indicate that GMI-1359 shows better in vitro effects as DTX sensitizing agent when compared to CTCE-9908 and comparable effects to CTCE-9908 as DTX reverting agent.

### 3.4. Compared effects of GMI-1359, CTCE-9908 and GMI-1271 Single Treatments in Locally Aggressive/Non Metastatic 22rv1 Xenografts

Next we compared the sensitization effects in vivo treating 22rv1 DTX sensitive subcutaneous xenografts with GMI1259, CTCE-9908 and GMI-1271 administered as single therapies in agreement with the protocol of treatments indicated in Figure 4A. We showed that GMI-1359 administration resulted in an approximate 30% reduction in 22rv1 tumour weight (0.725 g ± 0.071 vs. 1.069 ± 0.220- means ± SD, *p* < 0.05, Figure 4B).

CTCE-9908 administration resulted in a decrement of tumour weight by about 20% (0.845 g ± 0.090, *p* < 0.05) and GMI-1271 did not significantly impact tumour weight in this primary setting. The comparisons between GMI-1359 and CTCE-9908 shows a significant difference with *p* < 0.05 as indicated in the Appendix A, Appendix A. These data suggest that dual antagonism of CXCR4 and E-selectin afforded by GMI-1359 was greater of about 14.2% than the individual CXCR4 activity alone. Then we compared the time to progression (defined as the time necessary to have a doubling of tumour volume, TTP) on the basis of the growth curves plotted from single tumors. Control tumors showed a TTP of 9.0 ± 1.9 days (mean ± SE). This value reached 14.6 ± 2.0 days with GMI-1359 and 12.2 ± 2.2 days with CTCE-9908. The differences with untreated animals were statistically significant (*p* < 0.05) as indicated in Figure 4C and Appendix A
Appendix A. GMI-1271 showed a non-significant value of 10.4 ± 0.5 days. The comparison between TTP values shown in GMI-1359 and those shown in CTCE-9908-treated animals was not statistically significant. GMI-1359 administration show statistically significant values of TTP when compared to GMI-1271 (*p* < 0.05) whereas these values were not statistically different in the comparison between GMI-1271 and CTCE-9908. Kaplan Meyer curves were generated (Figure 4D) and hazard ratio values calculated by the evaluation of log-rank test (Figure 4E). Kaplan Meyer curves demonstrate that GMI-1359 was more effective in slowing the progression of the tumour as compared to CTCE-9908 and GMI-1271 (Figure 4D) with a HR = 2.8 (*p* < 0.05). In order to verify if GMI-1359 and CTCE-9908 are playing on CXCR4 and GMI-1359 and GMI-1271 are playing on E-selectin ligands we blotted 8 tissue extracts on nitrocellulose paper analysis by dot blot and serum from treated animals by ELISA.

We demonstrated that CXCR4 dots were similar to untreated animals when we considered extracts from GMI-1271, whereas this seems to be reduced in tumors treated with GMI-1359 or CTCE-9908, suggesting that GMI-1359 and CTCE-9908 are actually inhibiting CXCR4 since was reported that CXCR4 antagonists reduce CXCR4 expression in vivo due to internalization and degradation [60]. In Figure 4G we added a graphical representation of densitometric units collected for the single immune-spot grouped for each therapeutic arm. Regarding GMI 1271 (Figure 4), we observed that effectively GMI-1271 and GMI-1359 reduced the levels of HECA-452 when compared to control: vehicle 10.4 ± 1.8 pg/mL; GMI-1359 6.2 ± 1.3 pg/mL (*p* = 0.040) and GMI-1271 5.0 ± 0.7 (*p* = 0.016) whereas the comparison with CTCE-9908 showed similar results: 10.4 ± 1.8 (NS). No differences were observed between GMI-1271 and GMI-1359. In addition, no drug seemed to affect toxicity in vivo, as attested by evidence that mice did not significantly decrease their weight during treatment.

### 3.5. Compared Effects of GMI-1359, CTCE-9908 and GMI-1271 as DTX Chemo-Sensitizing Agents in 22rv1 Xenografts

Next we evaluated if GMI-1359, CTCE-9908 or GMI-1271 increased the in vivo efficacy of DTX (7.5 mg/kg/week, ip). We used the administration schedule shown in Figure 5A. We found that DTX showed a tumour weight reduction of about 50% (0.515 g ± 0.129, *p* < 0.05) versus untreated animals as indicated in Figure 5B and Appendix A, Appendix A.

The reduction of tumor weight after DTX administration was significantly different versus GMI-1359, CTCE-9908 and GMI-1271 treated animals (*p* < 0.05). The co-administration of DTX with GMI-1359 resulted in a tumor weight of 0.341 g ± 0.079. This value was significantly lower of DTX and GMI-1359 alone (*p* < 0.05) with an increment in the antitumor effectiveness of DTX of 1.51 times. The addition of DTX increased of about 2.13 times the efficacy of GMI-1359. The combination CTCE-9908 plus DTX showed a tumor weight of 0.485 g ± 0.035 (45% versus untreated animals). 

Tumor weight of this combination was significantly different versus CTCE9908 (*p* < 0.05) but not versus DTX alone, suggesting a reduced DTX sensitizing effect. The comparison between the combination CTCE-99087 plus DTX with the combination GMI-1359 plus DTX was to the advantage of the latter (*p* < 0.05). Tumor weight from GMI-1271 treated animals was 0.510 g ± 0.049 which was unable to increase DTX antitumor effects (approximately 6% of a not statistically significant in increment). The evaluation (Figure 5C and Appendix A
Appendix A) of TTP data confirms that the co-administration of DTX and GMI-1359 resulted in a significant increase in TTP values when compared to untreated controls (2.1 times) or single administrations (1.43 and 1.25 times versus GMI-1359 or DTX alone, respectively). The statistical analyses showed that GMI-1359 in combination with DTX increased in a statistically significant manner (*p* < 0.05) the TTP value in both comparisons with GMI-1359 and DTX single drug treatments as indicated in Appendix A. Similarly the combination between DTX and CTCE-9908 increased TTP values of 1.18 and 1.58 times when compared to DTX or CTCE-9908 alone. Nevertheless, this combination was unable to increase the effectiveness of DTX in a statistically significant manner. In addition CTCE-9908 plus DTX combination was significantly less effective of the combination of the combination GMI1359 plus DTX (*p* < 0.05) as indicated in Appendix A
Appendix A. The combination DTX plus GMI-1271 increased TTP values of DTX and CTCE-9908 alone of about 1.26 times, resulting less effective than GMI-1359 and CTCE-9908. Next we generated the Kaplan Meier curves (Figure 5D–F and Appendix A
Appendix A). This statistical analysis allowed us to exclude differences in the tumor engraftment. We revealed that the power (hazard ratio) by which combination treatments reduced tumour progression resulted more effective for GMI-1359 in the sensitization versus DTX when compared with CTCE-9908 and GMI-1271. The comparison between the combinations CTCE-9908 plus DTX and GMI-1359 plus DTX shows a HR = 2.9 to the advantage of the GMI-1359 plus DTX combination. Statistical analyses revealed a non-significant difference between the two treatments for the logRank comparisons. Although data of tumor weight (Appendix A) and TTP (Appendix A) indicate a statistically better efficacy of GMI1359 versus CTCE-9908 in the increased effects of DTX in the 22rv1 xenograft (*p* = 0.0187 and *p* = 0.0377, respectively), data obtained from Kaplan-Meier analyses (Appendix A) indicates that the sensitizing effects were similar. Considering that GMI1359 was administered for 14 days while CTCE-9908 for 28 days, the effects of GMI1359 could be better with a different dose administration. These considerations should be verified with another focused future experiments. The comparisons with the combination “GMI-1271 plus DTX” was statistically significant with HR = 4.8 for CTCE-9908 and HR = 7.9 for GMI-1359. The combination index was calculated for these combination. In the case of GMI-1271 we have a CI = 1.28. CI values of 0.91 and 0.99 are found for the combination of DTX with GMI-1359 and CTCE-9908, respectively. However, neither GMI-1359 nor GMI-1271 administration (alone or in combination with DTX) seemed to affect toxicity in vivo, as attested by evidence that mice not significantly decrease their weight during treatment (data not shown).

### 3.6. Intraventricular Tumour Cell Injection: Reduction of Bone Marrow Colonization (Anti-Bone Metastatic Activities) from GMI-1359, GMI-1271 and CTCE-9908

Next we verified if GMI-1359, GMI-1271 or CTCE-9908 influenced bone colonization of PCa cells. For this reason we used the bone metastatic PC3 cell derivatives, PCb2 cells, which were inoculated by an intra-ventricular (IV) route according to our previous reports [39]. X-ray determinations were performed weekly. We know that the overall rate of bone colonization by using this cell line results to be up to 75% [39]. The colonization of tibiae (80%), femurs (20%) or both (75%) was maximal whereas the localization in the other sites (anterior legs, vertebrae, skull or mandible) ranged between 2 and 5% (Appendix A). The overall percentage of bone metastases was 85%. In Appendix A we show that the PC3b2 cells colonize the bone marrow of tibiae or femurs. The staining with an epithelial markers expressed from PC3 and its cell derivatives as cytokeratin 18, CXCR4 and E selectin ligands allow us to detect epithelial tumor cells growing as dispersed or organized in sheets of cytokeratin (K18), CXCR4 and E-selectin positive cells (Appendix A). E-selectin expression was present both in the bone marrow and tumor cells. The expression of CXCR4 and the HECA-452 immunoreactivity were also evaluated in human tissues derived from bone metastases. CXCR4 expression was increased in bone metastases in agreement with our previous report [15].

Next we analyzed the effects of GMI-1359, CTCE-9908 and GMI-1271 on tumor engraftment into the bones. Treatments were performed 2 days after cell injection. In order to compare single treatments we focalized our attention on posterior legs. Nevertheless, we kept looking to note secondary metastasis sites which we recorded if they were observed, so the analyses illustrated in Figure 6 were performed keeping in mind all metastatic sites. We observed that GMI-1359 reduces the incidence of bone lesions and increases overall survival of mice with bone metastases This experimental model mimics the clinical condition of patients without clinical evidence of bone lesions, but at a high risk of bone metastasis. In Figure 6A we show the diagram of treatments used here. Radiographs of the mice showed that all animals formed osteolytic lesions characterized by obliteration of the partial proximal tibia as well as in the femurs. In Figure 6B we show representative radiological images of tumour lesions as they appear for controls and GMI-1359 treated animals at Faxitron on day 32, 40 and 46 from the start of treatment. In Figure 6C,D we show the time to bone metastases appearance (equivalent of the disease-free survival, DFS) in the different groups. In particular we found that untreated animals showed a DFS value of 30.7 ± 2.3 days. This was increased in CTCE-9908 treated animals at 44.3 ± 6.0 days. GMI-1359 administration increased DFS at 50.2 ± 5.6 days. A maximal increase of DFS was observed in GMI-1271 treated animals with 54.2 ±5.3 days. Statistical evaluations indicate that all treatments were able to reduce tumor engraftments in a statistically significant manner (*p* < 0.05). GMI-1271 showed a better effectiveness profile in this experimental aspect with statistical significance versus CTCE-9908 (*p* < 0.05) but not versus GMI-1359. The DSF value of the latter was significantly higher of CTCE-9908 (*p* < 0,05). Next we analyzed the Kaplan Meier curves generated for DFS values as shown in Figure 6E, F. Theses curves show that all treatments improved the DFS of mice with bone lesions. In Figure 6E we noticed that PCb2 cells produce X-ray positive bone lesions starting from day 25 in which 5/10 (50%) injected mice were affected bone metastases progressively increased to 80% (8/10) on day 32, to reach 100% (10/10) on day 46. Only 1/10 animals showed also little lesions in the lumbar-sacral vertebrae and front legs 70 days post-injection. In the contrast, we found radiographic evidence of lytic bone lesions in 3/10 (30%) mice treated with CTCE-9908 on day 25. Bone metastases reached 40% (4/10) on day 32, 60% (6/10) on day 40 with a final incidence of 70% (7/10) on day 46. This value was maintained until the end of experiments at day 70. We observed also that in GMI-1271–treated mice, only 1/10 (10%) showed X-Rays positive on day 32. Bone metastases reached 60% (6/10) on day 40 and remain at this percentage until day 70.The radiographic evidence of osteolytic bone lesions in GMI-1359 treated animals was 3/10 (30%) at day 32, reached 50% (5/10) at day 40, 60% (6/10) at day 46 and 70% (7/10) at day 53 without other animals with skeletal involvement until day 70. The analyses of hazard ratios (Figure 6F) revealed that although all treatments were statistically significant versus untreated animals, no statistically significant differences were observed in the comparisons between the single treatments. However a trend of better anti-metastatic activity from GMI-1359 when compared to the CTCE-9908 was observed also in these evaluations.

Next, we analyzed the bone lysis (Figure 6G) quantified at image J software as Densitometric Lytic Units (DLU). We observed that, in agreement with DFS analyses, DLUs were significantly higher in controls (CTRL) with respect to GMI-1359; CTCE-9908 or GMI-1271 treated animals at each days of analysis (32, 40 and 46 days). First we analyzed the doubling time as the time required for a quantity to double in size or value and successively the growth rate. The equation which characterized this parameter is dN/dt = rN where dN/dt is the population growth rate, “r” is the proportionality constant and N is the population size. This was considered accordingly Hirsch and Engelberg [61]. In this case the untreated animals show a GR = 0.71. The growth rates were 0.50, 0.35 and 0.42 in CTCE-9908, GMI-1359 and GMI-1271, respectively, so the rate of bone lysis is higher in CTCE-9908 when compared to GMI-1359 with *p* < 0,05. In addition, we found that CTX-I and mTRAP are significantly higher in the untreated animals when compared to treated animals. The comparison for CTX-I levels indicated that GMI-1359 was statistically more active with respect to CTCE-9908 and GMI-1271 (* *p* < 0.05). The levels of mTRAP were significantly lower in GMI-1359 when compared to CTCE-9908, whereas they were not statistically different in the comparison between GMI-1359 and GMI-1271. It is necessary to note that the CTX-I levels released in the serum of treated or not animals, measuring the effect of treatments on the osteoclasts activity, whereas those observed for mTRAP, being this an osteoclast marker, measure the amount of osteoclasts.

Post-mortem necroscopy documented a lower significant incidence of visceral metastases in GMI-1359- CTCE-9908- and GMI-1271-treated mice (Figure 6J) with respect to controls, although the differences observed between treatments were not significant. In Figure 6K we show the rate of mortality of untreated (CTRL) or treated animals with bone and visceral metastases. Commonly, mice euthanasia is necessary at distress signs evidence and came starting from 40–60 days (mean ± SD, 52.0 ± 1.6 days) in the untreated animals. Overall survival values grew until 58.7 ± 2.5 days in CTCE-9908 treated animals (*p* = 0.0002 vs. CTRL). GMI-1271 treated animals were euthanized after 59.9 ± 2.8 days (*p* = 0.0134 vs. CTRL, *p* = 0.1744, not significant, NS, vs. CTCE-9908) whereas GMI-1359 treated animals were euthanized after 62.9 ± 2.8 days (*p* = 0.0002 vs. CTRL, *p* = 0.0033 vs. CTCE-9908 and *p* = 0.005 vs. GMI-1271). In Figure 6L we show the Kaplan-Meier curves with relative hazard ratio values and statistics. Although the values of HR vs. controls are higher in the GMI-1359 treated animals (HR = 4.10) when compared to CTCE-9908 (HR = 2.29) the comparison between these two treatments reached a non-statistically significant value (HR = 1.78), so we can only state that there is a trend of better anti-metastatic activity from GMI-1359 when compared to the CTCE-9908.

### 3.7. GMI-1359 Affects Intra-Osseous Tumour Growth and Increases the Efficacy of Docetaxel

In order to directly evaluate the specific anti-tumour effects in bone microenvironment the intratibial tumour model was used. This model allowed us to inject a substantially higher number of tumour cells within the bone marrow compartment relative to the intracardiac tumour model. We previously demonstrated that approximately 3 days after intratibial tumour cell injection, tumour-bearing mice developed PC3M tumour foci [15,35,41]. Therefore, we decided to start treatments at this time. Treatments were stopped 28 days after drug administration. On day 15, 86.7% of tibiae developed radiographic evidence of bone lesions. Skeletal lesions were graded as described by Yang et al. [62]. In Figure 7A, we show the therapeutic schedule used here. In Figure 7B we show the representative radiological appearance of bone lesions as they appear to Faxitron analysis for the identification of osteolytic scores together to bioluminescence images. X rays and bioluminescence activity were evaluated at 7, 14 and 21 days. We used different modalities of analyses for the BLI values plotted for the time. In the first analysis (Figure 7C–E) we considered the increment of BLI (photons/sec) over time only in BLI positive tibiae in manner to exclude the negative tibiae having BLI values close to zero.

On the basis of these data, the single agent curves showed very similar BLI values for GMI-1359 and CTCE-9908 at the 21-day time point whereas GM1271-treated group appears to have the lowest BLI value at day 14, so we cannot discriminate and compare the effects of treatments. In this analysis modality, indeed, we do not consider if treatments influence the phase of consolidation or bone engraftment in the metastatic site (latency). In addition, during the time of experiment new tibiae resulted positive to bioluminescence at different controls (times). For this reason, we performed a new analysis considering the BLI values for all determinations. This new procedure reduced the mean values of BLI and increased the Standard errors (SE) of BLI distribution. Table 1 summarizes the results of these analyses. In this table we compared also the changes in Densitometric Lytic Activity measured at the same times considering, also in this case, the negative tibiae which received a score of zero. If we consider Table 1 data, the statement that GMI1351 is more active of CTCE-9908 and GMI-1271 reaches a higher meaning when we analyze of tumor growth rates. Here dN/dT indicates the changes in the amount of BLI respect to the T0 values for time unit. The growth rates calculated for the controls were 57.9 for the time interval T7-T21 days with a value of 8.7 for the initial phase (consolidation phase T7-T14 days) and 107.1 for the second phase (exponential proliferation phase). This suggests that in the untreated animals (CTRL) the exponential phase is dominant on the consolidation phase since the tumor cells are not subjected to any drug-mediated selective pressure. The DTX administration reduced of about 3.68 times the growth rate with a value of 15.7 (T4-T21 time interval) in which the consolidation phase shows a growth rate of 2.8 (4.35 times vs. CTRL) and a proliferation phase showing a growth rate of 28.6 (3.74 times). This indicates that DTX interferes both with the engraftment and the growth of tumor cells in the bone. Similarly, GMI-1359 treatment goes from a growth rate value of 7.8 (T7-T14 interval; 12% less respect to CTRL and 3.9 times higher respect to DTX), to 21.4 (T14-T21 time interval; 5.1 times lower of CTRL and 38% less of DTX) with an overall value of 14.3 (4.05 times lower of CTRL and 14% less to DTX), so GMI-1359 showed low antitumor activity when compared to DTX reducing, however, both the engraftment and the tumor growth in the bone. Next we considered the combination GMI-1359 plus DTX and we found that the calculated growth rate was 4.6 for the all considered phases. This support the concept that GMI-1359 increased the effectiveness of DTX reducing mainly the exponential growth of 4.65 times when compared to GMI-1359 alone and 6 times when compared to DTX alone. In CTCE-9908-treated animals we found that the growth rates go from a value of 10.4 (tumor engraftment phase) to 22.6 (exponential phase) with an overall value of 16.5. These data suggest that CTCE-9908 showed lower antitumor effectiveness when compared both to DTX and GMI-1359 being higher the growth rate both in the initial and exponential phases when compared to DTX and GMI-1359 single therapy. CTCE-9908 ameliorated the DTX effectiveness with an overall growth rate of 6.2 (4,1 and 8,2 for the initial and exponential phases, respectively). While the effects on engraftment phase was similar between two combinations the exponential phase was higher in the combination CTCE-9908 plus DTX when compared to GMI1359 plus DTX (1,78 times). This means that the later shows better effects on DTX sensitization. The comparison between the exponential phases of the combination CTCE-9908 plus DTX show also increased activity of about 3.5 time versus DTX and 2.75 times versus CTCE-9908 single treatments alone. 

GMI-1271 administration determined an overall growth rate of 25.5 (14.3 and 37.1 for the initial and exponential phases, respectively) which is lower respect to control, but higher with respect to those observed for single DTX or GMI1359 administrations. The analyses on the combination GMI-1271 plus DTX showed an overall growth rate of 6.7, which was maintained for all phases of the tumor growth, so GMI-1271 shows, also, DTX sensitizing effects that, however, are lower of combinations CTCE-9908 plus DTX and GMI-1359 plus DTX. We demonstrated also that the osteolysis were significantly reduced after co-treatment of GMI-1359, CTCE-9908 and GMI-1271 with DTX and that GMI-1359 effects were more marked when compared to GMI-1271 and very similar when compared to those observed for CTCE-9908. In Figure 7F we show representative images for tibiae with bone metastasis in GMI-1359, DTX and their combination. This was associated with changes in osteolytic activity (Table 1) and mTRAP5b activity (Figure 7G). A statistical significance was observed only for GMI-1359 plus DTX and CTCE-9908 plus DTX vs. controls (*p* < 0.05) Although we have previously demonstrated that mice with skeletal metastases show deterioration in the quality of life with asthenia and cachecsia, mice treated with these compounds demonstrated delayed cachecsia evidence measured as body weight wasting (Figure 7H). However statistical analyses revealed that only GMI-1359 plus DTX resulted significantly different to those observed for control animals (*p* < 0.05) whereas the comparisons with the effects of CTCE-9908 plus DTX and GMI1271 plus DTX were not statistically different versus the combination with GMI-1359 and DTX. This could be due to reduced number of animals for each harms.

### 3.8. Effects of CXCR4 and E-Selectin Antagonism on Osteoblast and Osteoclast Growth and Function In Vitro

During the dissemination to bone, metastatic cancer cells locate in a putative ‘metastatic niche’, a particular microenvironment that regulates the colonisation itself, the maintenance of tumour cell dormancy and the subsequent tumour regrowth [6,63]. It was widely demonstrated that bone metastatic cancer cells displayed a characteristic pattern of homing in the long bones, with the majority of tumor cells seeded in the trabecular regions. In addition, it was showed that the mobilization of Human Stromal Cells (HSCs) from the niche resulted in increased numbers of tumour cells disseminated in trabecular regions [64]. It has been also demonstrated that the administration of the CXCR4 antagonist AMD3100 seems to increase the circulating pool of mesenchymal stromal cells (MSCs) improving the fracture healing [65] In addition it was also demonstrated that CXCR4 sustain osteogenic differentiation [66]. Changes in the in vitro osteogenic and osteolytic potential of PCa cells were verified in presence of GMI-1359, CTCE-9908 and GMI-1271 by using osteoblasts purified from mouse calvaria or murine 3T3-MC1 cells grown in the presence of osteogenic medium (CTRL +) or conditioned media from six PCa cell lines (PC3, 22rv1, DU145, C4-2B, VCap and LnCaP cells).

After treatments and at the times indicated in material and methods, the cultures were analyzed for alkaline phosphatase (ALP), stained by using Alizarin Red or treated with β-glycerophosphate and citric acid for the Von Kossa reaction. As first analyses we verified the osteogenic potential of bone trophic or not PCa cells. In Appendix A
Appendix A we show representative images for Crystal Violet staining of osteoblasts derived from mice calvaria and 3T3C1 cells. Crystal Violet was solubilized and quantified as optical density (OD) measured at 595 nm. In Appendix A we show that the cell proliferation of both osteoblast populations (mouse (3T3-MC1 and osteoblasts from calvaria) was increased in presence of CMs collected from PC3, 22rv1 and C4–2B cell lines. The addition of 0.5 µM of GMI-1359 or CTCE-9908 reduced the growth of these cells in a significant manner (*p* < 0.05) for all three prostate cancer cell models, whereas GMI-1271 influenced significantly the growth of the osteoblastic C4–2B cells. Next we analyzed changes in the osteoblast function (Figure 8A–C and Appendix A). We observed that the more osteogenic cells induced most osteogenic activity (ALP, Von Kossa and Alazarin Red stains). Data on ALP activity (which is an index for osteoblast proliferation, Appendix A) suggests that C4–2B is the more osteogenic cell line when compared to PC3 and the ALP activity data in panel D appear to show that PC3 CM reduced ALP activity relative to control. PC3 are less osteogenic of C4–2B.

Alazarin Red solubilization showed values of optical density at 450 nm significantly higher (*p* < 0.05) in VCaP and LnCaP osteogenic cells. In addition we show that 22rv1 cells secrete osteogenic factors that bring it closer to the C4–2B and distance the PC3 in agreement with their capacity to produce mixed osteosclerosis/osteolytic lesions. C4–2B, VCaP and LnCaP cells represent the more osteogenic cells as widely demonstrated in the literature. The administration of GMI-1359 and CTCE-9908 induced ALP activity in PC3 and 22rv1 and also it was not able to increase this enzyme in the more osteogenic C4–2B cells (Appendix A). The evaluation of Alazarin Red images and optical density confirm the increases ability of GMI-1359 and in a less measure of CTCE-9908 in sustaining osteoblast function in PC3 and 22rv1 cells but not in C4–2B having, however, an elevated basal osteogenic activity (Figure 8B,C). Next we analyzed the effects of conditioned media of PCa cells, with or without GMI-1359, CTCE-9908 and GMI-1271, on osteoclast generation and differentiation (Figure 9). Raw275.6 cells were used as osteoclast precursor cell model. The addition of PC3 CM was able to increase significantly the proliferation of these cells (Figure 9A) whereas GMI-1359 and CTCE-9908 reduced these effects (Figure 9B). Similarly, osteoclast differentiation (Figure 9C) and function (Tartrate resistant acid phosphatase, Trap, Figure 9D) were induced by PC3 cells and reduced by CTCE-9908 and GMI-1359. These effects were lower in presence of C4–2B or 22rv1 cells.

## 4. Discussion

Metastatic prostate cancer patients present in two ways-with already disseminated disease at the time of presentation or with disease recurrence after definitive local therapy [4]. Androgen deprivation therapy is given as the most effective initial treatment to patients. However, after the initial response, almost all patients will eventually progress despite the low levels of testosterone. Disease at this stage is named castration resistant prostate cancer (CRPC). Before 2010, the taxane docetaxel (DTX) was the first and only life prolonging agent for metastatic CRPC (mCRPC). Resistance to DTX, however is a common event leading to a not curable disease. Aberrant expression of key gene products, such as CXCR4, has been considered to facilitate cell adhesion and invasion, angiogenesis and bone metastasis [15,16,17,18,19,20,21,22,23]. CXCR4 antagonists have been used in preclinical models of PCa including plerixafor [14] and CTCE-9908 [14,20]. In addition, it has been demonstrated that E-selectin ligand-1 controls circulating prostate cancer cell rolling/adhesion and metastasis [9,10,11]. Circulating PCa cells, indeed, preferentially roll and adhere on bone marrow vascular endothelial cells, where abundant E-selectin and stromal cell-derived factor 1 (SDF-1) are expressed, initiating a cascade of activation events that eventually lead to the development of metastases. CXCR4 is mainly expressed in the tumor cells and its ligand in bone marrow endothelial (BME) cells. Differently E selectin is expressed in BME cells and its ligands is expressed in tumor cells. So antagonists of these ways may modulate bone colonization and reduce tumor growth into the bone. Stromal cells surrounding primary tumours or metastatic lesions may secrete high levels of CXCL12/SDF1 and induce CXCR4 expression in tumour cells supporting the hypothesis that CXCR4 is important for tumour growth [67].

Therefore we verified in vitro that CAFs and bone derived cells amplified the expression of CXCR4. The induction of CXCR4 in bone tropic cells (PC3) was higher in the presence of conditioned media harvested from bone derived cells supporting a role of CXCR4 in bone engraftment of these cells. Lower effects were expressed in 22rv1 cells in agreement with their lower bone tropism. Similarly, the expression of E-selectin ligand recognized by HECA-452 antibody was significantly increased in PC3 cells after administration of CM from bone-derived cell cultures suggesting a possible role of this system in bone-PCa cell cross-talk. Although DTX administration is not able to modify the levels of immune-reactivity for HECA-452, DTX resistant cells as PC3DTXR showed higher HECA452 levels, suggesting that the system E-sectin/HECA452 could be important in osteolytic bone metastases. It has been also demonstrated that CXCR4 imparts docetaxel resistance upon CXCL12 stimulation in PCa cells which over-express this receptor [68]. The activation of CXCL12/CXCR4 signalling, indeed, was correlate with increased DTX-induced G2/M cell cycle arrest, DTX-induced microtubules stabilization and DTX resistance by LIMK1 activation [69]. We observed that DTX was also able to increase the expression of CXCR4 and CXCR4 expression was higher also in DTX resistant PC3, DU145 and 22rv1 when compared with drug sensitive parental cells. This was supportive for a role of CXCR4 in DTX sensitivity and CXCR4 antagonism offers a significant opportunity to treat lesions with elevated CXCR4 expression and DTX resistance. So, we analyzed the effects of dual E-selectin/CXCR antagonist GMI-1359 on the DTX effectiveness by using both subcutaneous xenograft and the intra-tibial model. Although it was previously reported that the sensitivity to DTX was increased in prostate cancer after administration of the CXCR4 antagonist, AMD3100, our study demonstrates for the first time that the CXCR4 inhibition increases antitumor effects of DTX alone in bone microenvironment. Conditioned media derived from osteoblastic cell cultures increased CXCR4 expression in a more marked extent respect to supernatants derived from bone marrow stromal cells or osteoclast like cells (RAW276.5) suggesting that the osteosclerotic lesions could store much more elevated level of CXCR4. In the most advanced metastatic stages, osteolysis take place to osteosclerosis on the basis of changes in the equilibrium between osteosclerotic and osteolytic factors, so osteolysis represents the more clinically relevant skeletal event which clinically predominates in the most advanced stages leading to the main problems (bone fractures) and that considerably impact on the quality of life of the patient. Bone lesions from prostate cancer are, however, mixed with osteosclerotic and osteolytic areas that mix in the same tissue sample. Although the 22rv1 cell line is the sole model able to induce this mixed osteosclerotic/osteolytic lesion when injected directly into the tibia [36,70], resulting hypothetically a model more close to the human situation, these cells show very low/absent ability to colonize the bone after intracardiac injection as well as reduced bone engraftment. In addition the elevated local 22rv1 tumor growth associated to the ability to sustain angiogenesis and mediate inflammation through the secretion of CXCR4 ligands, SDF1a and MCP1, makes 22rv1 cells a good model for studying the capacities of CXCR4 inhibitors in a non-metastatic primary tumor model as may be the subcutaneous xenograft model.

Here, we show that blockade of CXCR4 increased the sensitivity to DTX in drug sensitive cells and sensitizes to DTX in drug-resistant cells in vitro. Data shown in Figure 3 suggest, indeed, that the dual CXCR4/E-selectin antagonist (GMI1358) increased the efficacy and reduced the resistance versus the DTX. This effect was also observed for the CXCR4 antagonist (CTCE-9908). We observed that in the parental PC3 cells, DTX sensitization was higher in presence of a co-treatment with GMI-1359 (4.8 times vs. CTRL) when compared to CTCE-9908 (2.7 times vs. CTRL) with a 1.77 times iincrement. The differences observed for the other DTX sensitive cell lines were lower. The GMI-1359 increased (1.47 to 3.21 times) the sensitivity to DTX in resistant PCa cells. The increment of sensitization activity observed for CTCE-9908 showed values ranged between 1.76 and 2.64 times. The effects were lower in presence of a combination with GMI1271. The increment of sensitivity of GMI-1359 versus CTCE-9908 was ranged between the 8% to 40%. These data indicate that GMI-1359 shows better in vitro effects as DTX sensitizing agent when compared to CTCE-9908 and comparable effects to CTCE-9908 as DTX reverting agent. In vivo we observed that, as expected, the reduction of tumor weight after GMI1359 and CTCE9908 administration resulted lower of the DTX. The co-administration of DTX with GMI-1359 resulted in a significant reduction of tumor weight when we compared to DTX and GMI-1359 alone. Tumor weight in the combination CTCE-9908 plus DTX was significantly different to CTCE9908 but not to DTX alone, suggesting a reduced DTX sensitizing effect. GMI-1271 was unable to increase DTX antitumor effects. TTP data confirms that the co-administration of DTX and GMI-1359 resulted in a significant increase in TTP values when compared to single administrations. Conversely the combination between CTCE-9908 and DTX was unable to increase the effectiveness of DTX in a statistically significant manner. This combination was significantly less effective of the combination of the combination. GMI1271 confirmed lower DTX sensitizing effects on the basis of TTP changes. Analyzing Kaplan Meier curves we can state that the power (hazard ratio) by which combination treatments reduced tumour progression resulted more effective for GMI-1359 in the sensitization versus DTX when compared to CTCE-9908 and GMI-1271. In particular, the comparison between the combination CTCE-99087 plus DTX with the combination GMI-1359 plus DTX was to the advantage of the latter as indicated in Figure 5D by the HR=2.9. The hazard ratio value of this comparison was similar to those calculated for the comparison between GMI-1359 and CTCE-9908 single therapies (Figure 4E). However in the first case the hazard ratio resulted a statistically significant logRank analysis. This means that we did not conclude that GMI1359 was more active of CTCE-9908 in the DTX sensitization in vivo. We can only state that on the basis of sc xenograft results, there is an upward trend in DTX sensitization in GMI-1359 when compared to CTCE-9908. This could be due to the lack of a sufficient number of replicates (i.e., animals). In fact, assuming that the high HR value is maintained, it could have a significant statistical value if the number of replicates was sufficient to obtain a statistical significance. However, the data shown here for sc xenograft model indicates that there is an upward trend in DTX sensitization in GMI-1359 when compared to CTCE-9908.

Next we tried to verify the question of whether the sensitizing effect of the dual inhibitor could be attributed to its CXCR4-targeted activity and not the E-selectin-targeted activity. We can state only that the inhibitory ability against E-selectin played by GMI-1359 could ameliorate the efficacy of the single CXCR4 antagonism of this molecule. Another problem is the better activity of the dual inhibitor GMI-1359 when compared to single CXCR4 inhibitor, CTCE-9908, is a fair comparison with respect to CXCR4 inhibitory activity. Although comparative analyses are not available in the literature for the in vivo/in vitro activities of CTCE-9908 and GMI-1359 versus CXCR4 and to be sure of this, we tested CXCR4 activity by using a Phospho-CXCR4 (Ser339) colorimetric cell-based ELISA Kit. We observed that the effects of GMI-1359 and CTCE-9908 on CXCR4 phosphorylation/activity were similar at the commonly used concentrations (data not shown). In addition, to demonstrate that this was manifest also in vivo, we can observed the results shown in Figure 4F on the levels of CXCR4 after treatment with CTCE-9908 or GMI-1359 after dot blots analysis. Here we show that at commonly used pharmacological doses, the levels of CXCR4 were not significantly different between the two groups (Figure 4G) and showed similar decrements respect to those observed for the control (untreated animals). This result indicates that since the reduction/disappearance of CXCR4 is a marker for the effective CXCR4 inhibition [48], CTCE-9908 and GMI-1359 showed similar inhibitory effects on this event at the used dosage, so the greater activity observed with the dual inhibitor cannot be merely a result of a higher level of CXCR4 inhibition vs. the single target inhibitor. We demonstrated also that the in vivo administration of GMI-1359 and CTCE-9908 but not GMI-1271, reduced significantly the levels of CXCR4 in treated tumors, suggesting, inter alia, that GMI-1359 and CTCE-9908 are working on CXCR4. This is in agreement with a previous study demonstrating that plerixafor was able to reduce CXCR4 in target cancer cells [15]. Regarding GMI-1271 we demonstrated that GMI-1271 and GMI-1359 reduced the serum expression of E-selectin as demonstrated by the assay of E-selectin levels in the serum of patients enrolled in clinical trials with GMI-1271 indicating that GMI-1271 is acting on its target [71]. We observed also that GMI-1359 reduced the incidence of bone lesions and increase Overall Survival (OS) in mice injected by intracardiac model, an experimental model mimicking the clinical condition of patients without clinical evidence of bone lesions, but at a high risk of bone metastasis. Radiographs of the mice showed that all animals formed osteolytic lesions characterized by obliteration of the partial proximal tibia and femurs whereas rare secondary bone lesions were observed. GMI-1359 reduced significantly the incidence as well as delayed the time of appearance of bone lesions. CTCE-9908 and GMI-1271 showed lower effects of GMI-1359 supporting once again the hypothesis that the effectiveness of dual E-selectin and CXCR4 antagonism is better of single antagonists. Another question may be if other off-target effects may be impacting the GMI-1359-mediated growth inhibition in the presence of DTX. The fact that the dual inhibitor shows a greater impact on osteolysis could be linked to experimental evidence demonstrating that the SDF-1α augments E-selectin mediated endothelial cell (EC) adhesion and migration in a CXCR4-dependent manner [72]. E-selectin expression was found to be elevated in a wide cohort of endothelial cell and precursors [73]. Angiogenesis, however, is altered in the bone marrow as a result of osteolysis, tumor growth and changes in vascular network. CXCR4 and E-selectin antagonist may modify the composition of osteolytic behavior and this may be impacting bone tumor growth in the presence or absence of DTX and to be responsible for apparently “off-target effects”. The E-selectin-CXCR4 axis is considered an important pathway in the process of osteolysis [74,75], so the dual CXCR4/E-selectin inhibition produces maximal effects. The evaluation of bone lysis quantified as Lytic Units (LU), serum levels of CTX-I and TRAP-5b was significantly higher in controls (CTRL) with respect to GMI-1359; CTCE-9908 or GMI-1271 treated animals at each days of analysis suggesting that treatments can modify also tumour growth in bone microenvironment. Kaplan-Meier curves showed that treatments improved overall survival (OS) of mice with bone lesions. The number of mice which survived after treatment with GMI-1359, CTCE-9908 or GMI-1271 was indeed significantly higher (*p* < 0.05) with respect to controls. Kaplan Meyer analyses showed that GMI-1359 administration was statistically more effective with respect to GMI-1271 administration. However, once again, no statistically significant difference was found between GMI-1359- and CTCE-9908 or GMI-1271- and CTCE-9908 treated animals with respect to OS. Post-mortem necroscopy documented also a lower and significant incidence of visceral metastases with respect to controls, although the differences observed between treatments were not significant. In bone microenvironment, the power of GMI-1271 in the reduction of bone metastases was more significant respect to those observed in vitro and in the 22rv1 xenografts. This is in agreement with some reports showing that following intracardiac injection of FT6 transfected cells, bone colonization was robust respect to those observed in presence of parental cells [75]. In these experiments a FT6 inhibition by a fucose mimetic reduced significantly bone metastases. Similarly, it has been demonstrated that selectin ligand sialyl-Lewis antigen drives metastases in hormone dependent breast cancer [12]. A further question may be if the inhibitors impact cell dissemination into the bone. We do not have enough data to say that GMI-1359, GMI-1271 or CTCE-9908 have modified bone colonization/bone metastatization. The main motivation lies in the fact that the intracardiac cell injection model, which could serve this purpose, was not use here. We should have administered the various compounds before and after intra-cardiac cell inoculation. In this way the bone osteolytic micro-lesions, in the process of formation, could be hampered by the drugs. However a significant reduction of bone engraftment was observed for all experimental compounds.

In order to evaluate the anti-tumour effects in bone microenvironment bypassing the extravasation phenomenon, the intratibial tumour model was used. This model allowed us to inject a substantial higher number of tumour cells within bone marrow with respect to intracardiac tumour model. In addition, in order to evaluate possible effects of the compounds in analysis on intra-bone tumor growth, we injected into the tibiae luciferase transfected PC3 cells (PC3luc). Luciferase activity (BLI) was analyzed as indicated in the Material and Methods section. Effectively, we demonstrated that GMI-1359 affected intra-osseous tumour growth and increases the efficacy of docetaxel. Osteolysis and tumor growth were, indeed, significantly reduced after co-treatment of GMI-1359, CTCE-9908 and GMI-1271 with DTX. In addition these anti-tumour effects were stronger when compared to subcutaneous 22rv1 tumour suggesting a major role of CXCR4 in the bone microenvironment dependent tumour growth and drug resistance respect to a primary lesion. GMI-1359-mediated DTX sensitizing effects were more marked when compared to those observed for GMI-1271 and very similar, instead, when compared to those observed for CTCE-9908. Our report provides the rationale to use GMI-1359 in adjuvation to taxane-based chemotherapy in men with high-risk prostate cancer. Although the combination between GMI-1359 and DTX seems to be, indeed, low when compared to the combination between CTCE-9908 and DTX, we can be satisfied by a similar increment of activity. A low increase of effectiveness is welcome for the mCRPC treatments. It is necessary to consider, indeed, that the mCRPC is a lethal and almost incurable disease and the pharmacological therapies (ie. use of bisphosphonates) are aimed at improving the quality of life of metastatic patients through the reduction of osteolysis and bone fractures. In addition, we must also take into account that the use of targeted clinical trials is necessary to evaluate the feasibility in humans of the use of GMI-1359 and DTX combinations. Among other things, the inhibition of CXCR4 and E-selectin signaling pathways also modulates immune responses by regulating the recruitment of inflammatory cells in the tumor site and in bone metastases. To get answers on the clinical use of GMI-1359, it would also be necessary to use non-immuno-compromised animal models to evaluate efficacy in a native immune system.

## Figures and Tables

**Figure 1 cells-09-00032-f001:**
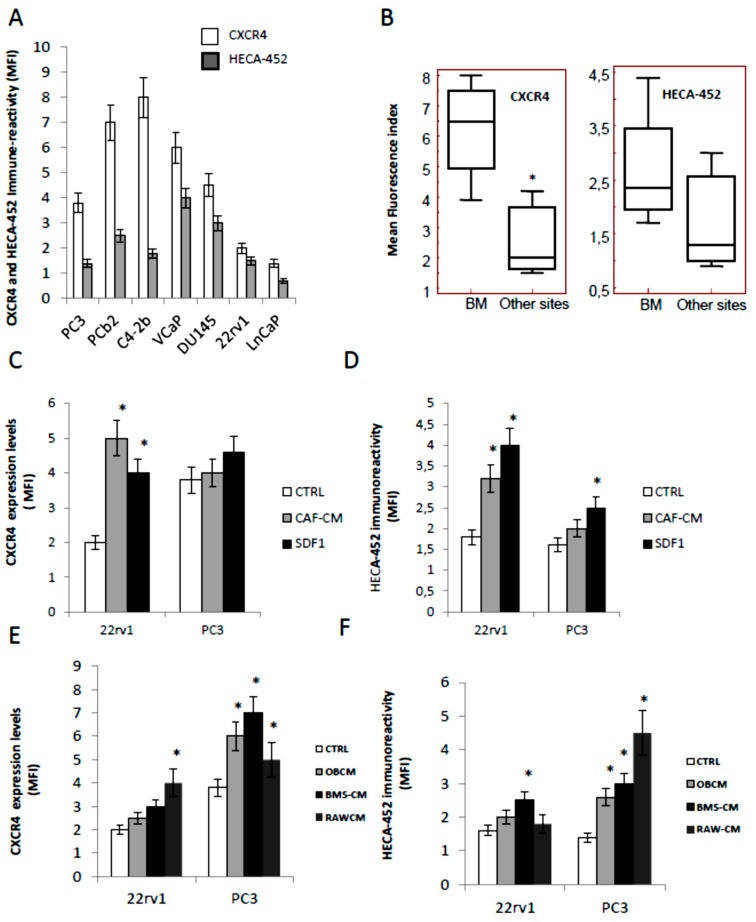
Immune-reactivity (IR) of CXCR4 and HECA-452 in prostate cancer cells. (**A**) Antigen quantification for both antibodies in seven prostate cancer cells (Mean Fluorescence Index, MFI ± Standard Deviation, SD from three separate analyses). (**B**) MFI values were grouped for bone metastatic and non-bone metastatic PCa cells. Box plots show median values of MFI and 95% of confidence. * *p* < 0.05 in the comparison between bone versus non bone metastatic sites. (**C,D**) Effects of CAF-CM (1:1 in complete medium) and exogenous (10 ng/mL) SDF1α on CXCR4 (**C**) and HECA-452 (**D**) immune-reactivity levels (MFI) in 22rv1 and PC3, used as models. (E, F) Effects of BMS-CM, Murine osteoblast-like MC3T3-E1 (OB) and RAW-CM cells on CXCR4 (**E**) and HECA-452 (**F**) levels by FACS assays in PC3 and 22rv1 cell models. Data represent the values of MFI calculated for each cell line as indicated in MM ± the values of standard deviation calculated from individual three FACS analyses. * *p* < 0.05 versus controls.

**Figure 2 cells-09-00032-f002:**
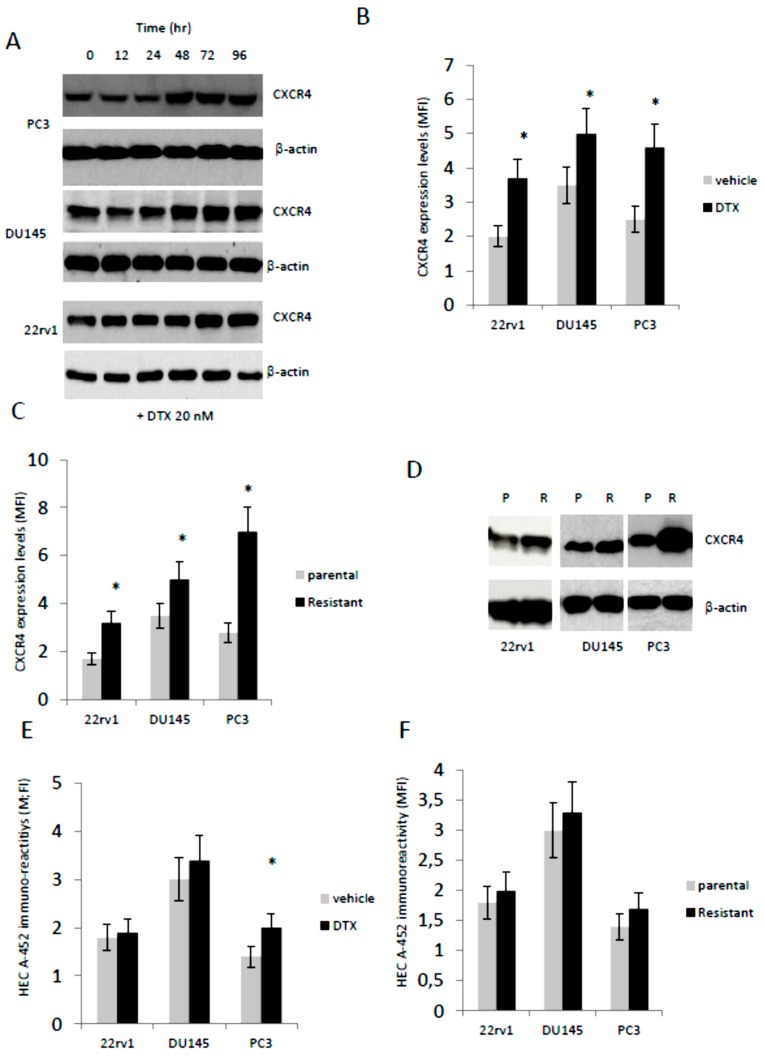
CXCR4 expression and HECA-452 immune-reactivity after treatment with non-toxic DTX doses. (**A**) Western blotting analysis performed on PC3, DU145 and 22rv1 cells cultured with 20 nM DTX (time course experiment). (**B**) Fluorescence analyses on 22rv1, DU145 and PC3 cells cultured with DTX for 96 hr. (**C**) CXCR4 expression levels by FACS (fluorescence index) on parental and resistant PCa cells. (**D**) Western blotting analyses performed on parental (P) and DTX resistant (R) 22rv1, DU145 and PC3 cells. (**E**) HECA-452 immuno-reactivity in parental cell treated with DTX and (**F**) DTX-resistant cells. Western blots were loaded with 40 µg/lane of proteins. Western blots images are representative o three different gels/experiments. MFI values were calculated for each cell line as indicated in MM ± standard deviation calculated from individual three FACS analyses. * *p* < 0.05 versus respective controls.

**Figure 3 cells-09-00032-f003:**
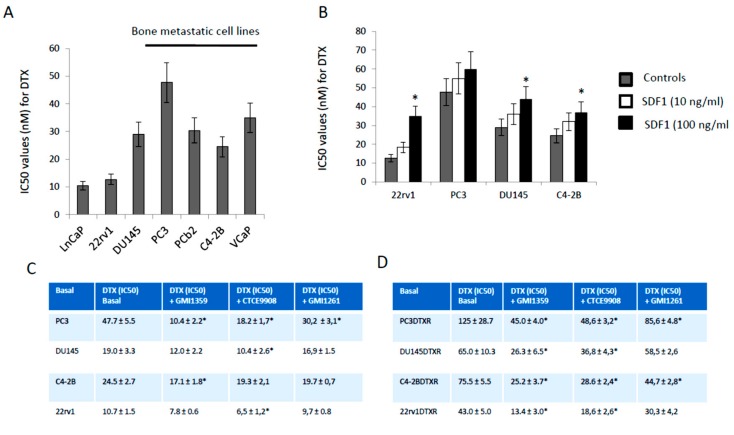
CXCR4 inhibition increases the sensitivity to DTX in vitro. (**A**) IC_50_ values for DTX calculated in cells derived from bone metastases (PC3, PC3b, C4–2B and VCaP), primary (22rv1), lymph-nodes (LnCaP) and brain (DU145). (**B**) IC_50_ values calculated for DTX in presence of 10–100 ng/mL SDF1α. (**C,D**) GMI-1359, GMI-1271 and CTCE-9908 were co-administrated with different doses of DTX (0–200 µM) and IC_50_ values for DTX calculated in DTX sensitive (**C**) and resistant (**D**) cells. Not to work at too high concentrations, which could mask the combined effects, GMI-1359, GMI-1271 and CTCE-9908 were used at doses close to their IC_20_ values, so GMI-1359 was administered at 12.5 µM (C4–2B), 8.7 µM (PC3), 15 µM (PC3DTXR) and 10.3 µM (DU145DTXR). GMI-1271 was administered at 10.7 µM (PC3 and PC3DTXR) and 6.9 µM (DU145 and DU145DTXR). Finally, CTCE-9908 was added at 25 ng/mL (PC3), 45 ng/mL (C4–2B), 15 ng/mL (DU145DTXR) and 58 ng (PC3). Data are representative of three similar experiments performed in triplicate. * *p* < 0.05.

**Figure 4 cells-09-00032-f004:**
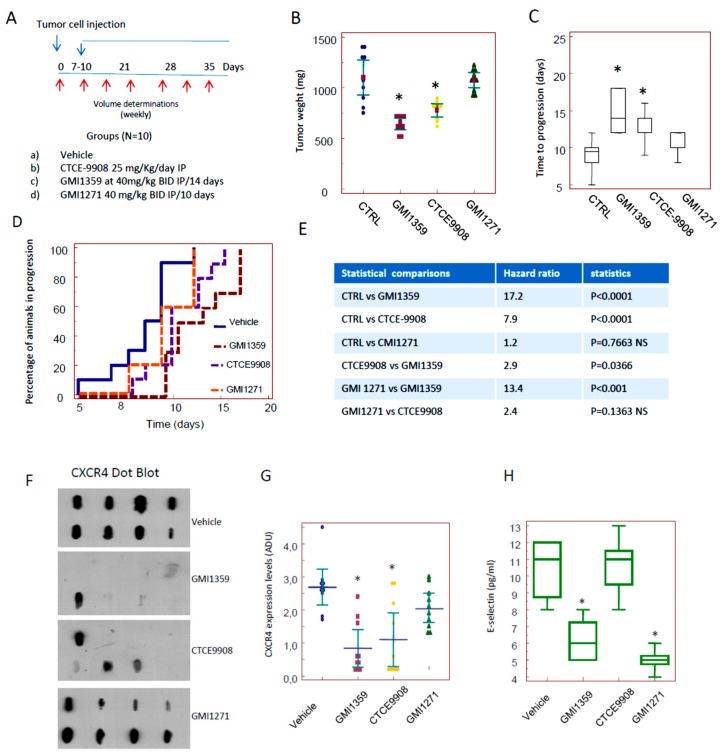
In vivo experiments: comparison of the effects assessed for GMI-1359, GMI-1271 and CTCE-9908 administered at the subcutaneous 22rv1 xenograft model. (**A**) Diagram of treatments: tumour bearing animals were randomized approximately 10 days or when tumors reached 80–100 mm^3^ of volume in four experimental groups as follows Group 1: mice receiving intraperitoneal (i.p.) injections of 100 µL PBS (vehicle); Group 2: mice receiving CTCE-9908 (25 mg/kg, i.p); Group 3: mice receiving GMI-1359 twice a day ip at 40 m for consecutive 14 days; Group 4: mice receiving GMI-1271 40 mg/Kg BID for consecutive 10 days; (**B**) 22rv1 tumour weights assessed after 35 days from cell injection and 28 from start of treatments. (**C**) Time To Progression analysis (median values and 95% CI); (**D**) Kaplan Meyer representation for the analyses of effects of GMI-1359, CTCE-9908 and GMI-1271; (**E**) hazard ratio values and relative significance for the different comparisons; (**F**) dot blots for CXCR4 performed on tissue extracts obtained from eight tumors/group. Two hundred µg of extracted proteins were spotted in a nitrocellulose paper. (**G**) CXCR4 densitometric units analyzed by Image J software as described in the Material and Methods section (H) ELISA determination for HECA-542 in the plasma of animal of control or treated withGMI-1359, CTCE-9908 and GMI-1271. Statistics for all comparisons are in the text and Appendix A. **p* < 0.05 versus respective controls (vehicle).

**Figure 5 cells-09-00032-f005:**
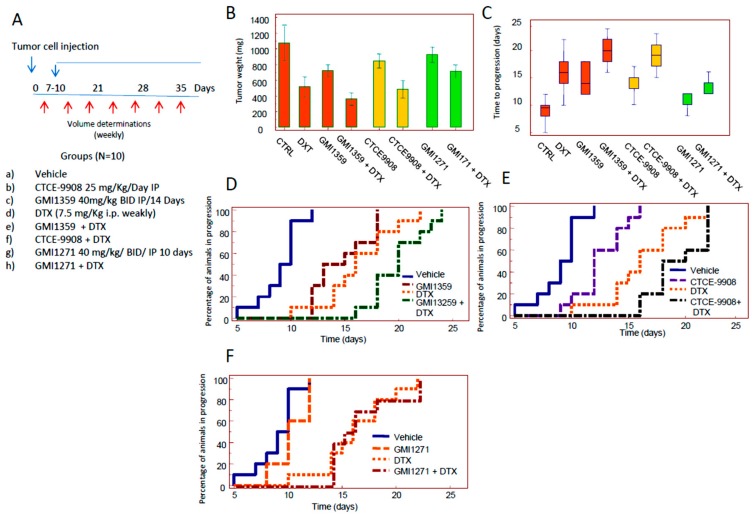
In vivo experiments: comparison of the DTX sensitizing effects of GMI-1359, GMI-1271 and CTCE-9908 administered in subcutaneous 22rv1 xenograft model. (**A**) Diagram of treatments: tumour bearing animals were randomized approximately 10 days or when tumors reached 80–100 mm^3^ of volume in eight experimental groups as follows: Group 1: mice receiving intraperitoneal (i.p.) injections of 100 µL PBS (vehicle); Group 2: mice receiving CTCE-9908 (25 mg/kg, i.p); Group 3: mice receiving GMI-1359 twice a day ip at 40 mg/ kg for consecutive 14 days; Group 4: mice received DTX (7.5 mg/Kg/week; Group 5: mice receiving GMI-1359 and DTX at doses described above for single administration; Group 6. mice receiving CTCE-9908 and DTX at doses described above for single administrations; group 7: mice receiving GMI-1271 40 mg/Kg once daily; Group 8: mice receiving GMI-1271 and DTX at doses described above for single administrations.(**B**) Tumor weight distribution in our treatment cohort. (**C**) TTP analysis. (**D**–**F**) Kaplan Meyer representation for the analyses of chemo-sensitizing effects of GMI-1359 (**D**), CTCE-9908 (**E**) and GMI-1271 (**F**) in 22rv1 xenograft model. Statistics for the comparisons of hazard ratios are shown in Appendix A.

**Figure 6 cells-09-00032-f006:**
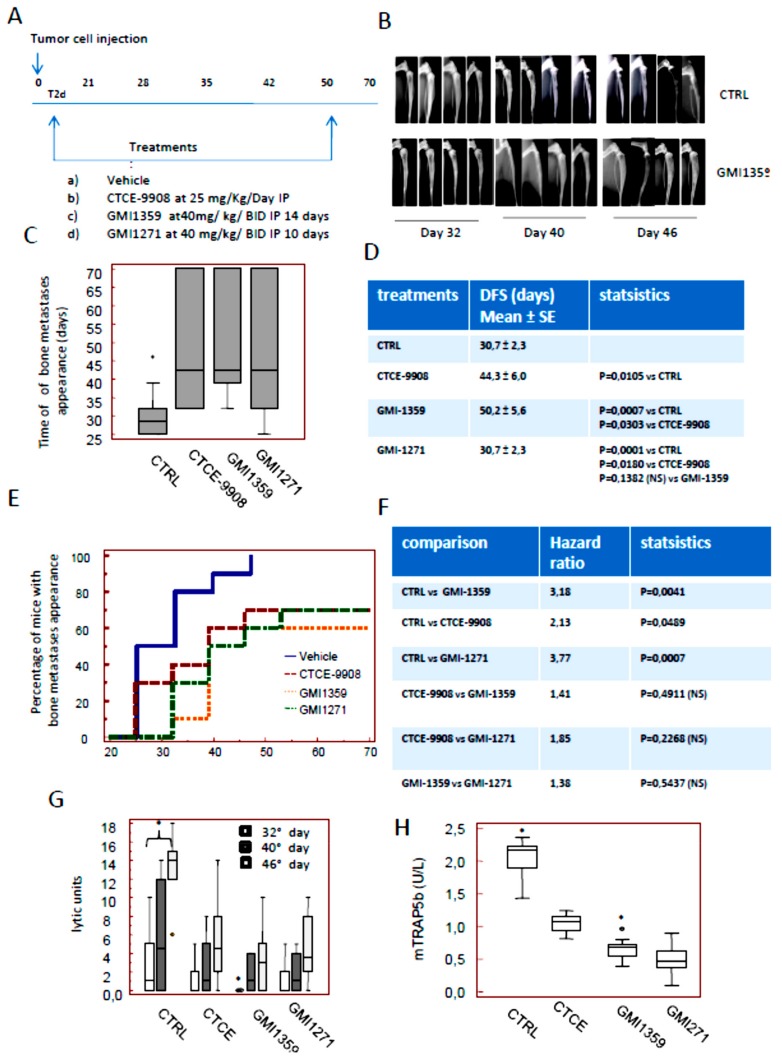
In vivo experiments: intra-ventricular cell injection. (**A**) Treatment schedule showing that pharmacologic treatments were performed two days after cell inoculation. (**B**) Representative radiological images of PC3 tumour lesions as they appeared by Faxitron analysis on day 32, 40 and 46 after heart injection in Controls (CTRL, vehicle) and GMI-1359-treated animals. (**C**) Graphical representation of time of bone metastases appearance in controls (CTRL) and GMI-1359, CTCE-9908 and GMI-1271 treated animals. (**D**) Statistical analyses of data shown in the panel C. (**E,F**) Kaplan Mayer analyses for mice with bone metastases in controls (CTRL) and GMI-1359, CTCE-9908 and GMI-1271 treated animals as indicated in the panel E with hazard ratios of 4.98, 3.79 and 2.81, respectively, as indicated in the panel F. (**G**) Tumour burden (osteolytic units) evaluation in the different treatment groups evaluated at 32, 40 and 46 days from the start of treatment. Osteolysis was calculated only in positive lesions. **p* < 0.05 between T32 and T46 analyses (**H**) Mouse Serum Tartrate-resistant acid phosphatase (mTRAP) serum assay. The asterisks indicate the statistical significance: *p* < 0.05 for the comparisons between CTRL and all treatments and for the comparison between GMI-1359 and CTCE-9908. (**I**) Serum Cross Laps (carboxy-terminal collagen cross links, CTX) assay in the serum. The asterisks indicate the statistical significance: *p* < 0.05 for the comparisons between CTRL and all treatments and for the comparison between GMI-1359 and CTCE-9908 (**J**) Post-mortem necroscopy to document the incidence of visceral metastases in GMI-1359- CTCE-9908- and GMI-1271-treated mice. * indicates a *p* < 0.05 versus control (vehicle). (**K**) Overall survival mean values. (**L**) Kaplan-Meier relative to Overall survival (OS) analyses.

**Figure 7 cells-09-00032-f007:**
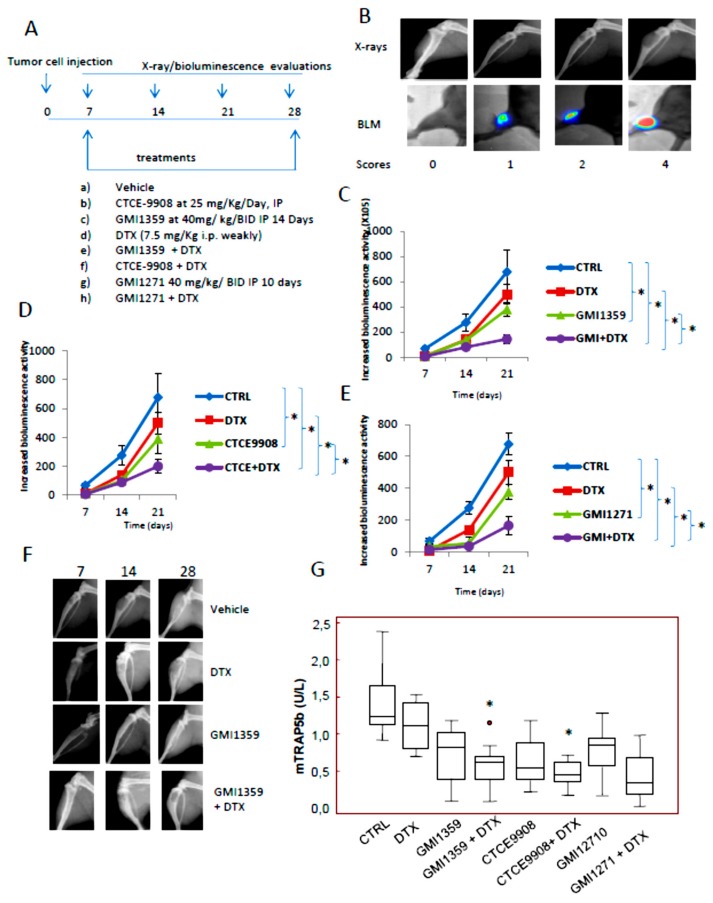
In vivo experiments: intratibial cell injection model. Evaluation of chemo-sensitizing effects of GMI-1359, CTCE-9908 and GMI-1271. (**A**) Treatment schedule; (**B**) Representative radiological appearance of bone lesions as they appear to Faxitron analysis and bioluminescent images for the identification of bone metastatic scores in agreement with Yang et al. [62]. (**C–E**) growth curves generated plotting in the time the increment in bioluminescence activity (see material and methods) for GMI-1359 and combination with DTX (**C**); CTCE-9908 and combination with DTX (**D**); and GMI-1271 alone and combination with DTX (**E**). * *p* < 0.05 for the comparisons versus CTRL in the panels C, E and D at time Day 21 (**F**) Representative radiological images at 7, 14 and 28 days in Vehicle, GMI-1359, DTX and combination GMI-1359 plus DTX. (**G**) mTRAP activity on blood samples harvested from untreated or treated animals. **p* < 0.05 versus control. (**H**) percentage of animals with signs of cachecsia. * *p* < 0.05 calculated by Chi-square test followed by Bonferroni correction as described in the Material and Methods section.

**Figure 8 cells-09-00032-f008:**
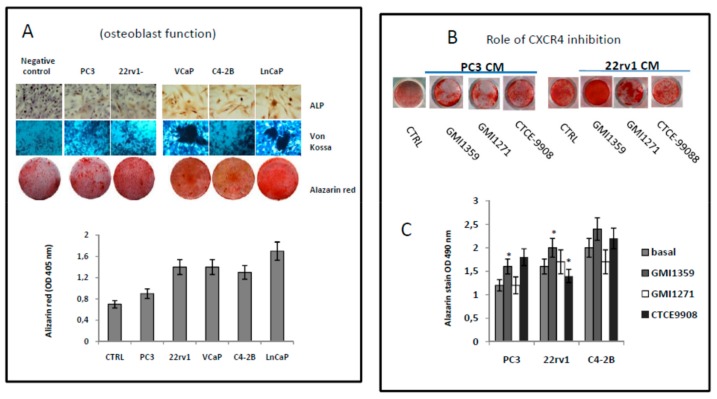
(**A**) Representative staining for ALP, Von Kossa and Alazarin Red staining in osteoblast cultures treated with PC3, LnCaP, 22rv1, C4–2B and VCaP cell conditioned media. (**D**) measured of OD at 450 nm of solubilised Alazarin Red-stained cultures. (**B**) Representative images for the effects of CTCE-9908, GMI-1359 and GMI-1271 on PC3 and 22rv1 conditioned osteoblast cultures on Alazarin Red staining. (**C**) Alazarin Red solubilization for PC3, 22rv1 and C4–2B conditioned osteoblasts. Statistics: * *p* < 0.05.

**Figure 9 cells-09-00032-f009:**
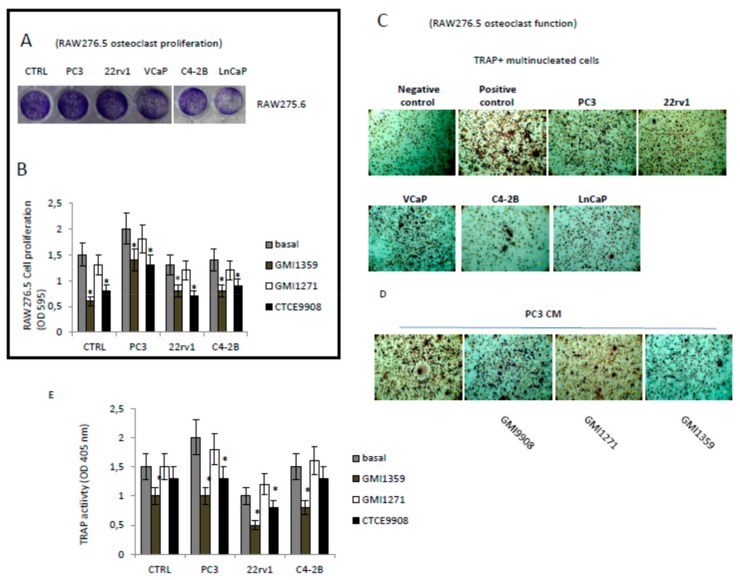
CXCR4 influence osteoclast proliferation: (**A**) RAW275.6 cells were grown in presence of conditioned media from PC3, 22rv1, LnCaP, VCaP and C42B and stained with crystal violet. (**B**) OD495 nm for solubilized RAW276.5 cultures and effects of GMI-1359, GMI-1271 and CTCE-9908 on C4-2B, PC3 and 22rv1 administration. (**C**) Representative staining for tartrate-resistant acid phosphatase (TRAP) in RAW276.5 plates treated with PC3, 22rv1, C4-2B, VCaP and LnCaP cells. (**D**) effects of 0.5 µM CTCE-9908, GMI-1359 and GMI-1271 on TRAP activity from RAW276.5 treated with PC3CM. (**E**) ALP activities (OD 405 nm) measured in RAW275.6 pre-treated with pharmacological doses of CTCE-9908, GMI-1359 and GMI-1271 and successively administered with CMs (diluted 1:4 in complete medium) derived from PC3, 22rv1 and C4–2B PCa cells treated. Statistics: **p* < 0.05.

**Table 1 cells-09-00032-t001:** Variations on osteolytic events and bioluminescence signals in tibiae injected with luciferase taggeted PC3 cells.

Treatment	7° day				14° day				21° day			
	Positive Tibiae	Lytic Scores	Bio Signal Mean ± SE	Lytic Units Mean ± SE	Positive Tibiae	Lytic Scores	Bio Signal Mean ± SE	Lytic Units Mean ± SE	Positive Tibiae	Lytic Scores	Bio Signal Mean ± SE	Lytic Units Mean ± SE
Vehicle	13/16 81.25%	0 (3/16); 1 (8/16); 2 (5/16)	40.2 × 10^4^ ± 7.3	8.5 ± 3.7	13/16 81.25%	0 (3/16); 1 (5/16); 2 (5/16); 3 (3/16)	101.0 × 10^4^ ± 16.3	16.8 ± 2.2	15/16 93.75%	0 (1/16); 1 (2/16); 2 (3/16); 3 (4/16); 4 (6/16)	850.3 × 10^4^ ± 85.0	32.3 ± 5.8
Docetaxel	6/16 37.5%	0(10/16); 1 (2/16); 2 (4/16)	31.9 × 10^4^ ± 11.8	3.2 ± 1.4	7/16 43.75%	0 (9/16); 1 (2/16); 2 (5/16)	45.3 × 10^4^ ± 17.7	3.9 ± 1.1	10/16 62.5%	0 (6/16); 1 (3/16); 2 (3/16); 3 (1/16); 4 (3/16)	348.8 × 10^4^ ± 76.5	19.4 ± 3.7
GMI1359	4/16 25.0 %	0(12/16); 1 (2/16); 2 (2/16)	25.5 × 10^4^ ± 12.4	2.4 ± 1.0	8/16 50.0%	0 (8/16); 1 (3/16); 2 (2/16); 3 (3/16);	98.1 × 10^4^ ± 31.6	3.4 ± 1.8	8/16 50.0 %	0 (8/16); 1 (1/16); 2 (3/16); 3 (3/16); 4 (1/16)	218.5 × 10^4^ ± 49.3	12.9 ± 4.0
GMI1359 + DTX	2/16 12.5 %	0 (14/16); 1 (2/16);	8.75 × 10^4^ ± 6.05	1.7 ± 0.5	4/16 25.0 %	0 (12/16); 1 (2/16); 2 (2/16)	37.5 × 10^4^ ± 19.05	2.0 ± 0.9	4/16 25.0 %	0 (10/16); 1 (2/16); 3 (2/16)	67.1 × 10^3^ ± 30.6	6.3 ± 1.7
GMI1271	8/16 50.0 %	0 (8/16); 1 (3/16); 2 (4/16); 3 (1/16)	23.4 × 10^4^ ± 7.3	6.7 ± 2.9	10/16 62.5 %	0 (6/16); 1 (2/16); 2 (3/16) 3 (2/16); 4 (3/16)	122.9 × 10^4^ ± 78.3	11.8 ± 3.5	10/16 62.5 %	0 (6/16); 1 (1/16); 2 (1/16); 3 (4/16); 4 (4/16)	385.0 × 10^4^ ± 96.6	22.4 ± 5.0
GMI1271 + DTX	4/16 25.0 %	0 (12/16); 1 (4/16)	8.8 × 10^4^ ± 8.2	1.7 ± 0.3	7/16 43.75%	0 (9/16); 1 (4/16); 2 (3/16)	55.8 × 10^4^ ± 22.7	2.5 ± 1.4	8/16 50.0 %	0 (8/16); 1 (2/16); 2 (2/16); 3 (3/16); 4 (1/16)	116.2 × 10^5^ ± 45.6	14.6 ± 2.7
CTCE9908	4/16 25.0 %	0 (12/16); 1 (2/16); 2 (2/16)	33.7 × 10^4^ ± 18.0	2.7 ± 0.8	6/16 37.5 %	0 (10/16); 1 (2/16); 2 (2/16); 3 (2/16)	105.8 × 10^4^ ± 35.6	4.0 ± 1.5	8/16 50.0 %	0 (8/16); 1 (2/16); 2 (2/16); 3 (2/16) 4 (2/16)	257.6 × 10^4^ ± 67.6	15.4 ± 4.2
CTCE9908 + DTX	3/16 18.25%	0 (12/16); 1 (4/16);	17.5 × 10^4^ ± 12.4	1.2 ± 0.4	5/16 31.25 %	0 (11/16); 1 (2/16); 2 (2/16); 3 (1/16)	46.3 × 10^4^ ± 18.2	1.8 ± 0.9	6/16 37.5 %	0 (10/16); 2 (2/16); 3 (2/16); 4 (4/16)	85.5 × 10^4^ ± 30.9	9.4 ± 2.3

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
