# Peer review of "Dual CXCR4 and E-Selectin Inhibitor, GMI-1359, Shows Anti-Bone Metastatic Effects and Synergizes with Docetaxel in Prostate Cancer Cell Intraosseous Growth"

_cells, 2019, doi:10.3390/cells9010032_

Round 1
Reviewer 1 Report
No Comments.
Author Response
We thank this reviewer to consider the report worthy of being published in the Journal "cells"
many thanks again
Claudio
Reviewer 2 Report
Review of cells-605296 (Festuccia et al.): Comments/Suggestions for Authors
This is an interesting paper that presents an impressive amount of data generated in the assessment of the efficacy of a novel small molecule dual inhibitor of CXCR4 and E-selectin, called GMI1359, both alone and in combination with the clinically relevant cytotoxic agent docetaxel (DTX), in models of prostate cancer growth and metastasis. They compare this dual CXCR4/E-selectin inhibitor to other compounds that inhibit just one of these targets: CTCE9908 for CXCR4, and GMI1271 for E-selectin. They conclude that their data provide a strong rationale for use of the dual inhibitor in conjunction with taxane-based chemotherapy in men with metastatic castration-resistant prostate cancer (CRPC) to prevent and reduce bone metastasis, which represents an additional clinical application for an agent that has been evaluated in clinical trials.
The paper is strengthened by the use of multiple cell lines and multiple in vivo models. Moreover, the authors appear to have research expertise in prostate cancer and in the models employed in this manuscript. Nonetheless, there are issues that must be addressed before it is suitable for publication.
1. The manuscript is not well prepared. There seems to have been a general lack of care and attention to detail in the preparation of the manuscript that goes beyond language-related issues. In short, the manuscript feels like an early draft, as opposed to a final submission-ready document. A thorough review and revision of the manuscript is required. Of potential relevance to this problem is that there are a number of highlighted passages in the manuscript suggesting that these passages reflect a revision of the manuscript, perhaps after an initial review(?). If that is the case, then it seems, after these additions, the changes were not carefully integrated into the manuscript to generate a lucid final submission-ready version. Examples of these issues include, but are not limited to, incomplete figure legends, inaccurate reference citations, inaccurate call-outs to figure panels, a call-out to a table that is not presented in the manuscript, discrepancies between what is written in the main text and what is presented in figures (such as methods described but not presented or data presented for which the methods are not described in the Materials and Methods), the apparent reporting of data from human specimens that is not described in the M&M, etc.
2. The conclusion that the findings support the use of the dual inhibitor GMI1359 in combination with DTX for treatment of mCRPC patients (presumably over that of the other compounds) does not appear to be strongly or consistently supported by the data. While GMI1359 appeared to outperform the E-selectin inhibitor (GMI1271) for most endpoints, it was often no better than the CXCR4 inhibitor (CTCE9908). This was the case, for instance, with respect to: (a) Sensitization to DTX and time-toprogression in the sc xenograft model, (b) Overall survival (claimed by the authors, but no data presented), and (c) mTRAP results in the intracardiac model. For other endpoints, clear statistical differences, though sometimes mentioned in the text, were not shown.
3. Specific issues: Specific items of concern are listed below, largely in the order encountered in the manuscript. These will include items that fall within both of the categories listed above.
Line 215 - 222: The M&M describes a method for determining synergy between 2 drugs in the in vivo experiments using the calculation of combination indices (CI). However, no data showing synergy or CI values are presented in the manuscript. Moreover, the term “50% cell kill” is used to indicate effect in single agent and combination treatments. If such an evaluation was performed, how was 50% cell kill determined in this in vivo context?
Line 250: Treatments for in vivo experiments a. Unlike the other agents, no frequency of administration is provided for CTCE9908. b. Group 4: The sentence needs to be re-written. As written, it is difficult to determine if these mice received DTX, vehicle or both.
Fig. 1 and associated text in Results. a. Please introduce these experiments with a statement on the objective of this series of experiments; i.e. why was the expression of CXCR4 and E-selectin ligands analyzed? b. Panel A. The Y-axis indicates CXCR4 expression, but the graph shows both CXCR4 expression and HECA-452 immunoreactivity, and the legend for this panel only mentions HECA immunoreactivity. Please correct the inconsistencies. c. Panels D and F. HECA-452 is an antibody specific for E-selectin ligands; thus, the Y-axis labels should not indicate HECA-452 expression. **Please note that there are multiple instances throughout the manuscript in which “HECA452 expression” or “HECA-452 levels” are said to be changed, induced, etc. (for example, in line 308, it is stated that “HECA-452 levels were significantly increased.”). As indicated above in c., this is incorrect. d. Lines 324 – 327. In reference to Fig. 1 data, the concluding sentence states, “Altogether these data support the experimental observations indicating that 22rv1 cells possessed very low bonetropism with absence of tumour take rate following intra-cardiac inoculation (bone homing) as well as the lower tumour take rate following intra-tibial injection.” Does the statement about the absence of tumor take after IC inoculation refer to previously published data or to the in vivo data the authors will present later in this article (Fig. 6 & 7). This needs to be clarified.
Fig. 2 and associated text. a. Lines 341 - 343: The text descriptions of panels E and F are switched compared to the figure/graphs themselves. b. Legend. The descriptions of panels E and F are missing.
Fig. 3 and associated text. a. Panel C. The value of this data is limited to demonstrating the success of generating DTXresistant cells. Perhaps, it would be more appropriate to either (i) present this data at the time the DTX-resistant cells are first introduced, which was in Fig. 2; or (ii) delete this graph and report the IC50 values in the section of the M&M in which the generation of the DTX-resistant lines is described. b. Panel D. Please indicate directly in the figure legend that the IC20s of GMI1359, GMI1271 and CTCE9908 were used in the combinations with DTX. Also, please indicate the specific dose levels that correspond to the IC20 of each agent. Also, indicate the dose range of DTX used. c. Line 394. The concluding sentence for this data set on the DTX-sensitizing effects of the CXCR4 and E-selectin single and dual inhibitors states, “These data suggest that a dual CXCR4/E-selectin antagonist reduced the DTX resistance in prostate cancer cells.” Although this statement is true based on the data shown, it is also true for the CXCR4 antagonist (CTCE9908), but this is not mentioned as a conclusion. This finding should be part of the summary statement for this data set. Also, this result raises the question of whether the sensitizing effect of the dual inhibitor could be attributed to its CXCR4-targeted activity and not the E-selectin-targeted activity. The authors should discuss this.
Fig. 4 and associated text.
a. Panel A indicates that GMI1271 was administered sc at 5 mg/kg once daily, but the corresponding text (line 401) indicates it was given at 40 mg/kg, ip, bid. Please note that this specific discrepancy occurs at least 2 more times in the manuscript; specifically, in Fig 7A and Fig. 8A. Please revise to report the accurate dose, treatment schedule and route of administration. Such a distinct difference in dose and route diminishes confidence in the data presented. b. Please provide a rationale for the doses and treatment regimens selected. This info can be placed here or in the M&M. c. Line 409. Differences were stated to be significant, but indicators of statistically significant differences are missing in the graph. d. The legend for panel A is inaccurate. The legend lists 4 treatment groups, but the diagram and the graphs in the other panels clearly show 6 groups. e. Panel E. The heading for the first column appears to be inaccurate as it indicates combination treatments with DTX. f. Line 431 and panels F & G: Dot blot data is shown, but a description of the dot blot and densitometry procedures are missing from the M&M. g. Panel G or H. The description of the densitometry data is missing from the legend.
Fig. 5 and associated text. a. Line 448. It is stated that, “In figure 5A we show a diagram of protocol of treatment.” This statement is inaccurate. There is no diagram in Fig. 5. b. Line 447 – 455. This paragraph describes effects of treatments on tumor weights and TTP and indicates that this data is in Fig 5B and C. This is inaccurate. The data and graphs are in Fig. 4. c. Line 452. Regarding the TTP data in Fig. 4C, it is stated that “co-administration of DTX with GMI1359 resulted in a significant increase in TTP when compared to untreated controls (2.1 fold) or single administrations (42,8% and 25% versus GMI1359 or DTX alone, respectively).” a. It seems that the very same thing could be said about CTCE9908 in combination with DTX, yet this is not mentioned by the authors. Please indicate the significant differences in the graph and include p-values. If the CTCE9908 + DTX combination did indeed significantly increase TTP compared to the single agents alone, then the authors must discuss this. d. Line 456 - 457. Based on this sentence (“These data suggest that the dual antagonism…activities alone.”), it is not clear what the authors are concluding from the sc xenograft studies. It seems they want to conclude that, in the sc xenograft experiments, the dual inhibitor treatment resulted in greater chemosensitization than either single target inhibitor alone; in other words, they suggest that, based on hazard ratio, combination treatment with GMI1359+DTX reduced tumor progression vs DTX more effectively than CTCE9908 or GMI1271. The authors need to be very clear about what their conclusion is and must include in their discussion that the CTCE9908+DTX combination also clearly suppressed tumor progression in a statistically significant manner (HR = 16.2 vs CTL, 14.8 vs DTX alone, and 5.2 vs CTCE9908 alone). Another important point that the authors must address is whether the comparison between the dual inhibitor GMI1359 and the CXCR4 inhibitor CTCE9908 is a fair comparison with respect to CXCR4 inhibitory activity. In other words, at the doses selected, is the CXCR4 inhibitory activity of the dual inhibitor comparable to that of the single inhibitor? If not, then maybe the greater activity observed with the dual inhibitor is merely a result of a higher level of CXCR4 inhibition vs the single target inhibitor. This is why the rationale for dose selection of these
agents is important to report, as well as the relative inhibitory activities of each agent on CXCR4, if known. Please include this information and discuss this point in the manuscript. e. Line 457. It is stated that the Kaplan-Meier analysis is shown in Fig. 5D-F. This is inaccurate. The graphs are in Fig 5A-C. f. Legend. The legend is inaccurate. It describes 6 panels (A-F), but only 4 are in the figure. It seems that revisions were made to Figs 4 and 5, but the legend and associated text were not updated.
Fig. 6 and associated text. a. Line 484-486. The authors state that the pattern of bone metastasis observed after intracardiac injection of cancer cells in the current experiment is consistent with “our previous reports [3439].” However, based on review of the author lists of these articles, it seems that only ref #34, 38 and 39 are from any of the authors, while the others are not. Moreover, it seems that ref #34, 38 and 39 do not even involve the use of this PC2b bone metastasis model. Please revise as needed to cite accurately the supporting references. b. Line 487. The authors state bone colonization percentages as follows: “tibiae (80%), femurs (40%) or both (95%).” This reporting of the data appears to be inaccurate. Based on Fig. 6B, the value for “both” should be close to 80%, not 95%. In fact, the % of mice with both tibial and femoral lesions is less than that of tibiae alone according to Fig. 6B. c. Line 489. “PC3” should be “PCb2.” d. Panel A. i. It is not clear or obvious that the 2 images in the upper left of the panel (the cartoon and photo of intracardiac injection) are necessary or helpful. Intracardiac injection is not a rare or novel technique and the images have little instructive value. Removal of these images would not detract from the paper. ii. In the title of this panel, the word "REF" appears; i.e. "(PCb2, REF)." Please clarify what this means and delete if not needed. e. Panel C. i. Need to add scale bars to images to indicate magnification. ii. Need to add arrows, arrowheads and/or some other marker to identify the features of interest, such as CK18-positive tumor cells, CXCR4-positive tumor cells, E-selectinpositive cells, etc. iii. In the 2 upper left panels designated as "No tumors (H&E)," what is the difference between the left and right images? Are they stained differently? iv. Based on the image labels, it seems that the 2 images in the lower right are bone metastasis samples from human PCa patients. If so, this needs to be explained/described in the legend, and the source of these human samples need to be described in the M&M. This description must include all of the info required for the use of human samples, for instance, assurance of institutional approval by appropriate review boards/committees, etc. v. The results from these human samples is not described in the legend nor in the main text of the manuscript.
Fig. 7 and associated text.
a. Panel A. The M&M states that treatments started 24 h BEFORE cell injection, but the diagram shows treatments starting AFTER cell injection. b. Panel B. At the bottom, the labels indicating the day of imaging seem to have small symbols to the upper right of the numbers. What are these symbols and are they necessary? The poor resolution of the image makes it difficult to determine. c. Panel D. The numbers listed in the main text for the % of animals showing bone metastasis at 30, 35, 50, and 70 days do not match the data in the Kaplan-Meier (KM) plot. Examples follow: i. For the vehicle control, the text says 80% of mice had bone lesions at day 70 (see line 516), but the KM graph shows 80% incidence at approximately day 32. Plus, the KM plot shows no control mice reaching day 70. In fact, all of the % values and days listed in the main text for the control group (see Lines 514-516) do not accurately represent the data in the KM graph. ii. Similar discrepancies can be noted for the drug-treated groups. For instance, at day 70, the % for all of the drug-treated groups are 60% or greater, but the main text reports 50% and 40% for GMI1271 and GMI1359, respectively (see Lines 521 and 522). With such wide discrepancies, it is unclear which data set reflects the true experimental results, which raises doubts about the reliability of these data. For instance, if the data in the KM plot is correct, then where did the numbers reported in the text come from, or vice versa? d. In panels C, E and F, there appear to be symbols above or below some of the bars. Are these meant to indicate statistical significance? Please make sure these symbols are aligned appropriately and define what these symbols mean in the legend. e. Legend, Lines 541-542. What does this sentence mean? “GMI1359 was more effective vs GMI1271 (HR=1.78) and CTCE (HR=1.27) also if the significance was not found.” f. Legend, Line 542 refers to “Osteolytic Units” in reference to data in panel E, but in line 548 of the main text, the term used is “Lytic Units.” Please revise the manuscript for consistency of expression. Also, the term “lytic units” is not defined and how it was calculated is apparently not described in the manuscript. This information should be included in the M&M. g. Line 553. In reference to mTRAP and CTX assay results, the authors state, “The levels of these markers were significantly lower in GMI1359 treated animals when compared to GMI1271 or CTCE-9908.” But this statement is not true for mTRAP for which the levels are clearly lower after GMI1271 treatment relative to GMI1359 treatment.
Fig. 8 and associated text. a. Panel B. i. The label above the radiograph images states, "Bone scores by Liang et al." It is not clear what this means. Are these images from a different article by Liang et al? Or, is the method of scoring used in this manuscript adapted from Liang et al.? If the latter is the case, then this scoring system should be described and/or referenced in the M&M, and the text (“Bone scoring by Liang et al.”) should be removed from panel B. ii. Line 568. The authors states that, “bioluminescence images for the identification of osteolytic scores” are shown in panel B. How were BLI values used to develop osteolytic scores? The M&M merely describes how BLI was used to identify/quantify the presence of tumor cells. It is not clear how the presence of luciferase-positive cells was used to generate an osteolytic score. This method must be described in the M&M or a relevant reference cited.
b. Panels C-E. In reference to the graphs in panels C-E, the authors state, “We demonstrated as GMI1359 shows a higher effectiveness in the reduction in tumor growth into the bone when compared with CTCE9908 and GMI1271.” It is very difficult to assess the validity of this claim. The single agent curves in panels C, D, E show very similar BLI values at the 21-day time point; specifically, the BLI values for all 3 of these agents at 21 days appear to be nearly identical (just under 400). In fact, the GM1271-treated group appears to have the lowest BLI value at day 14. Thus, support for this initial claim is lacking. If the authors want to make this claim, then they need to show/report the specific average BLI values for each agent +/- standard deviations and perform statistical analysis to assess differences in efficacy. Until these are shown, the statement that GM1359 is superior cannot be supported. c. Lines 574 – 578. The authors conclude that GMI1359 was superior or comparable to the other 2 agents with respect to osteolysis, and then refer to Table 1 for presentation of lytic units and Fig 8G for mTRAP assay results. First, Table 1 was not included in this manuscript. Secondly, there were no clear distinctions among the DTX+agent combination treatments in the mTRAP assay, and no indications of statistical differences to support the authors claim are presented. d. Lines 579-581. The authors claim that treatment with the compounds delayed the onset of cachexia and weight loss that is typically associated with bone metastasis. Please include these findings in the manuscript.
Fig. 9 and associated text. a. Panel B. The authors claim that Fig. 9B shows that osteoblast proliferation was increased by exposure to CM from PC3, 22rv1 and C42-B cells (Lines 627-629). It is not clear that the data in Fig 9B supports this statement. The Y-axis is labelled “Osteoblast proliferation vs control,” which suggests that the values in the graph are expressed as a fold-change vs control or normalized to control. If that's the case, then the PC-3 CM, with a value of 0.8, induced less proliferation than in control, while the other CMs induced proliferation (1.2, 1.6). If this is correct then the statement must be revised to reflect the data. If this is incorrect or there is a different way to interpret this data, then the authors must be clearer about how the endpoint is expressed. b. Panel A and B. In panel A, the CM of 5 different prostate cancer cell lines were tested for effects on osteoblast proliferation. It is not obvious that, based on the photographs of the CV-stained cells, that the PC3, 22Rv1 and C4-2B CMs induced more proliferation than the VCaP and LNCaP CM. Were OD values for solubilized CV measured for all of the cell lines? If so, these should be shown to demonstrate that PC3, 22rv1 and C42-B CM did indeed induce proliferation. c. Panel C. Regarding Fig 9C, the authors state only, "We observed that the more osteogenic cells induced most osteogenic activity (ALP, Von Kossa and Alazarin red stains)." This conclusion appears to be based solely on the photographs of the stained cells in panel C. It is not at all clear that CM from these cell lines increased staining relative to the other cell lines. In fact, the ALP activity data in panel D appear to show that PC3 CM reduced ALP activity relative to control. d. Panel E. It is suggested to arrange the photographs for each cell type so that the treatments are presented in the same order in both panels, and perhaps even in the same order as in the bar graph in panel F.
Fig. 10. a. Legend. There is no description of panel E in the legend. a. Panel B. Some of the asterisks in the graph are misaligned.
Line 684. It is not clear what this means: “if PC3DTXR”
Lines 687 and 689. References #53 and 54 are cited to support statements about the CXCL12/CXCR4 axis, but, based on their titles, they appear to be about E-selectin.
Lines 698-699. “It is not a simple case that DTX is administered to patients with metastatic and castration resistant disease (mCRPC).” Please clarify what this sentence means.
Lines 720 – 723. “The sensitization versus DTX played by GMI1271, a specific E-selectin antagonist, was still lower when compared to GMI1359 and CTCE9908 suggesting that E-selectin antagonism could increase the efficacy of a CXCR4 antagonism.” The logic of this statement is unclear. As written, this sentence appears to state that the E-selectin antagonist had a lower ability to sensitize to DTX, and thus combining both would increase efficacy. How? Please clarify.
Lines 761. This line contains an abrupt, odd shift to the first person (I.e. the use of “I”). Please revise.
Lines 765 – 771. In this passage, the authors state that overall survival was increased by all three of the agents tested, that GMI1359 (dual inhibitor) was statistically more effective than GMI1271 (E-selectin antagonist), but not different from CTCE-9908 (CXCR4 antagonist). First, while Kaplan-Meier plots were used to analyze time to progression and incidence of bone mets, no Kaplan-Meier data on overall survival was presented. Second, if the dual inhibitor was indeed no better than the CXCR4 inhibitor in prolonging overall survival, then this would not support the superiority of the dual inhibitor.

Author Response
We thank this reviewer for the wide and deep analysis of our report. Overall we agreed with almost all of the questions raised from this reviewer. The extensive changes, which have been required, result in a report which is strongly improved. The critical points raised by this reviewer have made also possible to find little further inconsistencies between text and figures and between figures and their legends in other parts of report and that were not noticed. Altogether all these critical points were addressed and I hope completely fixed.
General considerations.
Question 1. The manuscript is not well prepared. There seems to have been a general lack of care and attention to detail in the preparation of the manuscript that goes beyond language-related issues. In short, the manuscript feels like an early draft, as opposed to a final submission-ready document. A thorough review and revision of the manuscript is required.
Of potential relevance to this problem is that there are a number of highlighted passages in the manuscript suggesting that these passages reflect a revision of the manuscript, perhaps after an initial review(?).If that is the case, then it seems, after these additions, the changes were not carefully integrated into the manuscript to generate a lucid final submission-ready version. Examples of these issues include, but are not limited to, incomplete figure legends, inaccurate reference citations, inaccurate call-outs to figure panels, a call-out to a table that is not presented in the manuscript, discrepancies between what is written in the main text and what is presented in figures (such as methods described but not presented or data presented for which the methods are not described in the Materials and Methods), the apparent reporting of data from human specimens that is not described in the M&M, etc.
Reply 1. I agree with this reviewer on the lack of care in writing this report. As this reviewer rightly understood, some of the statements in the manuscript were related to review processes. In addition, these sentences were not well mixed with the rest of the text. In the first revisions we received comments on secondary aspects of work related to differences between osteosclerotic and osteolytic bone metastases. After having broadly indicated that the PCa osteogenic lines (22rv1, VCaP and C4-2B) had a low engraftment rate in the bone and it was therefore difficult to use them for in vivo studies, the old reviewer still claimed to use them in our studies in vivo. It must be considered that in any case, osteolysis is the final event of bone metastasis in the humans.
We believe this event should be prevented and well treated to improve the quality of life and the outcome of patients with mCRPC. I thought the old reviewer would see this new version. So, I had highlighted in yellow the appropriate statements. Many times in the text we have to justify the use of osteolytic cells. This have worsened/burdened the report creating confusion and difficulty in reading.
Now, since I am in front of another reviewer, I have reformulated the report sequence accordingly to the "most relevant" questions and for this I think again this reviewer. Accordingly these general considerations, the text was modified:
Question 2. The conclusion that the findings support the use of the dual inhibitor GMI1359 in combination with DTX for treatment of mCRPC patients (presumably over that of the other compounds) does not appear to be strongly or consistently supported by the data. While GMI1359 appeared to outperform the E-selectin inhibitor (GMI1271) for most endpoints, it was often no better than the CXCR4 inhibitor (CTCE9908).
This was the case, for instance, with respect to:
(a) Sensitization to DTX and time-to progression in the sc xenograft model,
Reply 2.
This reviewer believes that data presented here support only partially the conclusion that the dual inhibitor GMI-1359 in combination with DTX may be considered for treatment of mCRPC patients. The reviewer states that this combination is surely better to the GMI-1271 plus DTX treatment. According to the reviewer, this suggestion was particularly evident in the sensitization to DTX in the sc xenograft model. However, we do not completely agree with this suggestion. The reply to this question is present in a wider manner below in the reply to specific queries.
If CTCE-9908 increased the sensitivity to DTX in vitro with good efficacy in all cell lines, this effect was superior in the PC3 sensitive DTX cells for the combination GMI-1359 plus DTX. For DTX resistant cell lines GMI1359 showed more significant increments of sensitization to DTX of CTCE-9908. So, at least in vitro, GMI1359 shows the greatest sensitization effects for DTX.
For example see comments U2 -U4, U17, U18 etc....
i.e. lines 52-54
87-98
368-369, 375-391, 395-411 etc..... see specific points below
Reply 2a. Next we analyzed the in vivo data showed in figures 4 and 5.
See comments U18 (results lines 399-415)
Comment U lines……..
Comment U. lines ……
In figure 4E we demonstrated that the single administration of GMI-1359 shows a better effectiveness when compared to single administration of CTCE-9908 (HR=2.8, P=0.0366). In figure 5D , the statistical comparison between the combinations GMI1359 plus DTX and CTCE-9908 plus DTX shows the similar HR value (HR=2.9) to advantage of GMI-1359, accordingly to the performed LogRank analysis This means that GMI-1359 was 2,9 fold more active in the sensitization to DTX compared to CTCE-9908. Unfortunately, the statistical analyses reveled that this comparison was not significant whereas the comparison of single administrations (GMI-1359 versus CTCE-9908) showed different statistic values (despite having the same HR value). This mean that, assuming that HR value was confirmed, it was necessary a greater number of replicates (animals) to obtain more significant statistics. Nevertheless by the data showing here for sc xenografts, indicate that there is an upward trend in the sensitization to DTX in GMI1359 when compared to CTCE9908. Although the combination between GMI-1359 and DTX seems to be, indeed, low when compared to the combination between CTCE-9908 and DTX, we can be satisfied by a similar increment of activity. A low increase of effectiveness is welcome for the mCRPC treatments. It is necessary to consider, indeed, that the mCRPC is a lethal and almost incurable disease and the pharmacological therapies (ie. Use of bisphosphonates) are aimed at improving the quality of life of metastatic patients through the reduction of osteolysis and bone fractures.
see comment U36 lines 967-978
In addition, analyzing more widely figures 4 and 5 we noticed other inconsistencies. In the new figure 4E, repeating the statistical analyses for Kaplan-Meier data with the software "MedCalc" we have found that: (1) The terms of comparisons shown in the first column of table in this figure were inverted. So, the right comparisons were CTCE-9908 vs GMI1359 and GMI-1271 vs GMI-1359 whereas other comparisons were OK (we fixed this in the new figure).
The values of HR calculated for the comparisons DTX vs CTCE-9908 plus DTX and DTX vs GMI1359 plus DTX were also inverted in the old version of the report . So, the first comparison (DTX plus CTCE9908) had an HR=8,4 (P=0.0003) and the second comparison (DTX plus GMI-1359) had an HR=14,8 (P<0.001). This was fixed the new version of figure 5. This consideration supports the statements that GMI-1359 and CTCE-9909 were able to sensitize to DTX administration in vivo and that the dual CXCR4/E selectin antagonism could have better effectiveness. Altogether, these argumentations were added in the text of the new version of report.
(b) Overall survival (claimed by the authors, but no data presented),
Reply 2b. We agree with the Reviewer when states that no OS are presented in this version of report. When we had to fuse the data of two in vivo figures (in agreement with the request of the old reviewer), we have forgotten to insert the relative panels. So, in this report we added the two lacking panels (panels K and L) in the new figure 6: figure 6K we show the rate of mortality rates of untreated (CTRL) or treated animals. In figure 6L we show the statistical analyses of these evaluations.
Comment U29, lines 630-640
and (c) mTRAP results in the intracardiac model. For other endpoints, clear statistical differences, though sometimes mentioned in the text, were not shown.
Reply 2c. I agree with reviewer for the absence of statistical data presentation in the old figure 8G (now figure 7G) and its legend for mTRAP. So we added this information in the new version of report. However, as indicated mTRAP results indicate that GMI1359 effects were significantly different (p<0.05) when compared to those observed in the single CXCR4 antagonist (CTCE9908) and not for the comparison with GMI1271.
Comments U31 and U32
lines 709-711 and 727.
Specific issues:
Specific items of concern are listed below, largely in the order encountered in the manuscript. These will include items that fall within both of the categories listed above.
Question3. Line 215 - 222: The M&M describes a method for determining synergy between 2 drugs in the in vivo experiments using the calculation of combination indices (CI). However, no data showing synergy or CI values are presented in the manuscript.
Reply 3. We agree with this reviewer. The data on CI was not provided in the text. So, we added CI data in this new version of the report.
See comments: U9 (material and methods, lines 224-228)
Question 4. Moreover, the term “50% cell kill” is used to indicate effect in single agent and combination treatments. If such an evaluation was performed, how was 50% cell kill determined in this in vivo context?
Reply 4. We agree with the reviewer on the fact that this statement may be confusing. We noticed that confusing was also the formula for calculation of CI in vivo experiments when only single doses of compound were used in their combination. For this reason, we calculated again the CI by using the method published in the article entitled "Survival analysis: time-dependent effects and time-varying risk factors. written by Dekker FW, de Mutsert R, van Dijk PC, Zoccali C, Jager KJ. and published in Kidney Int. 2008 Oct;74(8):994-7. doi: 10.1038/ki.2008.328. Epub 2008 Jul 16". This reference was added in the text as REF 41. So, considering that the synergy/additivity index may be calculated using different modality as for example considering relative risks (RR), odds ratios (OR), or hazard ratios (HR), we preferred to calculate CI through HR as CI= [HR(a) + HR(b)]: HR(a+b). The statements relative to this aspect were changed accordingly.
See comment U9
Question 5. Line 250: Treatments for in vivo experiments.
Unlike the other agents, no frequency of administration is provided for CTCE9908.
Reply 5a. This was made. We added this information when required in the M&M section and figure legend.
See comment U11 lines 251-260
5b. Group 4: The sentence needs to be re-written. As written, it is difficult to determine if these mice received DTX, vehicle or both.
Reply 5b: This was made accordingly.
See comment U11 lines 259-268
Question 6. Fig. 1 and associated text in Results.
Please introduce these experiments with a statement on the objective of this series of experiments; i.e. why was the expression of CXCR4 and E-selectin ligands analyzed?
Reply 6a. This was clarified better accordingly.
See comment U11 lines 290-293
6b. Panel A. The Y-axis indicates CXCR4 expression, but the graph shows both CXCR4 expression and HECA-452 immunoreactivity, and the legend for this panel only mentions HECA immunoreactivity. Please correct the inconsistencies.
Reply 6b: This is clarified in this new version of figure 1A.
See comment U14 lines 338-340 (legend of figure 1)
6c. Panels D and F. HECA-452 is an antibody specific for E-selectin ligands; thus, the Y-axis labels should not indicate HECA-452 expression. **Please note that there are multiple instances throughout the manuscript in which “HECA452 expression” or “HECA-452 levels” are said to be changed, induced, etc. (for example, in line 308, it is stated that “HECA-452 levels were significantly increased.”). As indicated above in c., this is incorrect. d. Lines 324 – 327.
Reply 6c. I agree with the reviewer. The FACS determination was made with the antibody HECA-452 and for easiness we referred as HECA-452 expression. We modified this figure and text in all its part substituting the term HECA-452 expression with the term HECA-452 immune-reactivity (IR). In addition we added in the Y axis of panel 1A that both CXCR4 and HECA452 immuno-reactivity were analyzed.
See comment U14 lines 340-350
Question 7. In reference to Fig. 1 data, the concluding sentence states, “Altogether these data support the experimental observations indicating that 22rv1 cells possessed very low bone tropism with absence of tumour take rate following intra-cardiac inoculation (bone homing) as well as the lower tumour take rate following intra-tibial injection.” Does the statement about the absence of tumor take after IC inoculation refer to previously published data or to the in vivo data the authors will present later in this article (Fig. 6 & 7). This needs to be clarified.
Reply 7. I agree with the reviewer on the confusing sentence due mainly to different modifications required during the old revisions of this report. The sentence on tumor take after IC inoculation derived from literature and from previously our failed attempts to grow 22rv1 into the bone (data not shown since unnecessary). We eliminated this statement here since in different parts of the new report this is provided.
See comment U13 (elimination of this statement).
Question 8. Fig. 2 and associated text.
Lines 341 - 343: The text descriptions of panels E and F are switched compared to the figure/graphs themselves. Legend. The descriptions of panels E and F are missing.
Reply 8a, b: This was fixed accordingly. See new version of text and figure 2 legend.
See comments U15 lines 348-350 and U16 lines 368-369 and U19 lines 419-420
Question 9. Fig. 3 and associated text.
Panel C. The value of this data is limited to demonstrating the success of generating DTX resistant cells. Perhaps, it would be more appropriate to either (i) present this data at the time the DTX-resistant cells are first introduced, which was in Fig. 2; or (ii) delete this graph and report the IC50 values in the section of the M&M in which the generation of the DTX-resistant lines is described.
Reply 9a: Accordingly with this reviewer, panel C was eliminated and IC50 values reported in MM section. We added also the reference 47 in which DTX resistant cells were previously described. The numbering of references was fixed accordingly. See comment U6 lines 148-156
Panel D. Please indicate directly in the figure legend that the IC20s of GMI1359, GMI1271 and CTCE9908 were used in the combinations with DTX. Also, please indicate the specific dose levels that correspond to the IC20 of each agent. Also, indicate the dose range of DTX used.
Reply 9b: This was made accordingly. See comment U20 lines 429-435
c. Line 394. The concluding sentence for this data set on the DTX-sensitizing effects of the CXCR4 and E-selectin single and dual inhibitors states, “These data suggest that a dual CXCR4/E-selectin antagonist reduced the DTX resistance in prostate cancer cells.” Although this statement is true based on the data shown, it is also true for the CXCR4 antagonist (CTCE9908), but this is not mentioned as a conclusion. This finding should be part of the summary statement for this data set. Also, this result raises the question of whether the sensitizing effect of the dual inhibitor could be attributed to its CXCR4-targeted activity and not the E-selectin-targeted activity. The authors should discuss this.
Reply 9c.
See also Reply 2 to general queries.
I agree with this rewiever.
However, we have not stated that GMI-1359 was active as DTX sensitizing agent whereas CTCE-9908 was not active.
In vitro data:
We observed that GMI-1359 and CTCE-9908 potentiated the effects of DTX in drug-sensitive PC3 cells. The DTX sensitivity was higher in presence of a co-treatment with GMI-1359 (4,8 times vs CTRL) when compared to CTCE-9908 (2,7 times vs CTRL). However, GMI-1271 showed the lower effects.
When we compared the effectiveness of these combinations in DTX resistant cells we show that GMI-1359 was still more active (from 2,4 to 3,3 times vs CTRL) when compared to CTCE-9908 (1,8 to 2,5 times vs CTRL) and GMI-1271 (1,2 to 1,7 times vs CTRL) in all used cells.
In reference to the fact that the results shown in figure 3 raises the question of whether the sensitizing effect of the dual inhibitor could be attributed to its CXCR4-targeted activity and not the E-selectin-targeted activity, we cannot discriminate this.
We can state only that the inhibitory ability against E-selectin played by GMI-1359 could ameliorate (even if only slightly) the efficacy of the single CXCR4 antagonism of this molecule.
So in a close in vitro model the dual inhibitor plays better effects on DTX sensitivity.
This was added in the results
see comments U17: lines 375-391
U8: lines 395-411
U21: Lines 437-439
U37: lines 849-885
U43: lines 863-915
Question 10. Fig. 4 and associated text.
Panel A indicates that GMI1271 was administered sc at 5 mg/kg once daily, but the corresponding text (line 401) indicates it was given at 40 mg/kg, ip, bid.Please note that this specific discrepancy occurs at least 2 more times in the manuscript; specifically, in Fig 7A and Fig. 8A.
Please revise to report the accurate dose, treatment schedule and route of administration. Such a distinct difference in dose and route diminishes confidence in the data presented.
Reply 10a: I'm sorry for this mistake. I agreed with the reviewer that "this could reduce the confidence of our data" and I'm sorry for this again. We checked the recommended doses for all drugs by literature and confirmed that the dose of 40 mg/Kg ip bid was used for CTCE9908 [reference 55] and 5 mg/Kg/days [reference 56] was used for GMI-1271. we added also doses for GMI1359 [reference 54] and DTX [references 47, 57, 58]. This information was indicated in the new panels A of figures 4, 5, 6 and 7. So, this was fixed also in the text. I check, indeed, also this mistake in other parts of report and modified also the M&M specific section.
See also comments
U11 lines 259-268
U23 lines 464-467
Please provide a rationale for the doses and treatment regimens selected. This info can be placed here or in the M&M.
Reply 10b. Doses are selected from the literature for preclinical data [54-58]. This was added to M&M as suggested.
see comment U11 lines 259-268
U23 lines 464-467.
Line 409. Differences were stated to be significant, but indicators of statistically significant differences are missing in the graph.
Reply 10c. Statistical data for the panels B and C were omitted in the old version of figure 4. This figure was completely modified in this new version of report. We have added, indeed, two new panels that define the tumor weights (figure 4B) and TTPs (figure 4C) for GMI-1359, CTCE-9908 and GMI1271 as single therapies. So, we added also the statistical analyses in supplementary materials (figure S1A, B) and in the text.
See comment U24 lines 464-470.
The legend for panel A is inaccurate. The legend lists 4 treatment groups, but the diagram and the graphs in the other panels clearly show 6 groups.
Reply 10d. This has been fixed. Single treatments was analyzed only in the figure 4, in which four groups of treatments were considered (see new figure 4), whereas combination treatments are analyzed in figure 5 (see new figure 5) in which eight groups of treatment were considered.
See comment U22 lines 462-464
Panel E. The heading for the first column appears to be inaccurate as it indicates combination treatments with DTX.
Reply 10e. The panel was fixed as “Statistical comparisons”.
Line 431 and panels F & G: Dot blot data is shown, but a description of the dot blot and densitometry procedures are missing from the M&M.
Reply 10f. A section of dot blot procedure has been added in M&M.
See comment U8 lines 194-198
9 g. Panel G or H. The description of the densitometry data is missing from the legend.
Reply 10g. Data in the panel 4G represents arbitrary densitometric values (OD). Dots were converted at 16 bit in gray scale and analyzed by image J with the attribution of arbitrary units. These values were statistically analyzed and the * indicates a p<0,05 statistical value.
Panel 4H shows, instead, E-selectin Elisa determinations in the plasma of untreated or treated animals also in this case * indicated a p<005 statistical value.
This has been fixed in the figure legend.
See comment U25, lines 472-473.
Question 11. Fig. 5 and associated text.
Line 448. It is stated that, “In figure 5A we show a diagram of protocol of treatment.” This statement is inaccurate. There is no diagram in Fig. 5.
Reply 11a. I'm sorry for this mistake This was fixed in the new version of fig.5 with the new diagram of treatments containing 8 experimental groups. Figure 4 shows the diagram containing 4 groups only.
comment U27 lines 527-533
Line 447 – 455. This paragraph describes effects of treatments on tumor weights and TTP and indicates that this data is in Fig 5B and C. This is inaccurate. The data and graphs are in Fig. 4.
Reply 11b. This was described in figure 4B (I'm sorry) and this is due to some changes performed during old revision phases. This was fixed appropriately.
Line 452. Regarding the TTP data in Fig. 4C, it is stated that “co-administration of DTX with GMI1359 resulted in a significant increase in TTP when compared to untreated controls (2.1 fold) or single administrations (42,8% and 25% versus GMI1359 or DTX alone, respectively).”
(i). It seems that the very same thing could be said about CTCE9908 in combination with DTX, yet this is not mentioned by the authors.
Please indicate the significant differences in the graph and include p-values.
(ii) If the CTCE9908 + DTX combination did indeed significantly increase TTP compared to the single agents alone, then the authors must discuss this.
Reply 11c. see also reply 2 in the general questions and reply 9c of specific points. I agree with the reviewer . So, we added the statistical analyses in supplementary material for tumor weight (figure S1A), TTP (figure S1B). In the text we have added data for TTP determination relative to CTCE-9908 and GMI-1271 as suggested. However, we did not state that GMI-1359, and not CTCE-9908, increased TTP values in 22rv1 xenografts, but that the increment of TTP was higher in the animals treated with GMI-1359. So, in the new version of report we considered that: “GMI-1359 and CTCE-9908 increase TTP values.” See comment U26, lines 493-505
(iii). Line 456 - 457. Based on this sentence (“These data suggest that the dual antagonist activities alone.”), it is not clear what the authors are concluding from the sc xenograft studies. It seems they want to conclude that, in the sc xenograft experiments, the dual inhibitor treatment resulted in greater chemosensitization than either single target inhibitor alone; in other words, they suggest that, based on hazard ratio, combination treatment with GMI1359+DTX reduced tumor progression vs DTX more effectively than CTCE9908 or GMI1271. The authors need to be very clear about what their conclusion is and must include in their discussion that the CTCE9908+DTX combination also clearly suppressed tumor progression in a statistically significant manner (HR = 16.2 vs CTL, 14.8 vs DTX alone, and 5.2 vs CTCE9908 alone).
Reply 10c (iii) Statistical analyses for the hazard ratio analysis for each combination are moved in materials in the figure S1C. We considered that the CTCE9908+DTX combination also clearly suppressed tumor progression in a statistically significant manner vs untreated animal controls. This was added in the discussion. Although the data indicate that the combination CTCE9908 + DTX shows a clear sensitization versus DTX alone, the DTX was greater for the combination GMI1359 + DTX. The comparison between CTCE + DTX vs GMI1359 + DTX showed, indeed, an HR=2.8 with an advantage of the second combination on the first one. However, the statistical analyses was not significant. This means that if the number of treated animals had been greater and maintaining the HR value of HR=2.8 this comparison should be significant.
It necessary also to note that the HRs values for the comparisons GMI1359 + DTX and CTCE9908 + DTX versus the control have been inverted in this panel. In fact, considering the Kaplan-Meier curves we observe that the trend of the combination CTCE + DTX is closer to the DTX alone respect to those observed for GMI1359 + DTX. In addition the final values of the progression percentage differ between the two groups: in CTCE + DTX we have a maximal value of 22 days with 40% of animals in progression at this time whereas the final progression value in GMI1359 + DTX was 24 days with 30% animals showing a progression rate lower. The two curves (CTCE + DTX vs DTX alone) overlap while they are far apart in GMI1359 + DTX against DTX alone.
So I reckoned with the "medcalc software "and fixed the inversion. Changes were introduced at comment U26
(iv) Another important point that the authors must address is whether the comparison between the dual inhibitor GMI1359 and the CXCR4 inhibitor CTCE9908 is a fair comparison with respect to CXCR4 inhibitory activity. In other words, at the doses selected, is the CXCR4 inhibitory activity of the dual inhibitor comparable to that of the single inhibitor? If not, then maybe the greater activity observed with the dual inhibitor is merely a result of a higher level of CXCR4 inhibition vs the single target inhibitor.
This is why the rationale for dose selection of these agents is important to report, as well as the relative inhibitory activities of each agent on CXCR4, if known. Please include this information and discuss this point in the manuscript.
Reply 10c (iv). I agree with the reviewer that it should be stressed that the comparison between the dual inhibitor GMI1359 and the CXCR4 inhibitor CTCE9908 is not a fair comparison with respect to CXCR4 inhibitory activity and if, at the doses selected, is the CXCR4 inhibitory activity of the dual inhibitor comparable to that of the single inhibitor. Comparative analyses are not available in literature for the activities of CTCE-9908 and GMI1359 versus CXCR4, some data exist for in vitro assays. However, to be sure of the above, we tested CXCR4 activity by usingPhospho-CXCR4 (Ser339) Colorimetric Cell-Based ELISA Kit (OKAG01771, Aviva Systems Biology, Corp, San Diego, CA). No differences were noticed (data not shown)
In addition, to demonstrate that this was manifest also in vivo, we have to consider the results shown in figure 4F, G on the levels of CXCR4 after treatment with CTCE-9908 or GMI1359 through dot blots analysis. Here we show that at commonly used pharmacological doses, the levels of CXCR4 were not significant different between the two groups and showed similar decrement respect to those observed for the control (untreated animals). This result indicates that since the reduction/disappearance of CXCR4 is a marker for the effective CXCR4 inhibition, CTCE-9908 and GMI1359 showed similar inhibitory effects on this event at the used dosage.
So, the greater activity observed with the dual inhibitor cannot be merely a result of a higher level of CXCR4 inhibition vs the single target inhibitor.
This was stressed.
See comment U5, lines 101-102 and comment 40, lines 886-903
Line 457. It is stated that the Kaplan-Meier analysis is shown in Fig. 5D-F. This is inaccurate. The graphs are in Fig 5A-C.
Reply 10e. I'm sorry for this mistake. This was fixed in the new version of report.
Panel B and C of Figure 4 have been moved in the new figure 5 (B, C).
Statistical analyses present in the Table of panel 5D were moved in supplementary material figure S1C.
Legend.
The legend is inaccurate. It describes 6 panels (A-F), but only 4 are in the figure. It seems that revisions were made to Figs 4 and 5, but the legend and associated text were not updated.
Reply 10f. I'm sorry for mistake see reply 10e. This was fixed.
Question 11. Fig. 6 and associated text.
Line 484-486. The authors state that the pattern of bone metastasis observed after intracardiac injection of cancer cells in the current experiment is consistent with “our previous reports [3439].” However, based on review of the author lists of these articles, it seems that only ref #34, 38 and 39 are from any of the authors, while the others are not. Moreover, it seems that ref #34, 38 and 39 do not even involve the use of this PC2b bone metastasis model.
Please revise as needed to cite accurately the supporting references.
Reply 11a. This was fixed accordingly.
Line 487. The authors state bone colonization percentages as follows: “tibiae (80%), femurs (40%) or both (95%).” This reporting of the data appears to be inaccurate. Based on Fig. 6B, the value for “both” should be close to 80%, not 95%. In fact, the % of mice with both tibial and femoral lesions is less than that of tibiae alone according to Fig. 6B.
Reply 11b. I thanks the reviewer to note this discrepancy.
We verified the data and moved data in figure S2A, accordingly.
Line 489. “PC3” should be “PCb2.”
Reply 11c. This was fixed
Panel A. It is not clear or obvious that the 2 images in the upper left of the panel (the cartoon and photo of intracardiac injection) are necessary or helpful. Intracardiac injection is not a rare or novel technique and the images have little instructive value. Removal of these images would not detract from the paper.
Reply 11d (i). I agree with the reviewer that these pictures are not necessary. However, I think that this figure can be moved in supplementary materials as figure S2.
In the title of this panel, the word "REF" appears; i.e. "(PCb2, REF)." Please clarify what this means and delete if not needed.
Reply 11d(ii). REF indicates "to add reference". This was fixed..
Panel C. Need to add scale bars to images to indicate magnification.
Reply 11e (i). This was made
Need to add arrows, arrowheads and/or some other marker to identify the features of interest, such as CK18-positive tumor cells, CXCR4-positive tumor cells, E-selectin positive cells, etc.
Reply 11c (ii). This was made in the new figure S2A
iii. In the 2 upper left panels designated as "No tumors (H&E)," what is the difference between the left and right images? Are they stained differently?
Reply 11c (iii). No. Both images are H/E stains. The one on the right with a smaller magnification has taken less color and concentrates only on the medullar portion of the tibia bone marrow, a part that is less colored with eosin and more with hematoxylin. Since these images don't provide useful information for the aims of the report, I think that it is possible to move it in supplementary materials as Figure S2B.
Based on the image labels, it seems that the 2 images in the lower right are bone metastasis samples from human PCa patients. If so, this needs to be explained/described in the legend, and the source of these human samples need to be described in the M&M. This description must include all of the info required for the use of human samples, for instance, assurance of institutional approval by appropriate review boards/committees, etc.
Reply 11c (iv). Yes these images correspond to human samples and derived from a tissue micro-array purchased from US Biomax. Inc. This information was added in the legend of figure S2B.
The results from these human samples is not described in the legend nor in the main text of the manuscript.
Reply 11d (v).
This was fixed see reply 11c (iv)
Question 12.
Fig. 7 and associated text.
Panel A. The M&M states that treatments started 24 h BEFORE cell injection, but the diagram shows treatments starting AFTER cell injection.
Reply 12a. I'm sorry for this mistake in the text. As the two arrows related to cell inoculation and treatments occupy demonstrated a separate temporal portions of 2 days. So, I added "T2d" to indicated the start of experiments at 2 days.
Panel B. At the bottom, the labels indicating the day of imaging seem to have small symbols to the upper right of the numbers. What are these symbols and are they necessary? The poor resolution of the image makes it difficult to determine.
Reply 12b. The symbols represent the "°" or "th" for the days
i.e 20°; 30° etc. we change without symbols.
If the report will be accepted for publication, final report will be completed with high quality images (300-600dpi)
Panel D. The numbers listed in the main text for the % of animals showing bone metastasis at 30, 35, 50, and 70 days do not match the data in the Kaplan-Meier (KM) plot.
Examples follow:
For the vehicle control, the text says 80% of mice had bone lesions at day 70 (see line 516), but the KM graph shows 80% incidence at approximately day 32. Plus, the KM plot shows no control mice reaching day 70. In fact, all of the % values and days listed in the main text for the control group (see Lines 514-516) do not accurately represent the data in the KM graph.Similar discrepancies can be noted for the drug-treated groups. For instance, at day 70, the % for all of the drug-treated groups are 60% or greater, but the main text reports 50% and 40% for GMI1271 and GMI1359, respectively (see Lines 521 and 522).
Reply 12C.
We thank this reviewer for having noticed these inaccuracies between the text and figure 7. We performed our X-ray evaluations approximately 1 time/a week and here show results at days 32, 40, 46 e 53. So panel B and text was also fixed accordingly.
At these days correspond different percentage of positivity comparing different groups. I'm sorry for the erroneous comparison between text and Kaplan Meier analyses.
(i) In the panel D, we show that PCb2 cells produce X-ray positive bone lesions starting from day 25 in which 5/10 (50%) injected mice were affected . Bone metastases progressively increased to 80% (8/10) on day 32, to arrive at 100% (10/10) on day 46. So, Kaplan Meier plots do not show any control mice reaching day 70. This was fixed in the new version of the report.
In contrast, we found radiographic evidence of osteolytic bone lesions in 3/10 (30%) mice treated with CTCE-9908 on day 25. Bone metastases reached 40% (4/10) on day 32, 60% (6/10) on day 40 with a final incidence of 70% (7/10) on day 53 (until day 70), whereas in GMI1271 –treated mice was 1 out of 10 on day 32. Bone metastases reached 60% (6/10) on day 40 and remain at this percentage until day 70. The radiographic evidence of osteolytic bone lesions in GMI1359 treated animals was 3 out of 10 (30%) mice at day 32, reached 50% (5/10) at day 40 and 70% (7/10) at day 46 without other animals with skeletal involvement until day 70.
See comments U 30, lines 559-598
12d (iii). With such wide discrepancies, it is unclear which data set reflects the true experimental results, which raises doubts about the reliability of these data. For instance, if the data in the KM plot is correct, then where did the numbers reported in the text come from, or vice versa?
Reply 12d (iv). Experiment with bone metastases was performed together with other groups of treatment and so it is possible that data collection reflect a scarce attention of the principal investigator (and that is me) who assembled the data.
I take responsibility for this.
The figure, instead, were performed by my collaborators which have been much more careful of my. I'm sorry that he/she has raised doubts about the veracity of the data and reading the report I would have done it too. But I believe in this report, and among other things, none of co-authors, collaborators and previous reviewers noticed these inconsistencies and this makes me very unhappy.
I hope that this reviewer may be glad that after this review the report will result much improved and perhaps made worthy of being published.
In panels C, E and F, there appear to be symbols above or below some of the bars. Are these meant to indicate statistical significance? Please make sure these symbols are aligned appropriately and define what these symbols mean in the legend.
Reply 12d. * represent the statistical value p<0.05. I worked for the symbols to be aligned.
Legend, Lines 541-542. What does this sentence mean? “GMI1359 was more effective vs GMI1271 (HR=1.78) and CTCE (HR=1.27) also if the significance was not found.”
Reply 12e. We changed this statement as: Although differences were not significant GMI1359 shows a trend of better efficacy when compared to GMI1271 and CTCE-9908.
See comment U31, lines 630-640.
Legend, Line 542 refers to “Osteolytic Units” in reference to data in panel E, but in line 548 of the main text, the term used is “Lytic Units.” Please revise the manuscript for consistency of expression. Also, the term “lytic units” is not defined and how it was calculated is apparently not described in the manuscript. This information should be included in the M&M.
Reply 12f. Osteolytic units or lytic units were used to indicate the same effect. So, we uniformed the term as lytic units. This was fixed in the text.
see comment U10, lines 243-246.
Line 553. In reference to mTRAP and CTX assay results, the authors state, “The levels of these markers were significantly lower in GMI1359 treated animals when compared to GMI1271 or CTCE-9908.” But this statement is not true for mTRAP for which the levels are clearly lower after GMI1271 treatment relative to GMI1359 treatment.
Reply 12g. Thanks for this request of clarification.
The reviewer is right and the text is modified accordingly. So we state that:
CTX-I and mTRAP are significantly higher in the untreated animals (*p<0.05) when compared to treated animals. The comparison for CTX-I levels indicated that GMI-1359 was statistically more active respect to CTCE-9908 and GMI-1271 (*p<0.05). The levels of mTRAP were significantly lower in GMI-1359 when compared to CTCE-9908 whereas were not statistically different in the comparison GMI-1359 and GMI-1271. It is necessary to note that the CTX-I levels released in the serum of treated or not animals measure the effect of treatments on the osteoclasts activity whereas the those observed for mTRAP, being this an osteoclast marker, measure the amount of osteoclasts. Thus, our data suggest that GMI-1359 reduced osteoclast activity rather osteoclast number.
see Comment U32, lines 599-606
Question 13.
Fig. 8 and associated text.
Panel B. The label above the radiograph images states, "Bone scores by Liang et al." It is not clear what this means. Are these images from a different article by Liang et al? Or, is the method of scoring used in this manuscript adapted from Liang et al.?If the latter is the case, then this scoring system should be described and/or referenced in the M&M, and the text (“Bone scoring by Liang et al.”) should be removed from panel B.
Reply 13a (i). The images shown in the panel B are produced in our laboratory, So the statement bone scores by Liang et al was eliminated in this panel whereas in the text we state "Bone scores were generated accordingly to Liang et al. [62]"
Line 568. The authors states that, “bioluminescence images for the identification of osteolytic scores” are shown in panel B. How were BLI values used to develop osteolytic scores? The M&M merely describes how BLI was used to identify/quantify the presence of tumor cells. It is not clear how the presence of luciferase-positive cells was used to generate an osteolytic score. This method must be described in the M&M or a relevant reference cited.
Reply13a (ii). I'm sorry that the text is confusing. The bone scores were developed only from X-ray images accordingly to Liang et al whereas bioluminescence images were not used to generate any osteolytic score. Obviously, a lytic lesion/score may be associated with a determined quantity of tumor cells growing in the analyzed bone lesion which is, in turn, was defined by a specific bioluminescence quantity. This was not the contrary. Commonly we used the BLI to quantify the growth of PCa cells into the bone plotting BLI values versus time. To calculate BLI values as number of photons/sec/cm we used the Hamamatsu camera (Hamamatsu Photonics Italy S.R.L, Arese, Italy) with the Wasabi software. The BLI values were compared to the densitometric analysis performed on images with transformation at 16 bit and grey scale. This information is provide in MM
See comment U11, lines 246-255
Panels C-E. In reference to the graphs in panels C-E, the authors state, “We demonstrated as GMI1359 shows a higher effectiveness in the reduction in tumor growth into the bone when compared with CTCE9908 and GMI1271.” It is very difficult to assess the validity of this claim.
The single agent curves in panels C, D, E show very similar BLI values at the 21-day time point; specifically, the BLI values for all 3 of these agents at 21 days appear to be nearly identical (just under 400). In fact, the GM1271-treated group appears to have the lowest BLI value at day 14.
Thus, support for this initial claim is lacking. If the authors want to make this claim, then they need to show/report the specific average BLI values for each agent +/- standard deviations and perform statistical analysis to assess differences in efficacy. Until these are shown, the statement that GM1359 is superior cannot be supported.
Reply 13b. On the basis of the data presented here the reviewer is right.
In figure 7B we show the representative radiological appearance of bone lesions as they appear to Faxitron analysis for the identification of osteolytic scores together to bioluminescence images. X rays and bioluminescence activity was evaluated at 7, 14 and 21 days. We used different modalities of analyses for the BLI values plotted for the time. In the first analysis (figure 7 C-E) we considered the increment of BLI ( photons/sec) over time only in BLI positive tibiae in manner to exclude the negative tibiae having BLI values close to zero. On the basis of these data, the single agent curves showed very similar BLI values for GMI-1359 and CTCE-9908 at the 21-day time point whereas GM1271-treated group appears to have the lowest BLI value at day 14. So we cannot discriminate and compare the effects of treatments. In this analysis modality, indeed, we do not consider if treatments influence the phase of consolidation or bone engraftment in the metastatic site (latency). In addition, during the time of experiment new tibiae resulted positive to bioluminescence at different controls (times). For this reason, we performed a new analysis considering the BLI values for all determinations. This new procedure reduced the mean values of BLI and increased the Standard errors (SE) of BLI distribution. Table I summarized the results of these analyses. In this table we compared also the changes in Densitometric Lytic Activity measured at the same times considering, also in this case, the negative tibiae which received a score of zero. If we consider table I data, the statement that GMI1351 is more active of CTCE-9908 and GMI-1271 reaches a higher meaning when we analyze of tumor growth rates. Here dN/dT indicates the changes in the amount of BLI respect to the T0 values for time unit. The growth rates calculated for the controls were 57,9 for the time interval T7-T21 days with a value of 8,7 for the initial phase (consolidation phase T7-T14 days) and 107,1 for the second phase (exponential proliferation phase).This suggests that in the untreated animals (CTRL) the exponential phase is dominant on the consolidation phase since the tumor cells are not subjected to any drug-mediated selective pressure. The DTX administration reduced of about 3,68 time the growth rate with a value of 15,7 (T4-T21 time interval) in which the consolidation phase shows a growth rate of 2,8 (4.35 time vs CTRL) and a proliferation phase show a growth rate of 28,6 (3,74 times). This indicates that DTX interfere both with the engraftment and the growth of tumor cells in the bone. Similarly, GMI-1359 treatment goes from a growth rate value of 7,8 (T7-T14 interval; 12% less respect to CTRL and 3,9 time higher respect to DTX), to 21,4 (T14-T21 time interval; 5,1 times lower of CTRL and 38% less of DTX) with a overall value of 14,3 (4,05 times lower of CTRL and 14% less to DTX). So, GMI-1359 showed low antitumor activity when compared to DTX reducing, however, both the engraftment and the tumor growth in the bone. Next we considered the combination GMI-1359 plus DTX and we find that the calculated growth rate was 4.6 for the all considered phases. This support the concept that GMI-1359 increased the effectiveness of DTX reducing mainly the exponential growth of 4,65 times when compared to GMI-1359 alone and 6 times when compared to DTX alone. In CTCE-9908-treated animals we found that the growth rates go from a value of 10.4 (tumor engraftment phase) to 22,6 (exponential phase) with an overall value of 16,5. These data suggest that CTCE-9908 showed lower antitumor effectiveness when compared both to DTX and GMI-1359 being higher the growth rate both in the initial and exponential phases when compared to DTX and GMI-1359 single therapy. CTCE-9908 ameliorated the DTX effectiveness with an overall growth rate of 6.2 (4,1 and 8,2 for the initial and exponential phases, respectively). While the effects on engraftment phase was similar between two combinations the exponential phase was higher in the combination CTCE-9908 plus DTX when compared to GMI1359 plus DTX (1,78 times). This means that the later shows better effects on DTX sensitization. The comparison between the exponential phases of the combination CTCE-9908 plus DTX show also increased activity of about 3.5 time versus DTX and 2,75 times versus CTCE-9908 single treatments alone. GMI-1271 administration determined an overall growth rate of 25,5 (14,3 and 37,1 for initial and exponential phases, respectively) which is lower respect to control but higher respect to those observed for DTX or GMI1359 single administrations. The analyses on the combination GMI-1271 plus DTX showed an overall growth rate of 6.7 which was maintained for all phases of the tumor growth. So, GMI-1271 shows, also, DTX sensitizing effects that, however, are lower of combinations CTCE-9908 plus DTX and GMI-1359 plus DTX. We demonstrated also that the osteolysis were significantly reduced after co-treatment of GMI-1359, CTCE-9908 and GMI-1271 with DTX and that GMI-1359 effects were more marked when compared to GMI-1271 and very similar when compared to those observed for CTCE-9908.See comment U34, lines 651-704
Lines 574 – 578. The authors conclude that GMI1359 was superior or comparable to the other 2 agents with respect to osteolysis, and then refer to Table 1 for presentation of lytic units and Fig 8G for mTRAP assay results.
First, Table 1 was not included in this manuscript.
Reply 13c. I'm sorry for this inconvenience. I don't understand how he managed not to load it into the web. Table was now provided.
Secondly, there were no clear distinctions among the DTX+agent combination treatments in the mTRAP assay, and no indications of statistical differences to support the authors claim are presented.
Reply 13c (ii). Statistical was provided in the new version of report.
Lines 579-581. The authors claim that treatment with the compounds delayed the onset of cachexia and weight loss that is typically associated with bone metastasis. Please include these findings in the manuscript.
Reply 13d. this was made. Figure 7H and legend figure 7 (lines 727)
Question 14.
Fig. 9 and associated text.
Panel B. The authors claim that Fig. 9B shows that osteoblast proliferation was increased by exposure to CM from PC3, 22rv1 and C42-B cells (Lines 627-629). It is not clear that the data in Fig 9B supports this statement. The Y-axis is labelled “Osteoblast proliferation vs control,” which suggests that the values in the graph are expressed as a fold-change vs control or normalized to control. If that's the case, then the PC-3 CM, with a value of 0.8, induced less proliferation than in control, while the other CMs induced proliferation (1.2, 1.6). If this is correct then the statement must be revised to reflect the data. If this is incorrect or there is a different way to interpret this data, then the authors must be clearer about how the endpoint is expressed.
Reply 14a. Data from the figure 9B was erroneously shown as fold-changes vs control or normalized to control. This data represents, instead, the absolute values at optical density evaluated at 595 nm. Y axis was re-labeled in the new version of report.
Panel A and B. In panel A, the CM of 5 different prostate cancer cell lines were tested for effects on osteoblast proliferation. It is not obvious that, based on the photographs of the CV-stained cells, that the PC3, 22Rv1 and C4-2B CMs induced more proliferation than the VCaP and LNCaP CM. Were OD values for solubilized CV measured for all of the cell lines?
If so, these should be shown to demonstrate that PC3, 22rv1 and C42-B CM did indeed induce proliferation.
Reply 14b. We added a graph below CV images to show OD values for all cells.
Panel C. Regarding Fig 9C, the authors state only, "We observed that the more osteogenic cells induced most osteogenic activity (ALP, Von Kossa and Alazarin red stains)." This conclusion appears to be based solely on the photographs of the stained cells in panel C. It is not at all clear that CM from these cell lines increased staining relative to the other cell lines. In fact, the ALP activity data in panel D appear to show that PC3 CM reduced ALP activity relative to control.
Reply 14c.
In the new version of paper, this figure is now figure 8.
It was widely demonstrated that prostate cancer cells secreted osteogenic factors (CM) whereas PC3 cells and their cell derivatives show low activity for osteoblasts when their CMs were added to osteoblast's cultures. Osteogenic activity seems to be dependent to the ability of PCA cells to origin osteosclerosis rather osteolysis. PC3 cells are, indeed, more osteolytic rather osteosclerotic. Our statement that " more osteogenic cells induced most osteogenic activity" was not based only on photographs from panel 9C. Data on ALP activity (which is an index for osteoblast proliferation) suggests that C4-2B is the more osteogenic cell line when compared to PC3 and the ALP activity data in panel D appear to show that PC3 CM reduced ALP activity relative to control. PC3 are less osteogenic of C4-2B. Alazarin red solubilization show values of Optical Density at 405 nm higher in VCaP and LnCaP osteogenic cells. In addition we show that 22rv1 cells secrete osteogenic factors that bring it closer to the C4-2B and distance the PC3 in agreement with their capacity to produce mixed osteosclerosis/osteolytic lesions. C4-2B, VCaP and LnCaP cells represent the more osteogenic cells as widely demonstrated in literature.
see comment U38, lines 771-779
Panel E. It is suggested to arrange the photographs for each cell type so that the treatments are presented in the same order in both panels, and perhaps even in the same order as in the bar graph in panel F.
Reply 14d. This was made.
Question 15.
Fig. 10. a. Legend. There is no description of panel E in the legend.
Reply 15a. This was fixed in the legend of new figure 9. See comment U39, lines 801-803.
Panel B. Some of the asterisks in the graph are misaligned.
Reply 15b This was fixed
.
Question 16.
Line 684. It is not clear what this means: “if PC3DTXR”
Reply 16. this was fixed
Question 17.
Lines 687 and 689. References #53 and 54 are cited to support statements about the CXCL12/CXCR4 axis, but, based on their titles, they appear to be about E-selectin.
Reply 17. This was a mistake of different revisions which received this report. This is now fixed.
Question 18.
Lines 698-699. “It is not a simple case that DTX is administered to patients with metastatic and castration resistant disease (mCRPC).” Please clarify what this sentence means.
Reply 18. This was fixed
Question 19.
Lines 720 – 723. “The sensitization versus DTX played by GMI1271, a specific E-selectin antagonist, was still lower when compared to GMI1359 and CTCE9908 suggesting that E-selectin antagonism could increase the efficacy of a CXCR4 antagonism.” The logic of this statement is unclear. As written, this sentence appears to state that the E-selectin antagonist had a lower ability to sensitize to DTX, and thus combining both would increase efficacy. How? Please clarify.
Reply 19. This statement was effectively confusing. So, we changed this statement as:
GMI-1271, a specific E-selectin antagonist, showed a low DTX sensitization when compared to those observed for CTCE-9909, a specific CXCR4 antagonist. GMI-1359 showed the highest DTX sensitization supporting the idea that combining CXCR4 and E-selectin may have a greater overall effects.
Questipon 20
Lines 761. This line contains an abrupt, odd shift to the first person (I.e. the use of “I”). Please revise.
Reply 20.
This was fixed
Question 21.
Lines 765 – 771. In this passage, the authors state that overall survival was increased by all three of the agents tested, that GMI1359 (dual inhibitor) was statistically more effective than GMI1271 (E-selectin antagonist), but not different from CTCE-9908 (CXCR4 antagonist). First, while Kaplan-Meier plots were used to analyze time to progression and incidence of bone mets, no Kaplan-Meier data on overall survival was presented. Second, if the dual inhibitor was indeed no better than the CXCR4 inhibitor in prolonging overall survival, then this would not support the superiority of the dual inhibitor.
Reply 21. I'm sorry for the lack of two panels relative to overall survival. These were not included after previous revisions but in the new figure 7.
We changed text of this section as follows:
In figure 6K we show the rate of mortality of untreated (CTRL) or treated animals with bone and visceral metastases. Commonly, mice euthanasia is necessary at distress signs evidence and came starting from 40-60 days (mean ± SD, 52.0 ± 1,6 days) in the untreated animals. Overall survival values grow until 58,7 ± 2,5 days in CTCE-9908 treated animals (P=0,0002 vs CTRL). GMI-1271 treated animals were euthanized after 59,9 ± 2,8 days (P=0,0134 vs CTRL, P=0.1744, not significant, NS, vs CTCE-9908) whereas GMI-1359 treated animals were euthanized after 62,9 ± 2,8 days (P=0.0002 vs CTRL, P=0.0033 vs CTCE-9908 and P=0,005 vs GMI-1271). In figure 6L we show the Kaplan-Meier curves with relative hazard ratio values and statistics. Although the values of HR vs controls are higher in the GMI-1359 treated animals (HR=4,10) when compared to CTCE-9908 (HR=2,29) the comparison between these two treatments reached a not statistically significant value (HR= 1,78).

Reviewer 3 Report
This article describes the use of GMI1359 (a CXCR4 and E-Selectin inhibitor) against prostate cells and its anti-bone metastatic effect. It has been studied in combination with Docetaxel, a traditional prostate cancer agent which suffers from problems of resistance.
The study is very thorough examining many different aspects and variables both in vitro and in vivo. The results are very promising.
Just a few minor points
There are a lot of acronyms and abbreviations. Maybe these should be listed at the beginning or end of the paper.
There are several mistakes in the English language used. The manuscript should be checked and corrected.
Author Response
We thank this reviewer to consider the report worthy of being published in the Journal "cells"
Minor points were fixed accordingly
many thanks again
Claudio
Round 2
Reviewer 2 Report
The authors have made a strong effort to address the comments/request of this reviewer resulting in an improved manuscript, particularly as it relates to the addition of missing statistical analyses, missing table and figure panels, more complete descriptions and explanations of methods and results, and corrections of discrepancies between data shown and descriptions of results. However, some issues still remain both from the previous version of the manuscript and related to the new information presented. These issues are largely of the same type that plagued the previous version.
Page 5, Line 224, Materials and Methods, Evaluation of treatment response in vivo:
In response to this reviewer’s questions about the appropriateness of the methods described for calculating combination index (CI) and about missing CI data, the authors responded as follows:
(a) They added an alternative method of CI calculation which they described in the M&M.
(b) They cited a paper to support this method; specifically reference #51 (Dekker et al).
(c) They added CI data to the manuscript.
Unfortunately, the reference they cited (#51) does NOT appear to describe this or any method of CI calculation. Also, the formula for CI that they provide includes the letters “a” and “b,” but they do not define what these letters stand for. Presumably, a reader could look at the cited paper to get this information, but, again, the cited paper is apparently not the correct one.
Discrepancies remain among the lists of doses and routes for the administration of the inhibitors in the in vivo studies.
In the previous review, the reviewer identified discrepancies in the doses and routes listed in different parts of the manuscript and requested the rationale for dose selection. Unfortunately, in the newest version of the manuscript, the doses and routes of CTCE9908 and GMI1271 listed in the Materials & Methods, Figure panels, and references they cited to support the dose selection still do not match, despite the fact that the authors stated that they checked the recommended doses and confirmed the doses used. The table below summarizes the discrepancies which are highlighted (yellow for dose, green for route).
|
Materials & Methods (p6, line 260) |
Figure panels Fig 4A, 5A, 6A, 7A |
References cited |
GMI-1359 |
40 mg/kg, bid, IP |
40 mg/kg, bid, IP |
40 mg/kg, IP In Ref #54 |
CTCE9908 |
50 mg/kg/day, IP |
60 mg/kg/day, IP |
25 mg/kg, SC In Ref #55 |
GMI1271 |
5 mg/kg/day, [no route indicated] |
5 mg/kg/day, SC |
40 mg/kg, QD, IP In Ref #56 |
Please verify what doses and routes were actually used in the reported experiments and revise the manuscript accordingly. If the doses and routes used differ from those reported previously, please provide a rationale for the change. This is important because the different inhibitors are being compared with respect to efficacy. For example, the authors conclude that, in general, GMI1271 is less effective than the other 2 inhibitors, but the reported dose (5 mg/kg) is 8 fold-lower than the dose reported in the cited article.
Page 13, Line 507:
“figure 5B-D” should be changed to “Figure 5D-F.”
Page 13, Lines ~511-519:
The authors are trying very hard to argue for superiority of the GMI1359+DTX combination over the CTCE9908+DTX combination, despite a clear lack of statistical significance (P=0.4476, Fig. S1C). They argue that there is an “upward trend in DTX sensitization in GMI-1359 when compared to CTCE-9908.” It is not clear what is meant by “upward trend.” Based on the author’s response, it appears they are arguing that 24 days to maximal % progression for GMI1359 is markedly better than the 22 days for CTCE9908. This appears to be “hand waving” to support their hypothesis which is not supported under the experiment conditions tested in this specific experiment. The authors need to explain in the manuscript in more detail what is meant by “upward trend.”
Page 25, Line 916:
Ref #48 used to support the contention that a marker for CXCR4 inhibition is reduction in CXCR4 expression. I wanted to confirm that because it is interesting to me that an enzyme inhibitor would also be a reducer of enzyme expression. However, the reference cited is inaccurate. Ref #48 is a paper on fluorescence. Please correct this error. It is not fair to readers of this paper to expect them to search the reference list for the appropriate article to obtain information of interest.
Page 14, line 545:
In the previous review, the reviewer identified discrepancies between the data shown in a bar graph (now Fig. S2B) and the related text descriptions of that data. According to the authors’ response, they corrected this statement. BUT, this statement is STILL inaccurate. The newly revised statement is “tibiae (80%), femurs (40%) or both (85%).” BUT the figure shows that the % for femurs is ~20%, not 40%, and the % for both tibia+femur is ~75%, not 85%.
Page 14, line 547:
There are two inaccurate call-outs to figures in this line. This line should read, “…5% (Figure S1B). In Figure S2C,…”
Figure S2A:
In the previous review, the reviewer identified the word “REF” in the title of the panel (“PCb2, REF”). In their response, the authors stated that this was fixed…it was not fixed.
Figure S2C:
In the previous review, the reviewer requested that scale bars be added to the histologic images that are now in figure S2C. In their response, the authors stated that this was done…it was not done.
Figure S2C:
In the previous review, the reviewer asked that information on the human tissue samples be added to the Materials and Methods and the legend. This info was added only to the legend. Please include this info in the Materials and Methods.
Figure 6B:
Figure 6B, the legend for Figure 6B and page 15, line 564 state the days on which Faxitron radiographs were taken; but none of the 3 descriptions match! The discrepancies are summarized below:
Figure 6B: Days 32, 40, 46
Fig, 6B legend: Days 35, 42, 50
Line 564: Days 32, 46, 53
Please verify when this data was acquired and correct this information consistently.
Figure 6A:
In the previous review, the reviewer identified a discrepancy between the diagram in Fig 6A and the related text in the Materials and Methods regarding the timing of cell injection and treatment start. Accordingly, the authors clarified the diagram; i.e. they confirm that treatments started 2 days after injections of cancer cells; BUT they failed to correct the associated text (line 231-232).
Figure 6:
The legend needs to be re-written. The figure panels and their respective descriptions in the legend do not match.
Figure 7H:
This new figure shows data on the effects of the various treatments on cachexia. Important information is missing regarding this data set. What were the criteria for defining whether an animal was cachectic or non-cachexia? Was it a specific % weight loss vs starting weight? A specific body condition score? Also, at what time point was the assessment of cachexia performed for this analysis? Was this at end of study? It is best to include this information in a relevant section of the Materials & Methods. Also, were any statistical analyses performed for this data?
Author Response
I reiterate with this cover letter that the reviewer is allowing us to obtain a truly improved (as already considered by the auditor himself) and solid report. I always hope to find this type of colleagues in other reviewers that I will meet in the future in my submissions. Thank you so much for your excellent work.
Question Q1. (see in the text for comment U2)
Page 5, Line 224, Materials and Methods, Evaluation of treatment response in vivo:.........
In response to the questions about the appropriateness of the methods described for calculating combination index (CI) and about missing CI data, we did not added "an alternative method" of CI calculation which we described in the M&M but calculated CIs considering the HR values. In the first version we used the values of inhibition percentage performed on tumor weights calculated at the end of experiments. Reading again the present work with a cool mind and following the comments of this reviewer, it seemed more correct to compare the temporal trends (kaplan meier curves). So, I asked a question about research gate on "how to calculate the CI" using Kaplan Meier curves (see for example the webpage: https://www.researchgate.net/post/How_to_calculate_the_synergy_Index_when_the_measure_of_risk_is_hazard_ratio). Some methods have been proposed to me including the use of HRs values (in addition to those of relative risk or ORs) and I had followed the publications by Dekker and colleagues. We have read 5 of these with various experimental examples but, as you can see, we mentioned in the report the wrong one. Dekker and colleague published various reports on Kidney researches. It is necessary consider that several methods have been proposed in literature i.e. Linda Kalilani and Julius Atashili in the article entitled " Measuring additive interaction using odds ratios. Epidemiologic Perspectives & Innovations", December 2006, 3:5. doi:10.1186/1742-5573-3-5.
but there are others to mention. In addition, in the formula appear the letters a and b that this reviewer did not understand what they represent. These refer to drugs "1" and "2" when we used "1 and 2" drugs in combination. So the CI is the sum of single HR for the drug 1 and drug 2 divided the HR value obtained experimentally from the combination of both drugs. If the HR calculated for drug 1 + drug 2 is higher of the HR obtained from the combination drug 1 + drug 2 we have a CI>1.0 (antagonism); if the sum value is lower of the HR obtained from combination we have a CI<1.0 (additivity/synergism dependently to the amount of differences). See comment U1.
Q2. (see the text for comments U1, U3, U4, U5, U14, U6, U20, U21)
Discrepancies remain among the lists of doses and routes for the administration of the inhibitors in the in vivo studies.
So I gathered the technical staff who did the administrations as well as I required to Glycomimetics to get clarifications. We looked at the experiment books and in the abstract/Poster presented by us at ASH, AACR and EACR meetings and concluded that:
GMI1359 was administered 40 mg/Kg BID for 14 consecutive days as indicated in the abstracts: 202-ASH meeting 2016 abstract 2826 by Fogler et al and 106-AACR meeting 2015 by Gravina et al or in the Abstract 428: Giovanni L. Gravina, Andrea Mancini, Alessandro Colapietro, Simona D. Monache, Adriano Angelucci, Alessia Calgani, William E. Fogler, John L. Magnani, and Claudio Festuccia. Dual E-selectin and CXCR4 inhibition reduces tumor growth and increases the sensitivity to docetaxel in experimental bone metastases of prostate cancer. DOI: 10.1158/1538-7445. AM2015-428 Published August 2015. CTCE-9908 was administered at 25 mg/Kg/every days for 28 days IP (as described in the ref 55). GMI1271 was administered 40 mg/Kg/BID for 10 consecutive days as indicated in the 201-ASH meeting 2017 abstract 894. This reference substitutes reference 56 and in the EACR meeting 2015: Dual CXCR4 and e-selectin pharmacogical inhibition reduces tumour growth and increases the sensitivity to docetaxel in experimental bone metastases of prostate cancer.By G.L. Gravina, A. Mancini, A. Colapietro, S. Delle Monache, A. Angelucci, A. Calgani, P. Sanità, W.E. Fogler, L. Magnani, C. Festuccia (Italy) and in the abstract 428 AACR meeting 2015 (see above). This information was fixed in the text and figures/legends.
Q3. (see comments U16, U28, U33):
Page 13, Line 507:
“figure 5B-D” should be changed to “Figure 5D-F.”
this was made.
Q4. (see comment U17):
Page 13, Lines ~511-519: The authors are trying very hard to argue for superiority of the GMI1359+DTX combination over the CTCE9908+DTX combination, despite a clear lack of statistical significance (P=0.4476, Fig. S1C).............
Reply: we modified this statement as follows:
"Although data of tumor weight (figure S1A) and TTP (figure S1B) indicate a statistically better efficacy of GMI1359 versus CTCE-9908 in the increased effects of DTX in the 22rv1 xenograft (P=0.0187 and P=0.0377, respectively), data obtained from Kaplan-Meier analyses (figure S1C) indicates that the sensitizing effects were similar. Considering that GMI1359 was administered for 14 days while CTCE-9908 for 28 days, the effects of GMI1359 could be better with a different dose administration. These considerations should be verified with another focused future experiments"
ONLY for the reviewer:
This consideration could be valid considering that recent and not published data indicate that GMi1359 could be administered for all 28 days of treatments without evident side effects in the nude mice. This consideration was not inserted in the manuscript.
Q5. (see comments U13 and U34)
Page 25, Line 916:
Ref #48 used to support the contention that a marker for CXCR4 inhibition is reduction in CXCR4 expression....................
We modified accordingly
Q6. (see commentU18)
Page 14, line 545:
In the previous review, the reviewer identified discrepancies between the data shown in a bar graph (now Fig. S2B) and the related text descriptions of that data. According to the authors’ response, they corrected this statement. BUT, this statement is STILL inaccurate..........
This was fixed.
Q7. (see comments U18 and U28
Page 14, line 547:
There are two inaccurate call-outs to figures in this line. This line should read, “…5% (Figure S1B). In Figure S2C,…”
This was fixed
Q8. (see new figure S2A)
Figure S2A:
In the previous review, the reviewer identified the word “REF” in the title of the panel (“PCb2, REF”). In their response, the authors stated that this was fixed…it was not fixed.
I'm sorry this was fixed in this version.
Q9 (see comment U35)
Figure S2C:
In the previous review, the reviewer requested that scale bars be added to the histologic images that are now in figure S2C. In their response, the authors stated that this was done…it was not done.
This was fixed see comment 35.
Q10 see comment U8
Figure S2C:
In the previous review, the reviewer asked that information on the human tissue samples be added to the Materials and Methods and the legend. This info was added only to the legend. Please include this info in the Materials and Methods.
This was fixed see comment U8
Q11. See comments U22, U24
Figure 6B
Figure 6B, the legend for Figure 6B and page 15, line 564 state the days on which Faxitron radiographs were taken; but none of the 3 descriptions match! The discrepancies are summarized below.....................................................................
This was fixed
Q12.
Figure 6A:
In the previous review, the reviewer identified a discrepancy between the diagram in Fig 6A and the related text in the Materials and Methods regarding the timing of cell injection and treatment start. Accordingly, the authors clarified the diagram; i.e. they confirm that treatments started 2 days after injections of cancer cells; BUT they failed to correct the associated text (line 231-232).
I hope that this was completely fixed. I can no longer read the report again
Q13. See comment 24, U30, U33
Figure 6:
The legend needs to be re-written. The figure panels and their respective descriptions in the legend do not match.
This was made.
Q14.
Figure 7H: See comments U7, U30, U32, U33
This new figure shows data on the effects of the various treatments on cachexia. Important information is missing regarding this data set. What were the criteria for defining whether an animal was cachectic or non-cachexia?. Was it a specific % weight loss vs starting weight?
This was provide at comment U7 in M&M
A specific body condition score?
Not performed since the relative low number of animals.
Also, at what time point was the assessment of cachexia performed for this analysis? Was this at end of study? It is best to include this information in a relevant section of the Materials & Methods. Also, were any statistical analyses performed for this data?
The time for cachecsia detection was the end of the experience. We have noticed this phenomenon in the time but the dispersion of data not allowed to provide a clear time course data. For the statistical analyses we used a chi-square test with a correction through Bonferroni. This information was added in M&M and in the results

This manuscript is a resubmission of an earlier submission. The following is a list of the peer review reports and author responses from that submission.
Round 1
Reviewer 1 Report
The manuscript reported the bone metastasis inhibition effect of GMI1359 in prostate cancer. However, there are many concerns in the data presentation and discussion.
Materials and Methods:
Please clarify and indicate the (commercial) sources of the prostate cancer cell lines, the DTX resistant derivative lines, and the HBMEC, HUVEC cells. Confusingly, data from experiments using HBMEC and/or HUVEC cells were not able to be found in this manuscript, please clarify.
Please indicate the cell numbers injected in the intracardiac and intratibial tumor models.
Results:
General concerns for all the figures:
Usually * p<0.05, ** p<0.005 (or p<0.01), and *** p<0.0001 (or p<0.001) are used for indicating the significance. Please clarify and indicate the specific P values.
Please mark the * in the respective graph of the figures.
Need to indicate the “n=?” for each experiment group with means or average values
For data such as western blots, the authors need to clarify how many times the same experiments were repeated, were similar or same results obtained?
A subtitle is needed before Line 165.
Figure 1. How was the fluorescence index obtained/calculated? Indicate here or write a paragraph in the Materials and Methods. Line 172-174: the author claimed “significantly higher”…. Again, how was it calculated? Were the numbers mean or average value?
Please explain why choose 22Rv1 and PC3 for further studies, but not the other cell lines?
Figure 1B and 1C. what are the sources of CAF and OB for generating the conditioned medium? How were these cells cultured and How were the conditioned medium generated?
Figure 1C. CTCE9908 and GMI1271 should be included in the experiment to compare with the effect of GMI1359.
Line 203. (D) should be (C)
Line194-195: that data need to be shown.
Figure 2. As the authors cited in ref. 34, DTX increases CXCR4 had been reported in prostate cancer cells. The novelty and impact of this manuscript should be significantly enhanced if the authors could test whether the increased CXCR4 could be rescued by addition of the three drugs. Experiments testing the involvement of E-Selectin, its ligand (noticed the authors mentioned in Line 224-230), or its downstream signaling components in either the tumor microenvironment or PCa cancer cells are needed to explain GM1359, the dual CXCR4/E-selectin antagonist, had better effect than the individual antagonist.
Line 239 the authors wrote “… suggesting additive and synergistic interaction..” Could this be tested by further calculation based on the IC50?
Figure 4 and Figure 5 showed the data with 22Rv1 in subcutaneous tumor models, what’s the rational for the study? Please explain in Result.
The values of the graphs were exactly the same to the respected groups between Figure 4 and figure 5, as well as among figure 5. Did the authors use the same data generated multiple graphs? Please clarify. If some of the graph were indeed from the same data, it is Important to claim it and combine in one graph.
Figure 4A. Line 281. So the drug started one week after the tumor cell grafted. A diagram of treatment schedule will be helpful for readers. Please clarify at the time of treatment stated, what are the range of the tumor volumes or what are the range the calculated tumor weights? The author claimed the tumor bearing mice were randomly assigned to different treatment groups. However, it can’t be randomly unless all the tumors were at similar weights. The mice without tumors or with bigger tumors out of the range should be taken out of the experiments.
For the studies with subcutaneous tumors, a picture of the harvested tumors, or the mice bearing the tumors should be included.
Line 302. A claim title is needed prior here.
Figure 6. For the intracardiac injections, full body X-ray images should be included, because PC3 cells metastasize to many sites of the bones other than the right tibia. The progression of bone lesions using X-ray should show all the groups, not just the control and GM1359. The tumor-induced bone lesions need to be confirmed with HE staining to clearly show the tumor cells in the bones. The tumor volume versus the total or bone volume should be analyzed, especially the authors claimed that the effects in prostate cancer cell intraosseous growth. Without these data, one can only claim changes of tumor-induced bone lesions, but not the growth of tumor cells.
Figure 7. Treatment schedule indicated the treatment started at day 7, but in the text Line 354-355, the authors wrote that the treatments were started at day 3. Please clarify. Again, HE staining should be shown to confirm the growth of tumor cells in the tibiae, and bone morphometry analyses to indicate the changes of tumor growth in the bones.
Line 366 and Line 369, the authors made some claims with “data not shown”. However, these data are interesting and important, should be shown, at least as supplemental figures.
Discussion. Suggest revise. Currently, it is descriptive and repetitive for their results. Particularly to discuss their results with studies from previous literature and the limitations of this current studies.
Author Response
Reviewer # 1
The manuscript reported the bone metastasis inhibition effect of GMI1359 in prostate cancer. However, there are many concerns in the data presentation and discussion.
Materials and Methods:
Question 1a. Please clarify and indicate the (commercial) sources of the prostate cancer cell lines, the DTX resistant derivative lines, and the HBMEC, HUVEC cells.
I'm sorry for this forgetfulness. In addition we modified the report accodingli inserting that:
(1a) Prostate cancer cell lines were obtained from:
- ATCC (DU145 and PC3)
- Leibniz Institute DSMZ-German Collection of Microorganisms and Cell Cultures , Braunschweig, Germany (22rv1)
information was reported at lines 137-139
Question 1b .Confusingly, data from experiments using HBMEC and/or HUVEC cells were not able to be found in this manuscript, please clarify.
(1b) I agree with the reviewer who finds confusingly data from experiments using HBMEC and/or HUVEC cells. These two types of cells are not, indeed, used in this report. In reality we have used the murine bone marrow stromal (BMS) cells and osteoblasts. In all likelihood, there has been confusion in the preparation of the MM with processes of cut and paste from the protocol computer. So, HBMEC and/or HUVEC cells were deleted in the text while murine BMS cells and osteoblasts were added. In the development of the reply to reviewers it has been also necessary to consider also the effects on a third model for bone derived bone marrow cells as the osteoclast precursors with macrophage phenotype, RAW264.7 cells. MM section was modified accordingly.
Lines 156-176
Question 2. Please indicate the cell numbers injected in the intracardiac and intratibial tumor models.
I'm sorry for this forgetfulness. Now we present this information in the new version of report in the material methods section: "we injected 1 × 105 tumor cells in 0.1 ml of PBS for the intracardiac method and 1 × 105 cells/10 µl PBS for intratibial method."
This was added at lines 239 and 246-248.
Results:
Question 3. General concerns for all the figures: Usually * p<0.05, ** p<0.005 (or p<0.01), and *** p<0.0001 (or p<0.001) are used for indicating the significance. Please clarify and indicate the specific P values. Please mark the * in the respective graph of the figures.
We thank reviewer for her/his comment. Although the significance is frequently reported as above stated by reviewer, especially in translational manuscripts, this way to report the alpha error is not statistically sounds. The type I error rate or significance level is often sets to 0.05 (5%) or 0.01 (1%), implying that it is acceptable to have a 5% or 1% probability of incorrectly rejecting the null hypothesis. So once a researcher sets this threshold is statistically improper to use different thresholds such as p<0.05, (5%) p<0.005 (0,5%) p<0.01 (1%) and *** p<0.0001 (0,01%) and so on. Once the alpha error is under the pre-specified threshold the significance difference is simply obtained. So P values at least <0.05 were considered statistically significant. I added a sole asterisk to indicate results significant vs respective controls. Fort the analyses of Kaplan Meyer curves P values are considered in full (and this was appropriate) Figures 1- 3 were changed and legends were modified accordingly.
MM at lines 285-286 and figure legends 1-3
Question 4. Need to indicate the “n=?” for each experiment group with means or average values
I'm sorry for the lacking of information . We indicated the number of replicates for each group in the legend of figures 1-3 as follows: Graphical data are representative of three similar results performed in triplicate (lines 355-357, 366-367 and 419-420).
Question 5. For data such as western blots, the authors need to clarify how many times the same experiments were repeated, were similar or same results obtained?
Also in this case we indicated in the figure legends that : Western blots images are representative of three different gels/experiments and lanes were charged with 40 mg of proteins as indicated at lines 365-366
Question 6, A subtitle is needed before Line 165:
we add the follow subtitle: CXCR4 and HECA-452 immune-reactivity in prostate cancer cells.
Line 165 now is line 294
Question 7. Figure 1. How was the fluorescence index obtained/calculated? Indicate here or write a paragraph in the Materials and Methods. Line 172-174: the author claimed “significantly higher”…. Again, how was it calculated? Were the numbers mean or average value?
We thank reviewer for this her/his comment. The reply to this question allowed us, also, to note the absence of the FACS analysis method in MM section and secondly the inaccuracy or poor definition of the parameter used to compare the expression of the various cell lines and third that the mean values of CXCR4 for bone or non bone metastatic cell lines was wrongly reversed.
So, We added in MM section the method used for FACS analysis. Lines 167-184 OK
We defined, also, the Mean Fluorescence Intensity (MFI) as:
Basically this parameter is the average intensity of fluorescence of the sample in exam (considering all cells of interest) divided by the fluorescence intensity of the relative control (a non relevant immunoglobulin labeled with similar fluorochrome as the staining antibody). The MFI was calculated for each samples. Practically MFI is an index which measure the shift of the fluorescence peak with respect to the control: i.e. a value of MFI ≤1 indicates that the cell population is scarcely o is negative for the examined marker since the two fluorescence plots are similar. An MFI = 2 indicates that the fluorescence intensity of the given marker is 2 times greater than the threshold/negative value. We added this information in MM at lines 174-184 In addition values indicated at the old line 172-174 are the MEAN VALUEs ± RELATIVE STANDARD DEVIATIONS of single experimental conditions obtained analyzing the average values of MFIs calculated in three separate analyzes at FACS.
and finally, we add two graphical representation of comparisons on CXCR4 and HECA-452 levels between bone or non bone derived PCa cells (Figure 1B). Figure legend was changed accordingly (see also reply to question 11). Text lines 349-357 OK
Question 8. Please explain why choose 22Rv1 and PC3 for further studies, but not the other cell lines?
GMI1359 (dual CXCR4/E-selectin) has been compared with the single CXCR4 (CTCE-9908) and E-selectin (GMI-1271) antagonists. At first, we analyzed the expression of CXCR4 and the tetrasaccharide carbohydrate E-selectin ligand (sialyl Lewis x also known as CD15s) recognized by HECA-452 antibody in PCa cell lines derived from bone (C4-2B, VCaP, PC3 and PCb2) brain (DU145), lymph nodal (LnCaP) metastases and primary tumor (22rv1) grown in vitro. Next we analyzed the modulation of: (i) CXCR4 and HECA-452 immune-reactivity in PCa cells after administration of conditioned media (CM) harvested from murine cancer associated fibroblasts (mCAF) and bone derived cells; (ii) IC50 values calculated for Docetaxel (DTX) after administration of exogenous SDF1 and CM from mCAF and bone derived cells and (iii) the chemo-sensitizing effects of GMI1359, CTCE9908 and GMI1271 versus DTX in PC3, DU145, 22rv1 and C4-2B cell lines as well as in your DTX resistant derivatives. For in vivo studies we focalized our attention on 22rv1 (used in subcutaneous xenograft model) and PC3/PCb2 (used in bone metastatic models) cell lines.
Both cell lines are representative of castration resistant (CRPC) models able to be very aggressive in vivo but with differences in their tropism for the bone. PC3 cells line and its cell derivative, PCb2, are highly bone-tropic and able to colonize and grow into the bone when injected by intra-cardiac way. Differently, 22rv1 cell line shows limited bone tropism but high capacity to growth locally (aggressive/over proliferating model) with elevated tumor-mediated angiogenesis and inflammation. In addition, we have chosen to use the osteolytic bone metastatic model because this pathological appearance represents the more clinically relevant skeletal event which is associated to bone fractures that a clinician must counteract to improve the patient's quality of life. Human bone metastases from PCa are, however, mixed lesions showing osteolytic and ostesclerotic/osteoblastic areas present in the same tissue sample. In addition, it has been demonstrated thatthe bone lysis take place to osteosclerosis in the most advanced metastatic stages on the basis of changes in the equilibrium between osteogenic and lytic factors. So, PC3 / PCb2 are the only prostatic models that exclusively give osteolysis. The C4-2B [29] and VCaP [30] cell lines have been excluded from our in vivo study since, when injected directly into the tibiae, these cells induce exclusively osteo-inductive bone lesions and, therefore, are representative of a clinically less relevant skeletal event. In addition C4-2B and VCaP cell lines were not able to colonize the bones when injected by intracardiac way. The 22RV1 cells were able to induce mixed osteosclerotic/osteolytic lesions resulting a suitable murine model which is more closed to human situation. These cells are, however, unable to give bone metastases in mice when injected by intra-cardiac way and, moreover, show also a low rate of bone engraftment [31] when injected in the tibia. Nevertheless 22rv1 cells are able to modify massively tumor microenvironment increasing strongly angiogenesis and inflammation. Angiogenesis and inflammation are, as mentioned above, two pathogenetic and progression tumor mechanisms having CXCR4 and E-selectin as major players. Both receptors, indeed, modulate the recruitment and differentiation of endothelial precursors and circulating monocytes [32], sustain Tumor Associated Macrophage (TAM)-dependent inflammation, induce chemotaxis of human endothelial cells [33] in vitro and modulate angiogenesis and vasculogenesis in vivo [34]. The 22rv1 subcutaneous xenograft model was chosen also why despite being locally aggressive it maintained very low (or absent) metastatic potency of cells. In this manner we maintained separated tumor growth and bone metastatic potential.
These statements were added in the introduction at lines 76-117) and in discussion at line 646-661.
In addition, the administration of bone derived conditioned media derived from murine Bone stromal cells, osteoblasts or murine RAW (osteoclast precursor model) induced CXCR4 expression in PC3 as observed in the new figure 1E. The levels of induction of CXCR4 expression in 22rv1 cells were lower following the administration of CMs derived from BMS, OB and RAW cells confirming that 22rv1 cells possessed lower bone tropism when compared to PC3 cells and this would partially explain the absence of tumour take rate following intra‐cardiac inoculation as well as the lower tumour take rate following intra‐tibial inoculation. Murine CAF derived conditioned media were able to trigger a good induction of CXCR4 in 22rv1 suggesting that CXCR4 antagonists could be a role on the reduction of the growth of a non metastatic CRPCa model. This information was added in figure 1 C, D and in the text at lines 308-329.
HECA-452 levels were similar in PC3 and 22rv1 cell lines. HECA-452 levels were significantly increased in the 22rv1 cells after administration of conditioned media derived from mCAF. Differently the induction of HECA-452 in PC3 cells was minimal. When PC3 and 22rv1 cells were triggered with bone derived conditioned media derived from murine Bone stromal cells, osteoblasts or murine RAW (osteoclast precursor model) we observed a higher HECA-452 induction in PC3. This is information are inserted in the figure 1 E, F and in the text at lines 317-329
So, PC3 and PCb2 cells were chosen to test the effects of the bone metastatic microenvironment whereas 22rv1 cells were tested in a non metastatic model (subcutaneous xenograft).
Question 9. Figure 1B and 1C. what are the sources of CAF and OB for generating the conditioned medium? How were these cells cultured and How were the conditioned medium generated?
I'm sorry for the lacking of information on these two cell models. CAF were isolated following published procedures [37] whereas MC3T3-E1 cell line was used as model of murine osteoblast‐like cells (OB). Differentiative status of these cells was monitored by cytochemical analyses of ALP activity using reagents and protocols from the Sigma-Aldrich kit 104-LS. The methods were add in the MM at lines 146-166. OK
we inserted also the following reference as ref 37.
37. Geary LA1, Nash KA, Adisetiyo H, Liang M, Liao CP, Jeong JH, Zandi E, Roy-Burman P. CAF-secreted annexin A1 induces prostate cancer cells to gain stem cell-like features. Mol Cancer Res. 2014;12:607-21. doi: 10.1158/1541-7786.MCR-13-0469.
Question 10. Figure 1C. CTCE9908 and GMI1271 should be included in the experiment to compare with the effect of GMI1359.
We thank reviewer for this her/his comment.
However, the reply to this question allowed us to realize that to verify if the individual compounds (GMI1359, GMI1271 and CTCE9908) can modify the expression of the respective targets may have little meaning except to verify if these components act on their targets.
It has been demonstrated, indeed, that Plerixafor treated cells are able to increase the internalization, degradation and recycling in the membrane of CXCR4 and this was in agreement with our data indicating that GMI1359 was also able in untreated or treated cells to reduces CXCR4 expression.
However, our purpose was to verify if tumor micro-environments could to influence the CXCR4 and HECA-452 expression in PCa cells. This phenomenon could be due to the production of soluble factors i.e. SDF1a, TGFb1 etc..., able to modulate the expression of CXCR4 or HECA-452. Soluble factor are not targets of GMI1359, GMI1271 or CTCE9908.
So, it seemed right to us to remove this useless data and add, instead, the effects produced by conditioned media collected from different cultures of bone marrow stromal cells (BM) or osteoclast precursors (RAW264.7 cells).
A new figure 1 was generated and legend modified accordingly.
Reference of Hattermann et al was cited at line 435 as reference number.
We added the relative reference # 46. Hattermann K, Holzenburg E, Hans F, Lucius R, Held-Feindt J, Mentlein R. Effects of the chemokine CXCL12 and combined internalization of its receptors CXCR4 and CXCR7 in human MCF-7 breast cancer cells. Cell Tissue Res. 2014 Jul;357(1):253-66.
Question 11, Line 203. (D) should be (C)
This was changed accordingly. Figure was modified.
Question 12. Line194-195: that data need to be shown.
This was changed accordingly adding the new panels 2E and 2F
Question 13. Figure 2. As the authors cited in ref. 34, DTX increases CXCR4 had been reported in prostate cancer cells. The novelty and impact of this manuscript should be significantly enhanced if the authors could test whether the increased CXCR4 could be rescued by addition of the three drugs.
I agree with this reviewer when states that DTX increases CXCR4 had been reported in prostate cancer cells.
In a previous report was, indeed, demonstrated that the sensitivity to DTX was increased by administration of AMD3100 (plerixafor), a CXCR4 antagonist [45]. So our report could seem to have little novelties and impact.
The study of Domaska and co-workers [45] investigated whether inhibition of CXCR4, with the specific inhibitor AMD3100, sensitizes human prostate cancer cells to DTX by using ONLY a subcutaneous xenograft model. These researchers showed that a combination of DTX and AMD3100 exerted increased antitumor effect compared with DTX alone concluding that CXCR4 inhibition chemo-sensitizes prostate cancer cells. This report merely suggested the potential of CXCR4 inhibitors as chemosensitizing agents. Differently, we analyze the efficacy of single or dual CXCR/E-selectin antagonists also in bone microenvironment comparing the single to dual activity in the reduction of bone lysis and intra-bone growth. Differently, we do not agree with this reviewer when suggest to test whether the increased CXCR4 could be rescued by addition of the three drugs. This, in our opinion, is only a hypothetical regulatory effect of CXCR4 and HECA-452 antagonists on respective targets in vitro. We could at most expect, as indicated to reply to the question 10, that GMI1359, CTCE9908 and gMI1271, induce internalization, degradation and recycling in the membrane of the specific targeted receptor. These effects could be evident in the sensitive lines and in those resistant to the docetaxel.
These statements are included in the new version at lines 637-646
This epiphenomenon reveals to have scarce biological significance,
The novelty was however maintained by the study of effectiveness in bone microenvironment.
The situation changes, instead, if the same consideration is done in vivo as the inhibition of CXCR4, for example, can induce decreased expression of receptors through tumor-mediated (direct effects on tumor cells) and microenvironment-mediated (indirect effects) as changes in the recruitment of myeloid cells that regulate the inflammatory status of a treated or not tumor.
So, in the new version of figure 4 (panel E) we demonstrated that the in vivo administration of GMI1359 and CTCE9908 but not GMI1271, reduced significantly the levels of CXCR4 suggesting, inter alia, that GMI1359 and CTCE9908 are working on CXCR4. This is in agreement with what has been shown in vitro after administration of plerixafor. This agent was able to reduce CXCR4 [45] in breast cancer cell lines. Although the reduction of CXCR4 levels could be evident also in xenografts resistant to DTX (data not shown), this does not explain how sensitization to DTX may occurs. Therapeutic administration of CXCR4 antagonists may also reduce the levels of CXCR ligands down-modulating CXCR4 effects. In addition, the increased expression of CXCR4 in DTX treated may be due also through a selection of aggressive/resistant cell clones and not by an increased expression of protein. In this argumentation we did not want to introduce the recruitment of the cancer stem cells as responsible for the recurrence / pharmacological resistance to the DTX but an important role can be played by this cellular population that is commonly part of a tumor.
See lines 671-681
Question 14. Experiments testing the involvement of E-Selectin, its ligand (noticed the authors mentioned in Line 224-230), or its downstream signaling components in either the tumor microenvironment or PCa cancer cells are needed to explain GM1359, the dual CXCR4/E-selectin antagonist, had better effect than the individual antagonist.
We thank reviewer for her/his comment. We know that circulating prostate cancer cells preferentially roll and adhere on bone marrow vascular endothelial cells, where abundant E-selectin and stromal cell-derived factor 1 (SDF-1) are expressed, subsequently initiating a cascade of activation events that eventually lead to the development of metastases. So, CXCR4 is mainly expressed in the tumor cells and its ligand in bone marrow endothelial (BME) cells. Differently E selectin is expressed in BME cells and its ligands is expressed in tumor cells. In this report we analyzed the effects of dual E-selectin/CXCR antagonist GMI1359 on the DTX effectiveness. GMI1359 effects were compared to those observed with GMI1271 (sole E-selectin) and CTCE9908 (sole CXCR4) antagonists. GMI1271, differently to GMI1359 and CTCE.9908 show no significant contribution in DTX sensitivity, as indicated at lines 2241-230 of the old version of paper. Experiments testing the involvement of E-Selectin and its ligand in the GMI1359 increased effectiveness respect the sole CXCR4 antagonism was not analyzed in this report because argument of another report in the final writing phase which is aimed at evaluating the mechanisms underlying the invasion of the extracellular matrix (ECM) in the primary tumor (i.e. through the orthotopic intra-prostate model by using DU145 cells [experiments in progress and performed in collaboration with Dr Nadia Zaffaroni of the Milan Institute of Tumors] as well as in the adhesion and transmigration of the circulating tumor cells in models in vitro of endothelial cell transmigration. This further report will analyze the role of microenvironment through CAF and bone marrow stromal cells used alone or in combination with tumor cells in the analyses of the metastatization process. So, although GMI1271 was not able to increase the effectiveness of GIi1359 versus docetaxel, this compound is able to modulate significant aspect of local and distant diffusion contributing significantly to the GMI1359 effectiveness. We can further say that we have tested CTCE-9908 in combination with GMI1271 and we have seen that the combination is better than single doses and results similar to GMI1359.
These statements are added in the discussion at lines 745-750.
Question 15. Line 239 the authors wrote “… suggesting additive and synergistic interaction..” Could this be tested by further calculation based on the IC50?
Usually combination indices calculation was made accordingly to the formula.
CI=(D)1/(Dx)1+(D)2/(Dx)2+ alpha(D)1(D)2/(DX)1 (Dx)2. where (Dx)1= dose of drug 1 to produce 50% cell kill alone; (D)1= dose of drug 1 to produce 50% cell kill in combination with (D)2. (Dx)2= dose of drug 2 to produce 50% cell kill alone; (D)2= dose of drug 2 to produce 50% cell kill in combination with (D)1; and a= 0 for mutually exclusive or 1 for mutually nonexclusive modes of drug action. CI > 1.3 indicates antagonism, CI = 1.1 to 1.3 moderate antagonism, CI = 0.9 to 1.1 additive effect, CI = 0.8 to 0.9 slight synergism, CI = 0.6 to 0.8 moderate synergism, CI = 0.4 to 0.6 synergism, and CI = 0.2 to 0.4 strong synergism.
This was added in MM section at lines 230-237 OK
Question 16. Figure 4 and Figure 5 showed the data with 22Rv1 in subcutaneous tumor models, what’s the rational for the study? Please explain in Result.
As indicated previously the aim of this report was to compare two models of castration resistant PCa showing bone tropism (PC3) and non bone tropism (22rv1). The 22RV1 cells were able to induce mixed osteosclerotic/osteolytic lesions resulting a suitable murine model which is more closed to human situation. These cells are, however, unable to give bone metastases in mice when injected by intra-cardiac way and, moreover, show also a low rate of bone engraftment [31] when injected in the tibia. Nevertheless 22rv1 cells are able to modify massively tumor microenvironment increasing strongly angiogenesis and inflammation. Angiogenesis and inflammation are, as mentioned above, two pathogenetic and progression tumor mechanisms having CXCR4 and E-selectin as major players. Both receptors, indeed, modulate the recruitment and differentiation of endothelial precursors and circulating monocytes [32], sustain Tumor Associated Macrophage (TAM)-dependent inflammation, induce chemotaxis of human endothelial cells [33] in vitro and modulate angiogenesis and vasculogenesis in vivo [34]. The 22rv1 subcutaneous xenograft model was chosen also why despite being locally aggressive it maintained very low (or absent) metastatic potency of cells. In this manner we maintained separated tumor growth and bone metastatic potential.
These statements were added both in the introduction and discussion (see also reply to question 8)
Question 17. The values of the graphs were exactly the same to the respected groups between Figure 4 and figure 5, as well as among figure 5. Did the authors use the same data generated multiple graphs? Please clarify. If some of the graph were indeed from the same data, it is Important to claim it and combine in one graph.
As noticed by the reviewer, the values of respective controls were the same because were originated by the same experiment. In figure 4 we compare GMI1359 from GMI1271 and CTCE9908 to demonstrated the different potency in the reduction of tumor weight and time to progression. In figure 5 we analyze the effects of DTX alone or in combination with GMI1359, GMI1271 and CTCE9908. So the relative controls (vehicle, GMI1359, GMI1271 and CTC9908) were the same showed in figures 4 and 5.
Although I preferred to maintain two separated figures, Figure 5 and legends were modified and made more readable.
Question 18. Figure 4A. Line 281. So the drug started one week after the tumor cell grafted. A diagram of treatment schedule will be helpful for readers.
OK. The suggestion of reviewer was accepted. A diagram of treatments was added in a new figure4A, accordingly
Question 19. Please clarify at the time of treatment stated, what are the range of the tumor volumes or what are the range the calculated tumor weights?
In agreement with the request of this reviewer, who suggest us to show the range of tumors which were collected at the end of experiments, we replace the histogram chart (in the old figure 4A) with a plot chart (new figure 4B) and added relative information in figure legend.
The author claimed the tumor bearing mice were randomly assigned to different treatment groups.
Yes we randomly assigned tumors (animals) to different treatment. Randomization was performed when subcutaneous tumors reached volumes ranged between 80 and 100 mm3. This was commonly obtained 7-10 days after cell injection as indicated at lines 212-214 and 219-230. OK
Question 20. However, it can’t be randomly unless all the tumors were at similar weights. The mice without tumors or with bigger tumors out of the range should be taken out of the experiments.
Yes, the animals were randomized. The volumes were similar (80-100 mm3). Mice with the largest tumors were excluded from the randomization and lowest tumors were included when reached the volume of 80-100 mm3. In this case, the evaluation of growth rate was delayed of the days necessary to obtain the volume of 80-100 mm3. However, to reduced both the differences of single tumor volume measurements in the time linked to differences of engraftment efficacy of the tumor cells as well as the individual variability of the response (even though the mice were inbreed) we introduced the terms "Tumor Progression (TP)" and "Time To Tumor Progression" (TTP). In this manner, the preclinical evaluations on the pharmacological efficacy of a selected compound (evaluated on animals) may be bring to that used in humans. So, we added a specific paragraph in MM section at lines 219-230 OK
Question 21. For the studies with subcutaneous tumors, a picture of the harvested tumors, or the mice bearing the tumors should be included.
In order not to crowd the already full figures, I believe that adding images of the collected tumors from treated or not mice as well as of mice baring tumors is not necessary. In my opinion, indeed, this makes the data very redundant and not offer further information of the effects of selected our compounds. This request could be satisfied in the absence of the data shown in the new figure 5.
Question 22. Line 302. A claim title is needed prior here.
This was made accordingly (line 402)
Question 23. Figure 6. For the intracardiac injections, full body X-ray images should be included, because PC3 cells metastasize to many sites of the bones other than the right tibia.
The reviewer states the right when he says that that PC3 metastasize in other sites to the tibia.
These cells metastasize, indeed, also to the femurs, vertebrae, skull, mandible and anterior legs. So, in order to reply to this question as well as to the some questions of reviewer #2 which require me to provide representative images of the intracardiac injections, we generated a new figure 6 in which we graphically illustrated the intraventricular cell injection model and identified the metastatic sites with osteolysis by total body Xray (figure 6A). In addition in this panel we show a magnification of three more common metastatic sites: (1) Tibia and femur; (2) Lumbar/sacral vertebrae and (3) homerus or scapula (in this case in the gleno-humeral joint). Next, in the new figure 6B we show the main bone locations of the osteolytic model PCb2 in our previous experiments [34-39]. The overall rate of metastatization was up to 95%. The colonization of Tibiae (80%), femurs (40%) or both (95%) was maximal whereas the localization in the other sites (anterior legs, vertebrae, skull or mandible) ranged between 2 and 5%. So we focalized our attention on posterior legs and this was in agreement with several previous literature reports. Nevertheless, we were ready to consider secondary metastasis sites if they were noticed. So, the analysis illustrated above in figure 7 on the time of bone metastases appearance expressed in days, the percentage of animals with bone metastases and the lytic units were performed taking into account all metastatic sites. This information was have considered as indicated at lines 494-499 . So, in my opinion, the insertion in figure 6 of total body X rays, which substitute Xrays from tibiae/femurs, does not add further information since, if we are lucky, we could evidentiate only extremely few further sites having an osteolytic lesion. This would only fill the figure 6 (NEW FIGURE 7) in which are already present 24 Xrays from tibiae/femurs of CTRL mice (12 images) and GMI1359 treated mice (further 12 images). In the new figure 6C we show also the histological intra-osseous localization of PC3 cells indicating the presence of cytokeratin CK18 positive PC3 cells mixed with bone marrow cells. In the same panel we add also CXCR4 and E-selectin expression for PC3 cells and bone marrow cells showing as CXCR4 is expressed mainly in PC3 cells and E-selectin in bone marrow and some tumor cells. The description of this panel was shown at lines 499-504 and in figure legend at lines 511-516, considerations on the percentage of bone localization in secondary sites were shown at lines 513-514 and 497-498.
Question 24, The progression of bone lesions using X-ray should show all the groups, not just the control and GM1359.
Also in this case, considering that the X Rays showed in the new figure 7 are only representative of a time-dependent progression in CTRL and GMI1359 treated animals, we think that the addition of other Xrays for the lesions identified in CTCE-9908 (further 12 images) and GMI1271 (further 12 images total 24 new X rays) not offer further information and would fill the figure 6 (now FIGURE 7) which is already full of itself.
Differently the analyses of lytic units and of levels of mTRAP and CTX-I measured in the serum of offer more informative data measuring qualitatively and biochemically the levels of osteolysis present in treated or not tumor bearing animals.
However, I can deleted the panel B from this figure in a possible second round of review.
Question 25. The tumor-induced bone lesions need to be confirmed with HE staining to clearly show the tumor cells in the bones.
In the new figure 6 C we showed that prostate cancer cells localize in the bone marrow dispersed in the bone marrow stromal cells tumor microenvironment. These cells are positive for CK18 indicating the clear epithelial origins, CXCR4 and E-selectin. The latter antigen results also localized in bone marrow hematopoietic cells. It is necessary to consider, however, that since no bone metabolic disease is present in our animals if an osteolytic lesion is highlighted by XRay we, bona fide, are faced with the presence of a tumor-dependent osteolytic activity.
No HE staining were however performed routinely for all bone metastases.
To reply to this question (as well as to questions 26 and 28) and to be clear in the illustration and discussion of data generated on tumor growth into the bone, we recovered the analysis of bioluminescence which we had performed on intratibial injections of PC3 cells transfected with luciferase (PC3luc). In the old version of paper these analyses were used to demonstrate the presence of PC3luc into the bone and define the bone scores by Yang et al [40]. In the new report, instead, these data are used to quantify the tumor growth of PC3lucs into the tibiae and to compare the possible antiproliferative effects mediated by GMI1359, GMI1271 and CTCE9908 alone or in association with DTX . So we generated proliferation curves for each conditions considering the time dependent variation of bioluminescence units. So, in the new figure 7 we have lowered the tones of statements eliminating from the E panel the term "tumor burden" and left only that of lytic units whereas in figure 8 we add the new data on bioluminescence units.
These statements were added in the text at lines 254-263 and 573-579 OK
Question 26 The tumor volume versus the total or bone volume should be analyzed, especially when the authors claimed that the effects in prostate cancer cell intraosseous growth. Without these data, one can only claim changes of tumor-induced bone lesions, but not the growth of tumor cells.
This reviewer asserts the right when states that tumor volume versus the total or bone volume should be analyzed by histomorphometric analyses or microCT evaluations to claim that the effects in the PCa intra-osseous growth. However, this procedure was not performed here nor can we do it as we do not have available an animal dedicated microCT machine.
see reply to question 25.
The effects of CTCE9908, GMI1359 and GMI1271 on the PC3 tumor growth in the bone were monitored evaluating the variations of photons (bioluminescence arbitrary units) in the time as indicated in the new figure 8 C-E. Figure legend was changed accordingly. A table I was generated by using the data on lytic unit variations in the time (old panel 7C).
Question 27. Figure 7. Treatment schedule indicated the treatment started at day 7, but in the text Line 354-355, the authors wrote that the treatments were started at day 3.
I'm sorry for this mistake: the treatment was started at day 3. So we correct all document. .
Question 28. Please clarify. Again, HE staining should be shown to confirm the growth of tumor cells in the tibiae, and bone morphometry analyses to indicate the changes of tumor growth in the bones.
However, this procedure was not performed here nor can we do it as we do not have available an animal dedicated microCT machine. As indicated above we analyzed tumor growth by the variation in the time of bioluminescence and not by the sole X ray being injected luciferase transfected PC3 cells (PC3luc). See reply to questions 25 and 26.
Question 29. Line 366 and Line 369, the authors made some claims with “data not shown”. However, these data are interesting and important, should be shown, at least as supplemental figures.
As suggested by this reviewer we added in the figure 8 data on serum levels of mTRAP whereas the data on Type I collagen were data not shown.
mTRAP data are in agreement with reduced osteolysis evaluated with lytic units analyses (table I and figure 8G).
Question 30. Discussion. Suggest revise. Currently, it is descriptive and repetitive for their results. Particularly to discuss their results with studies from previous literature and the limitations of this current studies.
Text was widely modified accordingly.

Reviewer 2 Report
The topic of this article is very important. There are no curative therapies for bone metastatic prostate cancer so any inroads into potential therapies should be highlighted. It is well known that CXCR4 is critical in metastatic prostate cancer as well as E-selectin, however combination therapy of the two is very novel and the findings are promising.
There are several things that should to be addressed for this manuscript to be accepted.
1) Please provide rationale for targeting E-selectin, such as expression data (protein and gene). There was little to no difference in expression of HECA-452. The highest levels were in VCaP in DU-145, however, the authors imply that it's increased in the bone metastatic cells. This is not supported by the data provided.
2) If osteoblastic media initiated more CXCR4, why did the authors choose to use the more osteolytic PC3 cells for all of there analysis? What could be the discrepancy in the findings? Please provide evidence of these findings in the osteoblastic C42B and VCaP in comparison to the non-metastatic LNCaP.
3) The authors have shown no evidence that E-selectin is important for prostate cancer growth in this system yet the dual inhibitor has the greatest impact on bone osteolysis. The authors should discuss what other off-target effects may be impacting growth in the presence or absence of DTX.
4) The authors should provide evidence either by blood serum analysis or histology that the inhibitors are actually inhibiting CXCR4 and E-selectin in vivo.
Minor Points to Address:
1) Did the inhibitors impact cell dissemination into the bone? Please provide representative images of the intracardiac injections.
2) What does the hazard ratio data actually mean? Please explain in detail what this represents in reference to the data.
3) The survival curves don’t look like there are significant changes, however, they’re not presented as normal Kaplan meier horizontal plots so they’re difficult to interpret.
Author Response
reviewer # 2
The topic of this article is very important. There are no curative therapies for bone metastatic prostate cancer so any inroads into potential therapies should be highlighted. It is well known that CXCR4 is critical in metastatic prostate cancer as well as E-selectin, however combination therapy of the two is very novel and the findings are promising. because of what has been said
There are several things that should to be addressed for this manuscript to be accepted.
Question 1. Please provide rationale for targeting E-selectin, such as expression data (protein and gene). There was little to no difference in expression of HECA-452. The highest levels were in VCaP in DU-145, however, the authors imply that it's increased in the bone metastatic cells. This is not supported by the data provided.
I thank the reviewer for his/her request to better clarify the rationale on the use of E-selectin antagonists as well as the association with bone metastatization. The expression of HECA-452 was evaluated in series of aggressive PCa cell lines. HECA-452 values stood out on all in two PCa cell lines: VCaP (bone metastasis) and DU145 (brain metastasis). Intermediate values were shown in bone metastatic PCb2, C4-2b and PC3 whereas low levels were found in 22rv1 (primary) and LnCap (lymph nodes). HECA-452 was very low or absent in primary site model as LAPC-4 and CWR22 (data not shown). However, the statistical analyses for HECA-452 and CXCR4 performed in our cell line's cohort was added in figure 1B. These analyses show that the levels of HECA-452 was higher in bone metastatic cells also if not statistically supported (probably due to low number of available cell models). Differently the levels of CXCR4 were significantly higher in bone metastatic models. These considerations are included in the test. at line 55-62, 282-286 and 624-625 and 638-639
Question 2. If osteoblastic media initiated more CXCR4, why did the authors choose to use the more osteolytic PC3 cells for all of there analysis?
(a) choice of cellular models.
The choice to use 22rv1 and PC3 cells born from the purpose to compare in vivo two castration resistant models of which one was highly osteo-tropic and able to colonize and grow into the bone (PC3(PCb2) whereas the second was mainly an locally aggressive model. When injected directly to bone, 22RV1 cells were able to induce in the bone marrow a mixed osteosclerotic/osteolytic lesion resulting a more closed to human model. These cells are, however, unable to give bone metastases in mice when injected by intracardiac via. and show also a low rate of engraftment [43] when intra-tibia injected. 22rv1 cells are able to modify massively tumor microenvironment through elevated angiogenesis and inflammation.
In addition, PC3 cells showed high levels of CXCR4 while 22rv1 had a low CXCR4 content. The subcutaneous 22rv1 xenograft model was choose also why maintained very low (or absent) metastatic potency of cells. For example the intra-prostate injection of 22rv1cells could result in a low metastatic visceral engraftments.
Although C4-2B and VCaP cells induce osteo-inductive bone lesions (osteosclerosis), bone lesions from human patients are mixed with osteolytic and ostesclerotic areas present in the same tissue sample. In the most advanced metastatic stages, however, osteolysis take place to osteosclerosis on the basis of changes in the equilibrium between osteosclerotic and osteolytic factors (i.e. Wnt/DKK-1 activity and RANKL production). So, C4-2B and VCaP cells are cellular models representative of a clinically less relevant skeletal event (osteosclerosis). Osteolysis is, instead, the more clinically relevant skeletal event which is associated to bone fractures, that a clinician must counteract to improve the patient's quality of life. It is no coincidence that metastatic advanced stages of PCa are treated with Zometa, which is a drug with anti-osteolytic activity. So, even if I will be repetitive, osteolysis is the event that clinically predominates in the most advanced stages leading to the main problems (bone fractures) and that considerably impact on the quality of life of the patient: So, if I reduce osteolysis, I also reduce bone fractures. So, we have chosen to use PC3 and PCb2 as models for this study as they have a high rate of bone engraftment which was about 70% for PC3 and superior up to 90% for the more bone tropic PCb2. With this choice we have put ourselves in the worst condition of work because in these models the progression to osteolysis and fractures is very rapid and a reduction of tumor growth in the bone can have strong repercussions in terms of percentage, magnitude and manifestation of osteolytic lesions. In TAB I (only for this reviewer) we show the bone engraftment rates of different PCa cell lines. Data shown in this table are extrapolated from literature and our previous experiences.
Some of these statements are added mainly at lines 76-117 but some of these concept are dispersed in the new text.
Table I
we add also reference 43. Henry MD1, Silva MD, Wen S, Siebert E, Solin E, Chandra S, Worland PJ. Spiculated periosteal response induced by intraosseous injection of 22Rv1 prostate cancer cells resembles subset of bone metastases in prostate cancer patients. Prostate. 2005 Dec 1;65(4):347-54.
What could be the discrepancy in the findings?
Please provide evidence of these findings in the osteoblastic C42B and VCaP in comparison to the non-metastatic LNCaP.
I thank the reviewer for this request which allowed us to better define the effects of antagonists of CXCR4 and E-sectin we have been able to evaluate.
(b) discrepancy of findings by using osteosclerotic models.
We did not exclude this event since it has been demonstrated, in recent reports [51], that the complex structure of bone metastasis, consisting of cancer cells and numerous stroma cells, may have different molecular arrangements in presence of osteo-sclerosis or osteolysis. In this study, authors indicated that osteolytic cancer cells (PC-3 and MDA-MB231) induced transcriptome changes in the bone/bone marrow microenvironment (stroma). This stroma transcriptome differed from the stroma transcriptome of osteo-inductive cancer cells (VCaP). The authors found that differences are observed not in the "vascular/axon guidance" process, in the morphology of vessels which allow the establishment of different vascular niches representing functionally different hematopoietic stem cell niches. This finding suggests different growth requirements of osteolytic and osteo-inductive cancer cells and the need for a differential anti-angiogenic strategy to inhibit tumor growth in osteolytic and osteoblastic bone metastasis.
This was discussed widely at lines 724-735
However we would like to study osteolytic lesions which are the major clinic problem to treat advance and metastatic PCa. The study of the role of CXCR4 and E-selectin in the osteolysis, however, offers the advantage to treat a clinically prevalent and important bone lesion leading to fracture.
(c) Please provide evidence of these findings in the osteoblastic C42B and VCaP in comparison to the non-metastatic LNCaP.
This was not provided here. we show only that C4-2B cells show a good sensitivity to SDF1a which is able to increase IC50 values for DTX (figure 3B).
However, no bone localization was found after intraventricular cell injection.
We observed that 22rv1 cells increased weakly their CXCR4 expression when triggered in vitro with CMs derived from osteoblast cultures also if the CXCR4 induction was more evident when these cells were stimulated with CM of stromal bone marrow cells or of osteoclast precursors such as RAW264.7 cells (murine macrophages). These data are added in figure 1 D and into the text at lines 754-765
In figure 2B-E we compared the role of SDF1 (present in the bone microenvironment) in PC3 (bone metastatic, osteolytic), DU145 (brain metastatic), 22rv1 (non bone metastatic, but able, as state above, to elicit osteosclerotic and osteolytic bone reactions when injected into the tibiae) and C4-2B (bone metastatic, osteosclerotic). We demonstrate how the IC50 values for the DTX increased by 1.25 times for the PC3, 1.41 times for the C4-2B, 1.48 times for the DU145 and 3.0 times for the 22rv1. This last cell line showing mixed bone reactions results to be the more sensitive to SDF1 whereas cells with osteolytic or osteosclerotic ability showed similar increments. GMI1359 appeared to be more active in PC3 with a reduction of 4.59 times while the reduction values for the other lines were similar and ranged from 1.37 to 1.58 times. GMI1359 was therefore more active in cells with osteolytic ability.
The data suggest that the responses to GMI1359 was higher in cells with ability to elicit osteolysis.
Although LnCaP cells are Androgen Receptor (AR) positive, result to be androgen dependent cells and androgens may coordinate the expression of CXCR4 promoting CXCL12/CXCR4-mediated cell motility in these cells, we did not use LnCaP cells (as suggested by this reviewer) since the protein levels of CXCR4 were very low or absent and SDF1 was unable to increase the CXCR4 protein expression in these cells. Our choice derived also by their low invasive capacities, adhesion to the human umbilical vein endothelial cell monolayer and trans-endothelium migration (TEM). Altogether these characteristics represent a pre-requisite for bone invasion and growth into the bones. It has been also demonstrated that SDF1 increase the NF-kappaB-dependent transcriptional activity in PC-3 cells but not in LNCaP cells; and that SDF-1 increased adhesion and enhanced TEM in PC-3 but not in LnCaP cell.
In human patients, bone metastases are observed in association with castration resistant or hormone independent disease. AR may be expressed but its activation may be independent to androgens. The 22rv1 cells are a good model for AR+/CRPC disease stage being AR positive cells (with a truncated/active AR expression), PTEN positive (but showing PIK3CA mutations with activation of Akt pathways) and androgen sensitive but not dependent.
VCaP cells are AR positive, with a wild type AR, and androgen sensitive.
C4-2B are AR positive (mutated form) and castration resistant. 22rv1 (mixed osteosclerotic/osteolytic) and C4-2B (osteosclerotic) are compared in figure 2 for DTX sensitivity showing similar results.
Add reference
51. Hensel J, Wetterwald A, Temanni R, Keller I, Riether C, van der Pluijm G, Cecchini MG, Thalmann GN. Osteolytic cancer cells induce vascular/axon guidance processes in the bone/bone marrow stroma. Oncotarget. 2018 Jun 22;9(48):28877-28896.
Question 3. The authors have shown no evidence that E-selectin is important for prostate cancer growth in this system yet the dual inhibitor has the greatest impact on bone osteolysis. The authors should discuss what other off-target effects may be impacting growth in the presence or absence of DTX.
When an anti-target compound is used, off-target effects are possible.
The fact that the dual inhibitor (GMI-1359) shows a greater impact on osteolysis could be linked to experimental evidence demonstrating that the SDF-1 augments E-selectin mediated endothelial cell (EC) adhesion and migration in a CXCR4-dependent manner [47]. E-selectin expression was found to be elevated in a wide cohort of endothelial cell (HUVEC and bone marrow derived EC) and precursors [48]. Angiogenesis, however, is altered in the bone marrow as a result of osteolysis, tumor growth and changes in vascular network. Angiogenesis modifications by drug administration may modify the recruitment of precursors for osteoblasts/osteoclasts, reduce/increase the inflammation responsible for tumor growth, allow as well as hinder adhesion and the migration of endothelial and tumor cells to the peri-osteogenic bone marrow matrix. Therefore, the E-selettin-CXCR4 axis can be considered an important pathway in the process of osteolysis and therefore if I singularly block the molecular targets I have only a partial inhibition while with the dual inhibition the event is maximal. Furthermore, in preliminary experiments, we have seen that are actually inhibiting the combination of CTCE-9908 and GMI1271 showed effects quite close to GMI1359 but with a greater toxicity (monitored by body weight variation). So, a series of soluble factors are released in this microenvironments both from tumor cell growth and osteolysis and from the single compounds. These altered behavior may be impacting bone tumor growth in the presence or absence of DTX and to be responsible for off-target apparently effects.
some statements were reported in discussion at lines 694-710 and reference 48
48. Muz B, Bazai HY, Sekula A, Fogler WE, Smith T, Magnani JL, Azab AK. Inhibition of E-selectin or E-selectin together with CXCR4 resensitizes multiple myeloma to treatment [abstract]. In: Proceedings of the American Association for Cancer Research Annual Meeting 2017; 2017 Apr 1-5; Washington, DC. Philadelphia (PA): AACR; Cancer Res 2017;77(13 Suppl):Abstract nr 5005. doi:10.1158/1538-7445.AM2017-5005
In the present report we have not dealt with the phenomena of cell adhesion, migration and invasion mediated by SDF1 in PCa cells triggered with SDF1 or conditioned media harvested from CAF, bone marrow stromal cells or osteoblasts since this is the argument of a second report which is in preparation. I can only anticipate (and if necessary provide the reviewer with the relative data) that GMI1271 modulates E-selectin expression in endothelial cells derived from bone marrow and reduces the migration of these cells after stimulation with SDF1. CTCE-9908 inhibits this phenomenon to a lesser extent but is more active in modulating SDF-dependent tumor cell migration. GMI1359 offers the possibility of maximally impeding both endothelial and tumor cell migration.
In agreement with our data it has been also demonstrated that Dual E-selectin and CXCR4 inhibition reduces tumor growth and metastatic progression in an orthotopic model of osteosarcoma [ref8] and in several other tumor models.
RE8. Ju W, Yeung CL, Mendoza A, Murgai M, Kaczanowska S, Zhu J, Patel S, Stewart DA, Fogler WE, Magnani JL, Kaplan RN. Dual E-selectin and CXCR4 inhibition reduces tumor growth and metastatic progression in an orthotopic model of osteosarcoma [abstract]. In: Proceedings of the American Association for Cancer Research Annual Meeting 2018; 2018 Apr 14-18; Chicago, IL. Philadelphia (PA): AACR; Cancer Res 2018;78(13 Suppl):Abstract nr 5211.
r
Question 4. The authors should provide evidence either by blood serum analysis or histology that the inhibitors are actually inhibiting CXCR4 and E-selectin in vivo.
Enzymatic assays revealed the high affinity and specificity of GMI-1271 for E-selectin and GMI-1359 for both CXCR4 and E-selectin. These compounds do not shown cross specificities and do not inhibit other receptors.
To reply to the request of this reviewer on the fact that the inhibitors are actually inhibiting CXCR4 we add in figure 4 (new panel E) the dot blotting analysis performed on 200 mg of protein extracted from tissue samples harvested from animal treated or not with CTC-9908, GMI1271, GMI1359. Eight tissue samples/treatment were considered in this analyses. As we can see both CTCE-9908 and GMI1359 reduced the amount of CXCR4 and this was indicative that these compound are acting on CXCR4 since the inhibition of this pathway was demonstrated to induce CXCR4 internalization and protein degradation [38]
Regarding GMI 1271, instead, here we have not evaluated if this compound is actually inhibiting the E-selectin since we are confident that this happens. We know, indeed, that serum E-selectin levels, routinely measured in clinical trials with GMI-1271, show a drop in serum after drug administration in all patients indicating that GMI1271 is acting on its target. These results have been disclosed in public forums in poster format [45]. Similar results have obtained in murine models in preclinical models of myeloma [46].
some statements and references are reported in the discussion at lines 636-652
46. Hattermann K, Holzenburg E, Hans F, Lucius R, Held-Feindt J, and Mentlein R. Effects of the chemokine CXCL12 and combined internalization of its receptors CXCR4 and CXCR7 in human MCF-7 breast cancer cells Cell Tissue Res. 2014; 357(1): 253–266. doi: 10.1007/s00441-014-1823-y
45. Daniel J. DeAngelo, Jane L. Liesveld, Brian A Jonas, Michael E O'Dwyer, Dale L. Bixby, John L Magnani, Helen M. Thackray, and Pamela S. Becker. A Phase I/II Study of GMI-1271, a Novel E-Selectin Antagonist, in Combination with Induction Chemotherapy in Relapsed/Refractory and Elderly Previously Untreated Acute Myeloid Leukemia; Results to Date. Blood 2016 128:4049;
52. A Natoni,1 T A G Smith,2 N Keane,1,3 C McEllistrim,3 C Connolly,1,3 A Jha,4 M Andrulis,5,6 E Ellert,6 M S Raab,7,8 S V Glavey,1,3 L Kirkham-McCarthy,1 S K Kumar,9 S C Locatelli-Hoops,2 I Oliva,2 W E Fogler,2 J L Magnani,2 and M E O'Dwyer1,3,* E-selectin ligands recognised by HECA452 induce drug resistance in myeloma, which is overcome by the E-selectin antagonist, GMI-1271. Leukemia. 2017 Dec; 31(12): 2642–2651. doi: 10.1038/leu.2017.123
MINOR POINTS
1. Did the inhibitors impact cell dissemination into the bone?
Our data indicate that, with the treatment protocols and models used in this work, we do not have sufficient data to say that our treatments have an impact on bone metastasis. The Intracardiac model we wanted to mimic the clinical outcome of patients without clinical risk of bone lesions (NO LYTIC LESION). For this purpose the various compounds were administered after 5 days from the inoculation of the cells. In this way the micro-lesion had already been established in the bone. The experiment was therefore not designed for this purpose because if the aim would have been this the most suitable experiment was to treat mice with drugs (and possibly cells in vitro) in advance, inoculate the animals after a couple of days with the cells and continue to treat the animals as scheduled.
Some statements are reported in the discussion at lines 431-435, 674-7678.
2. Please provide representative images of the intracardiac injections.
In the new figure 6 we add details of intracardiac cell injection, protocol and intra-bone tumor growth
3. What does the hazard ratio data actually mean? Please explain in detail what this represents in reference to the data.
In survival analysis, the hazard ratio (HR) is the ratio of the hazard rates corresponding to the conditions described by two levels of an explanatory variable. For example, in a drug study, the treated population may die at twice the rate per unit time as the control population. So, HR is the measure of the effect which we are evaluating. If we would to compare the appearance of a complicancer after a treatment between male and female and we find a HR of 10 this mean that men receiving the same treatment may suffer a certain complication ten times more frequently per unit time than women.
In our experiments in the new figure 4D we evaluated tumor progression (percentage of mice in progression). GMI1359 shows a HR of 17,2 which is more of two fold better of CTCE9908 (HR=7,9) whereas GMI1271 was similar to untreated animals, This treatment differs of 0.2 from the value of HR=1 representing the totally overlap between treated and untreated and what is observed for GMI1271 (HR=1,2).
4. The survival curves don’t look like there are significant changes, however, they’re not presented as normal Kaplan Meyer horizontal plots so they’re difficult to interpret.
OK figures has been changed accordingly.

Round 2
Reviewer 1 Report
Still many errors due to careless preparation of the revision.
Some statements are not accurate, miss leading, and contradictory to their own. For example, the authors argument that the osteolytic model represents a more clinically relevant skeletal event. Later in the paragraph, the authors claimed that 22RV1 cells, induce mixed lesions resulting a suitable murine model which is more closed to human situation.
Rebuttal letter to Reviewer 1 was carelessly and poorly written. Apparently two individual people were addressing different Reviewers. Some questions were not adequately addressed.
Some of the figures were in very poor quality. For example, the figure 6A, C, figure 7B.
It’s not acceptable, in my opinion, making two figures with the exactly the same data. Specifically, the tumor weight and progression data in Figure 4B, C and Figure 5B, C.
Reviewer 2 Report
It appears that the authors are apprehensive to provide key data for bone metastatic prostate cancer.
1) The discussion of the cell lines chosen, although accurate in reference to the bone-mediated phenotype of each line, the authors are upsettingly inaccurate in their dismissal of the importance of osteosclerotic disease. A hallmark of prostate cancer in bone and a major distinguisher from prostate and other osteolytic cancers, such as breast and multiple myeloma. The author's determination that prostate cancer-induced osteogenesis is unimportant and should not be addressed shows a lack of knowledge of the field and the clinical outcomes. As shown in multiple papers, (here's only 2 of many:
https://www.ncbi.nlm.nih.gov/pubmed/17549396
https://www.eusupplements.europeanurology.com/article/S1569-9056(04)00082-X/pdf), prostate cancer osteogenesis is poorly understood and the authors need to provide at least some in vitro data to show how their compounds impact bone-inducing cells.
2) It is insufficient to state that what previous data has shown in reference to drug controls instead of providing it here in this paper. "Regarding GMI 1271, instead, here we have not evaluated if this compound is actually inhibiting the E-selectin since we are confident that this happens. We know, indeed, that serum E-selectin levels, routinely measured in clinical trials with GMI-1271, show a drop in serum after drug administration in all patients indicating that GMI1271 is acting on its target." Please provide the necessary data in this manuscript.